# Adaptive Estimation and Inference in Semi-parametric Heterogeneous Clustered Multitask Learning via Neyman Orthogonality

Hanxiao Chen [1]  Debarghya Mukherjee [1]

## Abstract

We study clustered multitask learning in a semi-parametric setting where tasks share a latent cluster structure in their target parameters but exhibit heterogeneous, potentially infinite-dimensional nuisance components. Such heterogeneity poses a major challenge for existing multitask learning methods, which typically rely on aligned feature spaces or homogeneous task structures. To address this challenge, we propose an *adaptive fused orthogonal estimator* that integrates Neyman-orthogonal losses with data-driven pairwise fusion penalties. Our framework leverages task-specific pilot estimates to calibrate the fusion penalties and combines adaptive aggregation with orthogonalization to mitigate the impact of nuisance-parameter estimation error. Theoretically, we show that the proposed estimator achieves exact recovery of the latent clustering with high probability and attains pooled parametric convergence rates proportional to cluster size. Moreover, we establish asymptotic normality and show that, asymptotically, our estimator matches the performance of an oracle procedure that knows the true clustering in advance. Empirically, we show that the proposed method consistently outperforms strong baselines in various simulation setups. A real-world application to U.S. residential energy consumption demonstrates the effectiveness of our approach in uncovering meaningful regional clustering in electricity price elasticity, showcasing the efficacy of our method.

## 1. Introduction

Multitask learning (MTL) aims to improve statistical efficiency and generalization by jointly learning multiple related tasks (Caruana, 1997; Zhang & Yeung, 2011; Duan

[1]Department of Mathematics and Statistics, Boston University. Correspondence to: Debarghya Mukherjee <mdeb@bu.edu>.

*Proceedings of the 43$^{rd}$ International Conference on Machine Learning*, Seoul, South Korea. PMLR 306, 2026. Copyright 2026 by the author(s).

& Wang, 2023; Bhattacharya et al., 2025). By exploiting shared structure, MTL can reduce variance, mitigate data scarcity, and uncover latent relationships across tasks. However, in many modern applications, tasks are only *partially related*: they may share a common target parameter while differing substantially in auxiliary features, data distributions, or other nuisance components. This heterogeneity poses a fundamental challenge for existing MTL methods, which often assume aligned feature spaces or homogeneous task structures (Zhang & Yeung, 2011; Evgeniou & Pontil, 2004). As a concrete example, arising in causal and policy learning across heterogeneous environments (Imbens & Rubin, 2015; Pearl, 2009), consider estimating the effect of a treatment across multiple hospitals/regions/platforms. While the causal effect itself may be shared across subsets of environments, each environment can involve distinct covariates, data-generating mechanisms, and high-dimensional nuisance functions. Naively pooling all data may lead to invalid inference due to potential model-mismatch across different environments, while estimating each task independently may sacrifice statistical efficiency.

Recent advances in *double machine learning* (DML) address part of this challenge by enabling valid estimation of low-dimensional target parameters in the presence of high-dimensional/nonparametric nuisance components (Chernozhukov et al., 2018; Mackey et al., 2018; Foster & Syrgkanis, 2023; Chernozhukov et al., 2022; Farrell, 2015; Oprescu et al., 2019). By constructing *Neyman-orthogonal* loss/score functions, DML ensures that the first-order error of nuisance estimation effectively does not contribute to the target estimation, yielding a $\sqrt{n}$-consistent and asymptotic normal (CAN) estimator of a finite-dimensional target parameter/smooth functional under mild conditions on the complexity of the nuisance parameter. It has since become a central tool in modern causal inference and semiparametric machine learning (Hays & Raghavan, 2025; Bach et al., 2022; Fuhr et al., 2024). However, the DML approach is a *single-task* procedure, as it neither leverages cross-task similarities directly nor discovers shared structure across multiple environments. Moreover, when per-task sample sizes are limited, DML estimators may suffer from high variance and instability, as observed empirically and theoretically (Fingerhut et al., 2022; Fuhr et al., 2024).

At the same time, *clustered multitask learning* (Jacob et al., 2008; Zhou et al., 2011; Okazaki & Kawano, 2024; Zhou & Zhao, 2015; Murugesan et al., 2017) has been widely studied as a way to capture latent group structure among tasks. Methods such as fusion penalty (Tibshirani et al., 2005; Tibshirani, 2011; Han & Zhang, 2015; Evgeniou & Pontil, 2004; Zhu et al., 2025) and centroid-based regularization (Duan & Wang, 2023) encourage tasks within the same cluster to share parameters, often achieving substantial gains in estimation accuracy by pooling data from related tasks. A recent seminal work by (Duan & Wang, 2023) has developed statistically principled parametric multitask estimators with adaptive clustering guarantees, *under known number of clusters*. Yet these approaches typically assume parametric models and do not accommodate complex or infinite-dimensional task-specific nuisance components. Therefore, when applied naively in semiparametric settings, they may invalidate the inference by entangling nuisance estimation errors with the estimator error of the parameter of interest across tasks.

**Our Contribution.** In this work, we bridge this gap by developing an adaptive semiparametric multitask learning framework that simultaneously (i) discovers and exploits shared structure among task-level targets, (ii) leverages Neyman orthogonality to mitigate the impact of nuisance estimation error, and (iii) establishes asymptotic normality of the estimators of target parameters at a minimax optimal pooled rate to reduce variance and enable valid statistical inference. We consider a multitask setting with $m$ tasks, where each task $j \in [m]$ has a finite-dimensional target parameter $\theta_j^*$ and a task-specific nuisance component $\eta_j^*$, potentially infinite-dimensional and heterogeneous across tasks (they may differ in dimension, smoothness, or other structural parameters). We assume that the target parameters $\{\theta_j^*\}_{j \in [m]}$ admit an unknown clustering structure with $K$ clusters: tasks within the same cluster share similar target parameters. However, nuisances remain unrestricted and may vary from task to task. This formulation naturally captures heterogeneous feature spaces, distributional shift, and task-specific confounding (Zhang & Yeung, 2011; Chernozhukov et al., 2018).

Our method proceeds in two stages. In Stage 1, we obtain task-specific pilot estimates of the target parameters using any conventional (not necessarily orthogonal) loss functions (e.g., treatment effect estimation via outcome regression or IPW, provided the estimator is consistent). These pilot estimators are used solely to quantify task similarity. Inspired by adaptive lasso and data-driven weighting strategies (Zou, 2006), we carefully construct *adaptive pairwise fusion penalties* using those pilot estimates that encourage fusion of target parameters among similar tasks, thereby recovering latent cluster structure while mitigating negative transfer. In Stage 2, we solve a penalized estimation problem using

task-specific orthogonal loss functions (with nuisance parameters estimated via sample splitting), together with the adaptive fusion penalties. We show that this procedure simultaneously recovers the latent clustering and yields target parameter estimators that are CAN at a pooled rate, where each task effectively pools data within its cluster. Crucially, nuisance estimation remains task-local throughout, allowing each task to use its own feature space and learning algorithm, while cross-task interaction occurs only through the target parameters, preserving causal and inferential validity. Our approach is a broad semiparametric modeling framework that covers many widely used statistical settings, including the partial linear model, average treatment effect estimation, causal mediation analysis, and difference-in-differences. We summarize our contributions below:

- We introduce an adaptive multitask learning framework that combines Neyman-orthogonality with data-driven pairwise fusion, enabling principled information sharing across tasks with heterogeneous nuisance structures.
- We establish exact recovery of the latent clusters with high probability and show that the proposed estimator attains pooled parametric rates proportional to cluster size under the growing cluster setup (i.e., number of clusters and machines can grow with number of samples).
- Despite adaptive aggregation and data-dependent regularization, we prove asymptotic normality for each task-level estimator, matching the oracle estimator that knows the true clustering in advance.
- Through extensive simulations and a real-world application to U.S. residential electricity demand, we demonstrate improved estimation accuracy, stability, and interpretable task clustering compared to other multitask and single-task baselines.

**Notations:** For a $p$-dimensional vector $\mathbf{x} = (x_1, \ldots, x_p)^\top$, its $\ell_q$-norm is $\|\mathbf{x}\|_q = \left( \sum_{i=1}^p |x_i|^q \right)^{1/q}$, and its outer product is $\mathbf{x}^{\otimes 2} = \mathbf{x}\mathbf{x}^\top$. For a matrix $A$, $\|A\|_2$ denotes its spectral norm. For matrices $A, B$, we write $A \succeq B$ when $A - B$ is positive semidefinite. For nonzero sequences $a_n$ and $b_n$, $a_n \lesssim b_n$ means there exists $C > 0$ such that $a_n \leq C b_n$; and $a_n \asymp b_n$ means both $a_n \lesssim b_n$ and $a_n \gtrsim b_n$ hold. We use $a_n = o(b_n)$ to denote $|a_n/b_n| \to 0$ as $n \to \infty$, and $a_n = O(b_n)$ to denote $\sup_n |a_n/b_n| < \infty$. For a sequence of random variables $X_n$, $X_n = O_p(a_n)$ means $X_n/a_n$ is stochastically bounded, $X_n = o_p(a_n)$ means $X_n/a_n \to 0$ in probability, and $X_n = \omega_p(a_n)$ means $a_n = o_p(X_n)$. For an integer $m \in \mathbb{N}$, $[m]$ is used to denote the set $\{1, 2, \ldots, m\}$.

## 2. Method: Adaptive Orthogonal Multitask Learning

### 2.1. Problem Setup

We consider $m$ tasks indexed by $j \in [m]$. For each task $j$, we observe a dataset $\mathcal{D}_j = \{Z_{ij}\}_{i \in [n_j]}$, generated from

an unknown distribution $P_j$. Each task is associated with a finite-dimensional *target parameter* $\theta_j^* \in \Theta \subseteq \mathbb{R}^d$ and a *nuisance parameter* $\eta_j^* \in \mathcal{H}_j$. The target parameter $\theta_j^*$ is defined through a population risk minimization problem,

$$\theta_j^* = \arg\min_{\theta \in \Theta} \mathbb{E}_{P_j} \left[ \ell_j^\dagger(\theta, \eta_j^*, Z_j) \right] := \arg\min_{\theta \in \Theta} \mathcal{R}_j^\dagger(\theta, \eta_j^*),$$

where $\ell_j^\dagger(\theta, \eta, Z)$ is a task-specific orthogonal loss function (to be defined later) and $\mathcal{R}_j^\dagger(\theta, \eta)$ is the risk/expected loss where the expectation is taken over $Z \sim P_j = P_{\theta_j^*, \eta_j^*}$. We allow both the nuisance spaces $\mathcal{H}_j$ and the distributions $P_j$ to vary across tasks, accommodating heterogeneous feature spaces and covariate distributions.

**Clustering structure**   We assume that the target parameters exhibit a latent cluster structure: there exists an *unknown partition* $\{S_k\}_{k=1}^K$ of $[m]$ such that $\theta_j^* = \beta_k^*$ for all $j \in S_k$. The number of clusters $K$ and the cluster memberships are unknown. For cluster identifiability, it is assumed that the separation $\|\beta_k^* - \beta_{k'}^*\|_2 \geq \delta$ for $k \neq k'$. Our objective is to adaptively recover this structure, efficiently estimate each $\theta_j^*$, and conduct valid statistical inference. Our method and theory are flexible enough to accommodate mild within-cluster heterogeneity; specifically, we allow $\|\theta_j^* - \beta_k^*\| \leq \xi_k$ for $j \in S_k$; see Section 3 (Theorem 3.7 and 3.8) for details. For clarity of presentation, however, we take $\xi_k = 0$ in the present discussion.

**Neyman-Orthogonal Loss Function.**   Neyman orthogonality plays a central role in semiparametric inference by ensuring that estimation of the target parameter is locally insensitive to unavoidable errors arising from high-dimensional or nonparametric nuisance estimation. The key idea is as follows: consider a loss function $\ell^\dagger(\theta, \eta, Z)$, where $\theta$ denotes the finite-dimensional parameter of interest and $\eta$ represents a (potentially infinite-dimensional) nuisance parameter. Let $D_\eta$ denote the Gâteaux derivative operator, defined by $D_\eta f(\eta)[h] := \frac{d}{dt} f(\eta + th)|_{t=0}$. More generally, $D_\eta^2 f(\eta)[h_1, h_2]$ denotes the second-order derivative applied to directions $h_1$ and $h_2$. We say that the loss $\ell_j^\dagger$ is *Neyman-orthogonal* over a set $\mathcal{T}_j \subseteq \mathcal{H}_j$ if,

$$D_\eta \nabla_\theta \mathbb{E}_{Z \sim P_j} [\ell_j^\dagger(\theta, \eta, Z)] \big|_{(\theta_j^*, \eta_j^*)} [h] = 0, \quad h \in \mathcal{T}_j.$$

This ensures that first-order errors in estimating the nuisance parameter $\eta_j^*$ do not affect the estimation of the target parameter $\theta_j^*$. As a result, $\sqrt{n_j}$-CAN estimation of the target parameter is possible even when $\eta_j^*$ is learned using flexible, nonparametric methods.

## 2.2. Two-Stage Adaptive Orthogonal Estimator

Our estimator combines task-local learning with adaptive multitask aggregation and proceeds in two stages. The workflow is summarized in Algorithm 1.

**Stage 1: Task-local initialization (structure discovery).** The goal of Stage 1 is *not* efficient estimation, but rather

to obtain a coarse and stable notion of similarity between tasks. For each task $j$, we compute an initial estimator

$$\hat{\theta}_j^{\text{init}} = \arg\min_{\theta \in \Theta} \sum_{Z \in \mathcal{D}_j} \ell_j^{\text{init}}(\theta, \hat{\eta}_j^{\text{init}}, Z), \quad (2.1)$$

where $\ell_j^{\text{init}}$ is a possibly non-orthogonal loss and $\hat{\eta}_j^{\text{init}}$ is a precomputed task-local nuisance estimator (may not be rate-optimal). Orthogonality is *not required* at this stage for two reasons: (i) non-orthogonal plug-in losses are often more stable in finite samples, and (ii) we only require consistency at some rate $r(n_j)$; the initial estimators are never used directly for inference, they are only used to construct the pairwise penalty as described below.

**Stage 2: Aggregation via adaptive fusion.**   In the second stage, we perform adaptive multitask aggregation while enforcing Neyman orthogonality to preserve valid inference. For each task $j$, we split the sample $\mathcal{D}_j$ into two parts, $\mathcal{D}_{j,1}$ and $\mathcal{D}_{j,2}$. We estimate $\eta_j$ using $\mathcal{D}_{j,1}$ and then estimate the target parameters using $\mathcal{D}_{j,2}$ by solving the following optimization problem:

$$\hat{\boldsymbol{\theta}} = \arg\min_{\theta_1, \ldots, \theta_m} \sum_{j=1}^m f_j^\dagger(\theta_j, \hat{\eta}_j) + \sum_{1 \leq j' < j \leq m} \lambda_{jj'} \|\theta_j - \theta_{j'}\|_2.$$
$$(2.2)$$

where $f_j^\dagger(\theta_j, \hat{\eta}_j) = \sum_{Z \in \mathcal{D}_{j,2}} \ell_j^\dagger(\theta_j, \hat{\eta}_j, Z)$ denotes the empirical Neyman-orthogonal loss for task $j$ and $\hat{\boldsymbol{\theta}} = \{\hat{\theta}_j\}_{j \in [m]}$. *Importantly, only the target parameters $\theta_j$ are fused across tasks. All nuisance parameters remain task-specific throughout the procedure* and may be constructed using any suitable nonparametric, regularization, or machine-learning method.

**Choice of penalty parameters.**   If the true cluster structure were known, an oracle estimator would enforce $\lambda_{jj'} = \infty$ for $j, j' \in S_k$ and $\lambda_{jj'} = 0$ otherwise. However, since cluster memberships are unknown, we approximate this oracle behavior using the pilot estimates from Stage 1. Specifically, we define

$$w_{jj'} = c_w \|\hat{\theta}_j^{\text{init}} - \hat{\theta}_{j'}^{\text{init}}\|_2^{-\gamma}, \quad (2.3)$$

for some constant $c_w > 0$ and set

$$\lambda_{jj'} = \begin{cases} \varepsilon_n, & w_{jj'} \leq \tau, \\ w_{jj'}, & w_{jj'} > \tau, \end{cases} \quad (2.4)$$

where $c_w, \gamma, \tau, \varepsilon_n \geq 0$ are tuning parameters. Intuitively, tasks with nearly identical pilot estimates are strongly fused, while tasks with distinct pilot estimates receive negligible penalties. Under mild separation conditions, this adaptive weighting scheme recovers the oracle fusion pattern with high probability. Theorem 3.5 requires $\epsilon_n$ to be small, so in practice we recommend setting it to a numerically negligible value arbitrarily close to zero. The theorem also suggests stability over a broad range of $\tau$, which is consistent with our empirical findings. As default hyperparameter choices, we recommend $\gamma \in [2, 4]$ and $c_w \in [0.1, 10]$.

**Remark 2.1** (Cross-fitting). *In Stage 2, we estimate the nuisances $\{\hat{\eta}_j\}_{j\in[m]}$ using $\{\mathcal{D}_{j,1}\}_{j\in[m]}$ and then estimate $\hat{\theta}$ on $\{\mathcal{D}_{j,2}\}_{j\in[m]}$ as in (2.2). A cross-fitted version (Chernozhukov et al., 2018) can be obtained by partitioning each $\mathcal{D}_j$ into two folds, estimating the nuisance functions on one fold and evaluating the corresponding orthogonal losses on the held-out fold, and vice versa. The resulting losses are averaged and minimized to obtain $\hat{\theta}$. For simplicity, we focus on a single-split implementation; the cross-fitted procedure is described in Appendix J.*

---

**Algorithm 1** Adaptive MTL via DML

---

1: **Input:** Datasets $\{\mathcal{D}_j\}_{j=1}^m$; initial losses $\{\ell_j^{\text{init}}\}_{j=1}^m$; orthogonal losses $\{\ell_j^\dagger\}_{j=1}^m$; hyperparameters $c_w, \gamma, \tau, \varepsilon_n \geq 0$.
2: **Output:** Final targets $\{\hat{\theta}_j\}_{j=1}^m$.
3: **for** $j = 1$ **to** $m$ **do**
4:     Construct $\hat{\eta}_j^{\text{init}}$ on $\mathcal{D}_j$ and then find $\hat{\theta}_j^{\text{init}}$ with (2.1).
5:     Draw a random partition $\mathcal{D}_j = \mathcal{D}_{j,1} \cup \mathcal{D}_{j,2}$.
6:     Find estimate $\hat{\eta}_j$ on $\mathcal{D}_{j,1}$.
7: **end for**
8: Jointly estimate $\{\hat{\theta}_j\}_{j=1}^m$ on datasets $\{\mathcal{D}_{j,2}\}_{j=1}^m$ by solving for (2.2).
9: **return** $\{\hat{\theta}_j\}_{j=1}^m$.

---

## 3. Theoretical Analysis

In this section, we establish theoretical guarantees for the proposed method, including cluster recovery, estimation accuracy, and inference. Under standard regularity conditions, we show that our procedure exactly recovers the latent clustering of task-level targets and attains oracle pooled rates within each cluster, while remaining robust to task-specific nuisance estimation errors. These results formalize the benefits of combining Neyman orthogonality with adaptive fusion in heterogeneous multitask settings.

Before stating the main results, we introduce the notation and assumptions used throughout this section. For $M > 0$, let $\mathcal{B}_{\theta,M} = \{\theta' : \|\theta' - \theta\|_2 \leq M\}$ denote an $\ell_2$-ball of radius $M$ centered at $\theta$. Similarly, let $\mathcal{B}_{\eta,M} = \{\eta' : \|\eta - \eta'\|_{\mathcal{H}} \leq M\}$ denote a ball in the (possibly infinite-dimensional) nuisance space under an appropriate norm $\|\cdot\|_{\mathcal{H}}$ (e.g., $L_2(P)$, Hölder, Sobolev, or RKHS norm). Without loss of generality, we assume in this section that $|\mathcal{D}_{j,1}| = |\mathcal{D}_{j,2}| = n_j$ for $j \in [m]$. We define the minimum task sample size as $n_{\min} := \min_{j\in[m]} n_j$. Also, let $N_k := \sum_{j\in S_k} n_j$ denote the pooled sample size across all tasks in cluster $k$, and $N_{\min} = \min_{k\in[K]} N_k$. The size of the $k^{th}$ cluster is $m_k = |S_k|$. Moreover, let $\mathcal{T}_{n_j} \subseteq \mathcal{H}_j$ denote the nuisance realization set, i.e. $\hat{\eta}_j \in \mathcal{T}_{n_j}$ with probability going to 1. As mentioned previously, we use $(\theta_j^*, \eta_j^*)$ to denote the true target and nuisance parameter of $j^{th}$ task, and we have $\theta_j^* = \beta_k^*$ for all tasks in $k^{th}$ cluster. Our theoretical analysis is based on the following assumptions:

**Assumption 3.1** (Loss regularity). *For each $j \in [m]$, $\ell_j^\dagger(\theta, \eta, Z)$ is assumed to be convex in $\theta$ for all $\eta \in \mathcal{B}_{\eta_j^*, M}$ and $Z$. Furthermore, there exist constants $\rho, \kappa, (\sigma_i)_{i=1}^5 > 0$ and $r_1, r_2 > 2$ such that the following holds:*

1. ***Curvature:*** *Let $H_j(\theta) := \nabla_\theta^2 \mathcal{R}_j^\dagger(\theta, \eta_j^*)$ denote the population Hessian. We assume $H_j(\theta)$ satisfies $\rho I \preceq H_j(\theta) \preceq \kappa I$ for all $\theta \in \mathcal{B}_{\theta_j^*, M}$ and $j \in [m]$.*

2. ***Score:*** *The score has mean $0$ and finite $r_1$-moment, i.e., $\nabla_\theta \mathcal{R}_j^\dagger(\theta_j^*, \eta_j^*) = 0$, $\mathbb{E}_{P_j}[\|\nabla_\theta \ell_j^\dagger(\theta_j^*, \eta_j^*, Z_j)\|_2^{r_1}] \leq \sigma_1$.*

3. ***Hessian moment:*** *For any $v \in \mathbb{S}^{d-1}$ and $\theta \in \mathcal{B}_{\theta_j^*, M}$, we have $\mathbb{E}_{P_j}[|\langle v, (\nabla_\theta^2 \ell_j^\dagger(\theta, \eta_j^*, Z_j) - H_j(\theta))v\rangle|^{r_2}] \leq \sigma_2$.*

4. ***Uniform Lipschitzness of Hessian:*** *The Hessian of $\ell_j^\dagger$ is assumed to satisfy a uniform Lipschitz condition in a neighborhood around the true parameter:*

$$\mathbb{E}\left[\sup \frac{\|\nabla_\theta^2 \ell_j^\dagger(\theta, \eta, Z_j) - \nabla_\theta^2 \ell_j^\dagger(\theta', \eta', Z_j)\|_2}{\|\theta - \theta'\|_2 + \|\eta - \eta'\|_{\mathcal{H}_j}}\right] \leq \sigma_3$$

*where the supremum is over $\theta \neq \theta' \in \mathcal{B}_{\theta_j^*, M}$ and $\eta \neq \eta' \in \mathcal{B}_{\eta_j^*, M}$.*

5. ***Orthogonality:*** *For any $\eta \in \mathcal{T}_{n_j}$, define $h = \eta - \eta_j^*$ then the mixed Gâteaux derivative has mean $0$, $D_\eta \nabla_\theta \mathcal{R}_j^\dagger(\theta_j^*, \eta_j^*)[h] = 0$.*

6. ***Bounds on moments:*** *Let $\bar{\eta}_j$ be on the segment between $\eta_j^*$ and $\eta \in \mathcal{T}_{n_j}$, then both $\mathbb{E}_{P_j}\|D_\eta \nabla_\theta \ell_j^\dagger(\theta_j^*, \bar{\eta}_j, Z_j)[h]\|_2^2$ and $\mathbb{E}_{P_j}\|D_\eta^2 \nabla_\theta \ell_j^\dagger(\theta_j^*, \bar{\eta}_j, Z_j)[h, h]\|_2$ are upper bounded by $\sigma_5 \|h\|_{\mathcal{H}_j}^2$.*

**Assumption 3.2** (Nuisance regularity). *There exists a rate function $s(n) = \omega(n^{1/4})$ and $s(n) \lesssim n^{1/2}$ such that for $j \in [m]$, $\mathbb{E}_{P_j}[\|\hat{\eta}_j - \eta_j^*\|_{\mathcal{H}_j}] \leq \sigma_4/s^2(n_j) = o(n^{-1/2})$ for some $\sigma_4 > 0$.*

**Assumption 3.3** (Consistent initial estimators). *The initial estimators $\{\hat{\theta}_j^{\text{init}}\}_{j\in[m]}$ are assumed to satisfy for any $\epsilon > 0$:*

$$\mathbb{P}\left(\forall j \in [m], \ n_j^\alpha \|\hat{\theta}_j^{\text{init}} - \theta_j^*\|_2 < \epsilon\right) \geq 1 - p_\epsilon(n_{\min})$$

*for some $\alpha \in (0, 1/2]$, where $p_\epsilon(n_{\min}) \downarrow 0$ as $n_{\min} \uparrow \infty$.*

**Remark 3.4** (Neyman Near-Orthogonality). *Assumption 3.1(5) requires the risk $\mathcal{R}_j^\dagger$ to be exactly orthogonal with respect to nuisance perturbations. However, this condition can be relaxed to the Neyman near-orthogonality condition (Definition 2.2 of (Chernozhukov et al., 2018)), that requires $\|D_\eta \nabla_\theta \mathcal{R}_j^\dagger(\theta_j^*, \eta_j^*)[\eta - \eta_j^*]\|_2 = o(N_k^{-1/2})$ for all $\eta \in \mathcal{T}_{n_j}$. This weaker condition is sufficient to ensure that any first-order bias induced by nuisance estimation is asymptotically negligible. We impose exact orthogonality, as it simplifies the exposition without affecting the substance of the analysis or the resulting guarantees. Our proofs will work verbatim under this weaker assumption.*

**Discussion of assumptions.** Assumption 3.1 is standard in semiparametric $M$-estimation and is mild in a wide range of practical models. Item (1) imposes uniform strong convexity and smoothness on the population risk *only in a neighborhood of the true parameter*, ensuring identifiability and stability of the target parameters; it holds for GLMs, likelihood-based losses, and squared-error objectives under mild design regularity. Items (2) and (3) require only finite moments of the score and Hessian fluctuations, consequently *substantially weaker than uniform boundedness/sub-gaussian conditions often assumed in the literature, and can accommodate heavy-tailed behavior*. Item (4) is a local smoothness condition controlling uniformly second-order variations of the loss with respect to both the target and nuisance parameters and is *only required in a neighborhood of the truth* (see (Mei et al., 2018; Duan & Wang, 2023)). Item (5) is the standard Neyman orthogonality condition, guaranteeing first-order insensitivity to nuisance estimation error, and can be imposed by construction via orthogonal scores. Item (6) additionally imposes a second-order smoothness condition on the $\theta$-score with respect to $\eta$, which is standard and closely mirrors the conditions in (Foster & Syrgkanis, 2023). Assumption 3.2 requires the nuisance to be estimated at the usual $o_p(n_j^{-1/4})$ rate. Finally, Assumption 3.3 requires only consistency of the initial estimator $\hat{\theta}_j^{\text{init}}$ (as $\alpha$ can be very small), without the need for a $\sqrt{n_j}$ convergence rate.

We now present our first main theorem, which shows that, with high probability, the estimators within the same cluster are fused and achieve an aggregated rate determined by the total sample size of the cluster:

**Theorem 3.5** (Cluster Recovery). *Under Assumptions 3.1–3.3, for any choice of $(\varepsilon_n, \gamma, \tau)$ in the definition of $\{\lambda_{jj'}\}$ that satisfies (i) $n_{\min}^{\alpha\gamma} \geq c_0\, n_{\max}$, (ii) $\varepsilon_n \leq c_1 m^{-2} N_{\min}^{\zeta}$ for any $\zeta < 1/2$, and (iii) $\tau \in (c_2, c_2 n_{\min}^{\alpha\gamma})$, for some constants $c_0, c_1, c_2 > 0$, the estimators $\{\hat{\theta}_j\}_{j \in [m]}$ satisfy the following with probability $1 - t^{-1} - p_{\delta/4}(n_{\min})$:*

*(1) **Exact clustering:** $\hat{\theta}_j = \hat{\theta}_{j'}$ for all $j, j' \in S_k$ and $\hat{\theta}_j \neq \hat{\theta}_{j'}$ whenever $j \in S_k$, $j' \in S_{k'}$, $k \neq k'$.*

*(2) **Oracle rate:** For every $j \in S_k$,*
$$\|\hat{\theta}_j - \theta_j^*\|_2 \leq C(K^{1/r_1} + b_{k,n})N_k^{-1/2}t + \tilde{C}\,N_k^{\zeta-1},$$
*for some constants $C, \tilde{C} > 0$ for all $1 < t < a(n_{\min})$ and for some function $a(n_{\min}) \uparrow \infty$ as $n_{\min} = \omega(m^{c_3})$ for some $c_3 > 0$ mentioned explicitly in the proof, with $b_{k,n} = Km_k^{1/2}\max_{j \in S_k} n_j^{1/2}s(n_j)^{-2}$. In particular, since $\zeta < 1/2$, $\|\hat{\theta}_j - \theta_j^*\|_2 = O_P(N_k^{-1/2})$ under a fixed $K$ and bounded $b_{k,n}$.*

Theorem 3.5 delivers two practical guarantees. First, it establishes *exact clustering*: tasks that share a true target are fused, whereas distinct clusters remain separated. Second, it shows that each task-level estimator attains a *pooled parametric rate*, $\|\hat{\theta}_j - \theta_j^*\|_2 = O_P(N_k^{-1/2})$ for $j \in S_k$, so small

tasks benefit directly from borrowing strength within their cluster if $b_{k,n} = O(1)$. This condition is easily satisfied because it is only slightly stronger than Assumption 3.2. See the proof in Appendix D.1 for details.

In our next theorem, we show that our proposed estimators are asymptotically normal at a pooled rate when $j \in S_k$. Towards that end, define the within-cluster loss of $S_k$ as $F_k^{\dagger}(\beta, \boldsymbol{\eta}_k) = \sum_{j \in S_k} f_j^{\dagger}(\beta, \eta_j)$ with $\boldsymbol{\eta}_k = \{\eta_j\}_{j \in S_k}$.

**Theorem 3.6** (Asymptotic Normality). *Under Assumptions 3.1–3.3, and same conditions in Theorem 3.5, if $K = O(1)$ and $b_{k,n} = o(1)$, $\{\hat{\theta}_j\}_{j \in [m]}$ satisfy*
$$\sqrt{N_k}(\hat{\theta}_j - \theta_j^*) \implies \mathcal{N}\left(0, \Psi_k^{-1}\Omega_k\Psi_k^{-1}\right),$$
*where for $k \in [K]$, the matrix $\Psi_k = \mathbb{E}[N_k^{-1}\nabla^2 F_k^{\dagger}(\beta_k^*, \boldsymbol{\eta}_k^*)]$ and the matrix $\Omega_k = \mathbb{E}[N_k^{-1}\nabla F_k^{\dagger}(\beta_k^*, \boldsymbol{\eta}_k^*)^{\otimes 2}]$.*

The proof of this theorem is deferred to Appendix D.2. The theorem establishes a *pooled* asymptotic normality result: following exact cluster recovery, each estimator $\hat{\theta}_j$ is asymptotically normal at the pooled rate $\sqrt{N_k}$. The limiting variance is the same as that of the oracle estimator that knows the true clustering and uses the same orthogonal losses. In practice, $\Psi_k$ and $\Omega_k$ can be estimated consistently using cluster-wise averages of scores and Hessians with plugging in $(\hat{\theta}_j, \hat{\eta}_j)$, enabling standard Wald-type confidence intervals and hypothesis tests for $\theta_j^*$.

**Near-homogeneous clusters.** We further allow for mild within-cluster heterogeneity. For each cluster $k$, there exists a centroid $\beta_k^*$ such that the target parameter $\theta_j^*$ lies in its neighborhood $\|\theta_j^* - \beta_k^*\|_2 \leq \xi_k$ for $j \in S_k$. Thus, tasks in the same cluster are allowed to fluctuate around a common centroid. Across clusters, we impose the same separation condition as before: for any $k \neq k'$, $\|\beta_k^* - \beta_{k'}^*\|_2 \geq \delta$. Under the above setup, Theorems 3.7 and 3.8 follow as extensions of Theorems 3.5 and 3.6, respectively. The proof can be found in Appendix L.4.

**Theorem 3.7.** *Suppose Assumptions 3.1–3.3 hold, and choose $(\varepsilon_n, \gamma)$ as in Theorem 3.5. Assume further that (i) $\tau \in (c_2, c_2\left\{n_{\min}^{-\alpha} + \xi_{\max}\right\}^{-\gamma})$, and ii) $\xi_k \leq c_\xi \min_{j \in S_k} n_j^{-1/2}$, for some constants $c_2, c_\xi > 0$ and $\xi_{\max} := \max_{k \in [K]} \xi_k$. Then, the following hold with probability $1 - t^{-1} - p_{\delta/4}(n_{\min})$:*

*(1) **Exact clustering:** $\hat{\theta}_j = \hat{\theta}_{j'}$ for all $j, j' \in S_k$ and $\hat{\theta}_j \neq \hat{\theta}_{j'}$ whenever $j \in S_k$, $j' \in S_{k'}$, $k \neq k'$.*

*(2) **Oracle rate:** For every $j \in S_k$,*
$$\|\hat{\theta}_j - \theta_j^*\|_2 \leq C(K^{1/r_1} + b_{k,n})N_k^{-1/2}t + \tilde{C}(N_k^{\zeta-1} + \xi_k),$$
*for all $1 < t < a(n_{\min})$, where $a(\cdot)$ and $b_{k,n}$ are defined as in Theorem 3.5 and $C, \tilde{C} > 0$ are some constants.*

Theorem 3.7 shows that the adaptive fusion procedure continues to recover the latent cluster partition even when the

true clusters are approximately homogeneous. The convergence rate contains the terms in Theorem 3.5 and an additional heterogeneity bias of order $\xi_k$. In particular, since $\zeta < 1/2$, if $b_{k,n} = o(1)$ and $\xi_k = O(N_k^{-1/2})$, then $\|\hat{\theta}_j - \theta_j^*\|_2 = O_P(N_k^{-1/2})$, and the pooled oracle rate is retained. The next result establishes asymptotic normality under the stronger condition that the perturbation $\xi_k$ is asymptotically negligible relative to the pooled rate.

**Theorem 3.8.** *Under Assumptions 3.1–3.3 and same conditions in Theorem 3.7, if $K = O(1)$, $b_{k,n} = o(1)$, and $\xi_k = o(N_k^{-1/2})$ for $k \in [K]$, then for $j \in S_k$*

$$\sqrt{N_k}(\hat{\theta}_j - \theta_j^*) \implies \mathcal{N}\left(0, \tilde{\Psi}_k^{-1} \tilde{\Omega}_k \tilde{\Psi}_k^{-1}\right),$$

*where the matrix $\tilde{\Psi}_k = \mathbb{E}[N_k^{-1} \sum_{j \in S_k} \nabla^2 f_j^\dagger(\theta_j^*, \eta_j^*)]$ and the matrix $\tilde{\Omega}_k = \mathbb{E}[N_k^{-1} \{\sum_{j \in S_k} \nabla f_j^\dagger(\theta_j^*, \eta_j^*)\}^{\otimes 2}]$.*

The additional condition $\xi_k = o(N_k^{-1/2})$ is needed because the adaptive fusion estimator shrinks all tasks in the same cluster toward a common value. When the true parameters are not exactly identical, this shrinkage induces a bias of order $\xi_k$. Hence, for this bias to vanish after $\sqrt{N_k}$ scaling, the within-cluster heterogeneity must be asymptotically smaller than the pooled estimation error. A similar bias term caused by within-cluster shrinkage appears in Theorem 4.4 of Duan & Wang (2023).

# 4. Simulation

We evaluate the proposed adaptive fusion estimator on simulated data under three canonical semiparametric models: (i) the partially linear model (**PLM**), (ii) average treatment effect estimation (**ATE**), and (iii) difference-in-differences (**DID**). In all settings, we employ standard Neyman–orthogonal losses (Robins & Rotnitzky, 1995; Chernozhukov et al., 2018; Sant'Anna & Zhao, 2020; Foster & Syrgkanis, 2023).

## 4.1. Experimental design

**Design overview.** Across all experiments, we simulate $m = 20$ tasks partitioned into $K = 3$ latent clusters, where tasks within a cluster share a common target parameter and differ only in sample size, covariate dimension, and nuisance structure. We vary the cluster separation parameter $\delta \in \{1/3, 2/3, 1\}$ to control task heterogeneity. For each task, we construct a Neyman–orthogonal loss using sample splitting and flexible nuisance estimation, and compare taskwise estimation accuracy and cluster recovery across competing multitask estimators. The details on initial estimators and hyperparameters are provided in Section 4.3.

**Task and cluster structure.** We generate $m = 20$ tasks partitioned into $K = 3$ latent clusters. Each task is assigned uniformly at random to a cluster. Task $j$ has sample size

$n_j = 3200 + 80j$ and covariate dimension $p_j = 5 + j$, with covariates $X_{ji} \sim \mathcal{N}(0, I_{p_j})$. Tasks in cluster $k$ share a common parameter $\theta_j^* = \beta_k^*$, where $\beta_k^* = k\delta - (K+1)\delta/2$. We vary the cluster separation $\delta \in \{1/3, 2/3, 1\}$.

**Common estimation protocol.** For each task, the data are split into two equal halves $\mathcal{D}_j = \mathcal{D}_{j,1} \cup \mathcal{D}_{j,2}$. Nuisance functions are estimated on $\mathcal{D}_{j,1}$ using LightGBM (Ke et al., 2017), and the taskwise target parameter is obtained by minimizing Equation (2.2) on $\mathcal{D}_{j,2}$.

## 4.2. Models

**Partially linear model (PLM).** In the first simulation setup, we consider partially linear model, where we observe $(Y_{ij}, T_{ij}, X_{ij})$ from each task generated as follows:

$$T_{ji} = h_j(X_{ji}) + \nu_{ji}, \quad Y_{ji} = \theta_j^* T_{ji} + g_j(X_{ji}) + \varepsilon_{ji}.$$

The noises are generated as $\nu_{ji}, \varepsilon_{ji} \sim \mathcal{N}(0,1)$, and the non-parametric mean functions are set as:

$$h_j(x) = \tfrac{1}{5} \tanh(\textstyle\sum_r x_r), \ g_j(x) = \textstyle\sum_r (-0.8)^r \sigma(x_r),$$

where $\sigma(x) = 1/(1+e^{-x})$. The parameter of interest is $\{\theta_j^*\}_{j \in [m]}$. Define $m_j(x)$ to be the conditional mean function of $Y$ given $X$ on the $j^{th}$ task. We use $\mathcal{D}_{j,1}$ to estimate $\hat{\eta}_j = (\hat{h}_j, \hat{m}_j)$ by regressing $T$ on $X$ and $Y$ on $X$ respectively. On $\mathcal{D}_{j,2}$, we estimate $\theta_j^*$ using the following orthogonal loss function in Equation (2.2):

$$f_j^\dagger(\theta) = \sum_{i \in \mathcal{D}_{j,2}} \left\{ (Y_{ji} - \hat{m}_j(X_{ji})) - \theta(T_{ji} - \hat{h}_j(X_{ji})) \right\}^2.$$

**Average treatment effect (ATE).** We simulate a standard binary treatment setting in which $\theta_j^*$ corresponds to the average treatment effect under the unconfoundedness assumption. The treatment assignments are generated from the following propensity score:

$$\pi_j(x) = \mathbb{P}(D_{ji} = 1 \mid X_{ji} = x) = \sigma(x_4 x_5 - x_1 x_2)$$

to ensure numerical stability, we clip the propensity score to 0.05-0.95. The responses are generated as:

$$Y_{ji} = \theta_j^* D_{ji} + g_j(X_{ji}) + \varepsilon_{ji}, \ g_j(x) = \textstyle\sum_{r=1}^{p_j} (-0.8)^r \sigma(x_r).$$

It is immediate from the above model that the ATE is $\theta_j^*$ for $j^{th}$ task. Define $m_{aj}(x)$ to be the conditional mean of the $Y$ given $X$ and on the group $D = a$ for $a \in \{0,1\}$. Using $\mathcal{D}_{j,1}$, we estimate these nuisance parameters $\hat{\eta}_j = (\hat{\pi}_j, \hat{m}_{1,j}, \hat{m}_{0,j})$, and then use the second half to construct the following doubly-robust response $\hat{Y}_{ji}$:

$$\hat{Y}_{ji} = \left( \frac{D_{ji}(Y_{ji} - \hat{m}_{1,j})}{\hat{\pi}_j} - \frac{(1-D_{ji})(Y_{ji} - \hat{m}_{0,j})}{1-\hat{\pi}_j} + \hat{m}_{1,j} - \hat{m}_{0,j} \right)(X_{ji}).$$

Therefore, we use $f_j^\dagger(\theta) = \sum_{i \in \mathcal{D}_{j,2}} (\theta - \hat{Y}_{ji})^2$ in Equation (2.2) to estimate $\hat{\theta}_j^*$.

**Difference-in-differences (DID).** We consider a two-period DID design with covariate-dependent nonlinear trends, with $\theta_j^*$ representing the constant treatment effect. Each task consists of two-period observations

$(Y_{ji0}, Y_{ji1}, D_{ji}, X_{ji})$. The treatment assignment $D_{ji}$ is binary and generated using the same propensity score $\pi_j(x)$ as defined in the ATE simulation setup. The responses of two time periods $T = 0$ (pre-treatment) and $T = 1$ (post-treatment) generated as:

$$Y_{ji0} = \mu_{j0}(X_{ji}) + \varepsilon_{ji0}, \quad Y_{ji1} = \theta_j^* D_{ji} + \mu_{j1}(X_{ji}) + \varepsilon_{ji1},$$

with the mean functions being:

$$\mu_{j0}(x) = \sum_r 0.7^r \sigma(x_r), \quad \mu_{j1}(x) = \sum_r (-0.7)^r \sigma(x_r).$$

To estimate $\hat{\theta}_j^*$, we implement the doubly robust DID estimator of (Sant'Anna & Zhao, 2020), which requires the estimation of the nuisance parameters $\hat{\eta}_j = (\hat{\pi}_j, \hat{m}_j)$, where $\hat{m}_j$ is an estimator of $m_j(x)$, the conditional mean function of $\Delta Y_j = Y_{j1} - Y_{j0}$ given $X$. The orthogonal loss function for estimating $\theta_j^*$ takes the form $f_j^\dagger(\theta_j) = a_j(\theta_j - b_j)^2$ for some appropriately defined $(a_j, b_j)$ depending on $\hat{\eta}_j$ (see Appendix F for details).

### 4.3. Estimators

We compare the performance of our proposed method with six different types of estimators of the parameter of interest:

**(i) Task–individual/Personalized:** We estimate $\hat{\theta}_j$ separately for task/machine, i.e. $\hat{\boldsymbol{\theta}}^{\mathrm{per}} = \arg\min_{\boldsymbol{\theta}} \sum_j f_j^\dagger(\theta_j)$.

**(ii) ARMUL:** (Duan & Wang, 2023) can also solve the clustered multi-task learning problem, although *it requires the user to specify the number of clusters $\hat{K}$ as opposed to our method*. Given $\hat{K}$, ARMUL solves the following:

$$(\hat{\boldsymbol{\theta}}_{\hat{K}}^{\mathrm{armul}}, \hat{\boldsymbol{\gamma}}, \hat{\mathbf{c}}) = \arg\min_{\boldsymbol{\theta}, \boldsymbol{\gamma} \in \mathbb{R}^{\hat{K}}, \mathbf{c} \in [\hat{K}]^m} F_{\hat{K}}^{\mathrm{armul}}(\boldsymbol{\theta}, \boldsymbol{\gamma}, \mathbf{c})$$

$$F_{\hat{K}}^{\mathrm{armul}}(\boldsymbol{\theta}, \boldsymbol{\gamma}, \mathbf{c}) = \sum_{j=1}^m f_j^\dagger(\theta_j) + \sum_{j=1}^m \lambda_j \left\| \theta_j - \gamma_{c_j} \right\|_2.$$

Here $c_j \in [\hat{K}]$ assigns task $j$ to a cluster and all $\theta_j$ in a cluster are shrunk towards a common $\gamma_{c_j}$. Following their paper, we set $\lambda_j = C_\lambda n_j^{-1/2}$ and choose $C_\lambda \in \{1, 10, 100\}$ that yields best performance. To assess its sensitivity, we report three versions with $\hat{K} \in \{K-1, K, K+1\}$, where $K$ is the oracle value used in the data–generating process.

**(iii) Cluster norm (CN):** (Jacob et al., 2008) proposes a convex relaxation for clustered multi-task learning via covariance regularization. The estimator is defined as

$$\hat{\boldsymbol{\theta}}^{\mathrm{cn}}, \hat{\Sigma} = \arg\min_{\boldsymbol{\theta}, \Sigma} \sum_{j=1}^m f_j^\dagger(\theta_j) + \lambda \operatorname{tr}(\tilde{\theta} \Sigma^{-1} \tilde{\theta}^\top),$$

subject to $\tilde{\theta} = \theta \Pi$, $\alpha I \preceq \Sigma \preceq \beta I$, and $\operatorname{tr}(\Sigma) = \gamma$, where $\Pi$ is the centering matrix. We set $(\lambda, \alpha, \beta, \gamma) = (0.1, 0.1, 2, 3)$, chosen by grid search over admissible values satisfying the required constraints.

**(iv) Flexible clustering (FC):** (Zhou & Zhao, 2015) introduces a representative-based clustering approach that learns an assignment matrix. In our setting, it reduces to

$$\hat{\boldsymbol{\theta}}^{\mathrm{fc}}, \hat{Z} = \arg\min_{\boldsymbol{\theta}, Z} \sum_{j=1}^m f_j^\dagger(\theta_j) + \frac{\lambda}{2} \sum_{j,j'} Z_{jj'}(\theta_j - \theta_{j'})^2 + \frac{\mu}{2} \|Z\|_{1,2},$$

subject to $Z \geq 0$ and $Z^\top \mathbf{1} = \mathbf{1}$. We tune $\lambda, \mu \in \{0.1, 1, 10\}$ and report the best-performing configuration.

**(v) MeTaG:** (Han & Zhang, 2015) proposes a pairwise fusion estimator of the form

$$\hat{\boldsymbol{\theta}}^{\mathrm{metag}} = \arg\min_{\boldsymbol{\theta}} \sum_{j=1}^m f_j^\dagger(\theta_j) + \lambda \sum_{j,j'} \|\theta_j - \theta_{j'}\|_2.$$

This estimator can be viewed as a special case of (2.2) without adaptive weighting, i.e., using a uniform fusion penalty across all task pairs. In our experiments, we set $\lambda = 0.01$; the effect of this hyperparameter is discussed further below.

**(vi) Adaptive fusion (our proposed method):** Last but not least, we implement our proposed method. In Stage 1, the initial estimator is set to be: $\hat{\boldsymbol{\theta}}^{\mathrm{init}} = \hat{\boldsymbol{\theta}}^{\mathrm{per}}$. Then we minimize the loss as defined in Equation (2.2). For all the experiments, we fix the same set of hyperparameters in (2.4) as follows: $\gamma = 2, c_w = 0.1, \varepsilon_n = 10^{-12}$, and $\tau = 10$.

### 4.4. Results

We now present our simulation results. We repeat each experiment over 100 Monte Carlo runs. Estimation quality is measured by $\mathrm{RMSE} = (m^{-1} \sum_j (\hat{\theta}_j - \theta_j^*)^2)^{1/2}$ while Adjusted Rand index (ARI) (Rand, 1971; Hubert & Arabie, 1985) assesses the agreement between estimated $\{\hat{S}_k\}_{k \in \hat{K}}$ and true cluster $\{S_k\}_{k \in [K]}$ (1 = perfect clustering). See Appendix H for the exact definition of ARI.

**Estimation accuracy.** Figure 1 (left panels of each block) shows that the adaptive estimator $\hat{\boldsymbol{\theta}}^{\mathrm{ada}}$ consistently attains the lowest median RMSE and exhibits a favorable left-skew, indicating both accuracy and stability across all settings. The oracle ARMUL estimator $\hat{\boldsymbol{\theta}}_K^{\mathrm{armul}}$ performs competitively when the number of clusters is correctly specified ($\hat{K} = K$) for the ATE and DID models; however, its performance degrades when (i) applied to the PLM setting, (ii) the cluster count is mis-specified ($K-1$ or $K+1$), or (iii) the clusters are weakly separated ($\delta = 1/3$). In contrast, personalized baseline $\hat{\boldsymbol{\theta}}^{\mathrm{per}}$ is uniformly suboptimal across all regimes due to the absence of cross-task information sharing. $\hat{\boldsymbol{\theta}}^{\mathrm{cn}}$ and $\hat{\boldsymbol{\theta}}^{\mathrm{fc}}$ perform comparably to $\hat{\boldsymbol{\theta}}^{\mathrm{per}}$. As $\delta$ increases, the performance of $\hat{\boldsymbol{\theta}}^{\mathrm{cn}}$ deteriorates, while $\hat{\boldsymbol{\theta}}^{\mathrm{fc}}$ shows some improvement. In contrast, $\hat{\boldsymbol{\theta}}^{\mathrm{metag}}$ fails to accurately estimate the underlying parameters.

**Cluster recovery.** The right panels of Figure 1 report the ARI over 100 simulations. The personalized estimator $\hat{\boldsymbol{\theta}}^{\mathrm{per}}$ exhibits essentially no clustering signal (ARI $\approx 0$), and same conclusion holds also for $\hat{\boldsymbol{\theta}}^{\mathrm{cn}}$ and $\hat{\boldsymbol{\theta}}^{\mathrm{fc}}$. In contrast, the adaptive fusion $\hat{\boldsymbol{\theta}}^{\mathrm{ada}}$ consistently recovers the true partition, achieving ARI $\approx 1$ across all settings without requiring knowledge of the oracle $K$. ARMUL is competitive only when the cluster count is correctly specified: $\hat{\boldsymbol{\theta}}_K^{\mathrm{armul}}$ attains near-perfect ARI, whereas $\hat{\boldsymbol{\theta}}_{K-1}^{\mathrm{armul}}$ under-clusters by merging true groups and $\hat{\boldsymbol{\theta}}_{K+1}^{\mathrm{armul}}$ over-splits them. Although all methods improve as the separation $\delta$ increases, the strong

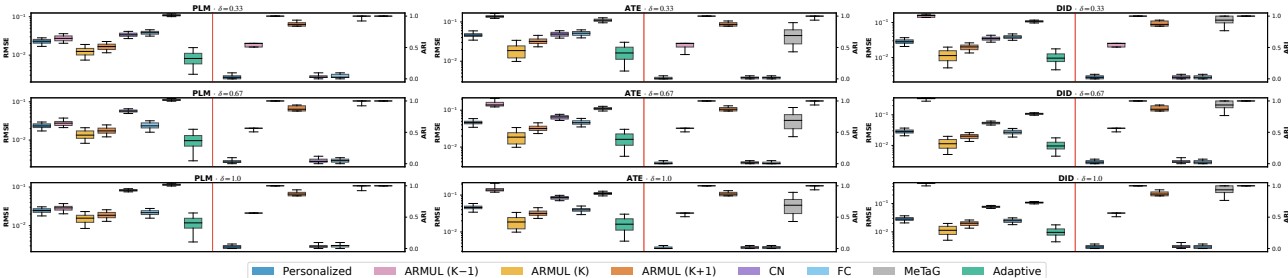

Figure 1. Comparison of estimation accuracy (RMSE on log scale, left panels) and cluster recovery (ARI, right panels) across three models, PLM, ATE, DID, under increasing cluster separation levels $\delta \in \{1/3, 2/3, 1\}$. Each panel shows boxplots over 100 simulations for estimators: Personalized, ARMUL with $K-1$, $K$, and $K+1$ clusters, CN, FC, MeTaG, and the adaptive estimator. Adaptive method achieves the lowest RMSE and near-perfect ARI, demonstrating accurate estimation and robust recovery of the latent clusters.

dependence of ARMUL on $K$ highlights the robustness advantage of our proposed approach. Finally, $\hat{\theta}^{\mathrm{metag}}$ is able to recover the latent clusters only in the PLM setting.

**Distributional behavior.** Figure 2 displays the empirical distributions of the task-level estimators under weak separation ($\delta = 1/3$) for each model. For each method, we concatenate the estimates $\hat{\theta}_j$ from all $m = 20$ tasks across 100 simulation runs and plot the histograms of resulting 2000 estimators. Vertical red dotted lines mark the true cluster parameters $(\beta_1^*, \beta_2^*, \beta_3^*) = (-1/3, 0, 1/3)$. The adaptive estimator $\hat{\theta}^{\mathrm{ada}}$ exhibits substantial variance reduction, with tightly concentrated and approximately normal distributions centered at the true cluster means. In contrast, the misspecified ARMUL estimators $\hat{\theta}_{K-1}^{\mathrm{armul}}$ and $\hat{\theta}_{K+1}^{\mathrm{armul}}$ display asymmetric, multimodal, or highly dispersed shapes, indicating bias and instability. The oracle ARMUL estimator $\hat{\theta}_K^{\mathrm{armul}}$ performs competitively but remains slightly more variable than the adaptive approach, while the Personalized estimator $\hat{\theta}^{\mathrm{per}}$ exhibits substantial dispersion due to the lack of inter-task information sharing. All of $\hat{\theta}^{\mathrm{cn}}$, $\hat{\theta}^{\mathrm{fc}}$, and $\hat{\theta}^{\mathrm{metag}}$ exhibit bias toward a global mean. In particular, $\hat{\theta}^{\mathrm{metag}}$ effectively reduces variance, but at the cost of increased bias. We further empirically verify the asymptotic normality of our estimators and provide the details in Appendix G.

**Fixed vs adaptive penalty.** To isolate the effect of adaptive weighting, we compare the adaptive estimator with a *fixed* baseline that employs a constant pairwise penalty $\lambda_{jj'} \equiv \lambda$ in Equation (2.2). Through extensive simulation (see Appendix I for details), we show that for large $\lambda$ there is a significant bias as it pulls all the estimators to a common centroid, whereas for small $\lambda$, the estimators exhibit significant variability. On the contrary, our adaptive fusion simultaneously achieves accurate estimation and reliable cluster recovery by a careful bias-variance tradeoff.

## 5. Real Data Analysis

**Data and Tasks.** We apply our method on the 2020 Residential Energy Consumption Survey (RECS), a nation-

ally representative dataset (`eia.gov/consumption/residential`) of U.S. housing units that records household demographics, building characteristics, appliance usage, and energy behaviors, to estimate state-level price *elasticities of electricity demand*, which is the percentage change in electricity consumption resulting from a one-percent change in its price. We partition the sample by state and treat each of the 50 U.S. states plus Washington, D.C., as a separate task, yielding $m = 51$ tasks. The raw data contains 18,496 observations and 799 variables. We first remove quantities that are downstream of consumption rather than causal predictors. After filtering variables with substantial missingness and applying a LightGBM-based feature-importance screening step, we retain 50 predictors. These predictors form the feature vector in our empirical analysis. The details of these preprocessing steps are provided in Appendix K and additional results are provided in Appendix M.

**PLM specification.** We estimate the (task-specific) elasticity of electricity consumption using a partially linear model. For each task $j \in [m]$ and unit $i \in [n_j]$ we observe $(Y_{ji}, T_{ji}, X_{ji})$ where

$$Y_{ji} = \log(\mathtt{KWH}) \quad \text{(log annual electricity use)},$$

$$T_{ji} = \log\left(\frac{\mathtt{DOLLAREL}}{\mathtt{KWH}} + 1\right) \quad \text{(log avg. electricity price)},$$

and $X_{ji}$ collects the remaining 50 covariates, where `DOLLAREL` is a code used in the RECS to denote total electricity cost in dollars. We model the data-generating process via the following PLM:

$$
\begin{aligned}
T_{ji} &= h_j(X_{ji}) + \nu_{ji}, & \mathbb{E}[\nu_{ji} \mid X_{ji}] = 0, \\
Y_{ji} &= \theta_j^* T_{ji} + g_j(X_{ji}) + \varepsilon_{ji}, & \mathbb{E}[\varepsilon_{ji} \mid T_{ji}, X_{ji}] = 0,
\end{aligned}
\tag{5.1}
$$

where $\theta_j^*$ is the task-specific slope parameter of interest, while $h_j(x)$ and $m_j(x) = \mathbb{E}[Y_j | X_j = x]$ are nuisance parameters, estimated using LightGBM.

**Results.** We apply our proposed adaptive fusion method to the dataset to investigate regional heterogeneity in electricity price elasticity. Throughout this analysis, hyperparameters are fixed at $\gamma = 2$, $c_w = 0.1$, $\varepsilon_n = 10^{-12}$, and $\tau = 5$. The

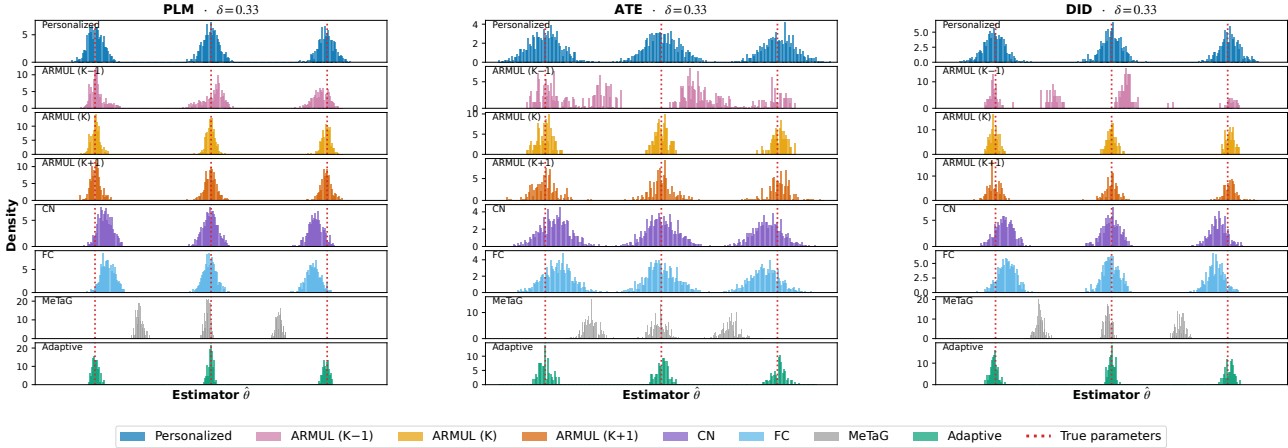

*Figure 2.* Distribution of task-specific estimators $\hat{\theta}_j$ across three models PLM, ATE, DID, at separation level $\delta = 1/3$. Each column corresponds to one model type, and each row to an estimator: Personalized, ARMUL with $K-1$, $K$, and $K+1$ clusters, CN, FC, MeTaG, and the adaptive estimator. Red dotted lines mark the true parameters $\beta_1^* = -1/3$, $\beta_2^* = 0$, and $\beta_3^* = 1/3$.

*Table 1.* Clusters: estimated electricity–price elasticities (mean $\pm$ SE) and member tasks.

| Group | Estimate $\pm$ SE | Member tasks |
|-------|-------------------|--------------|
| Cluster 0 | $-1.138 \pm 0.189$ | VA |
| Cluster 1 | $-0.788 \pm 0.051$ | KY, AL, OK, TN |
| Cluster 2 | $-0.221 \pm 0.009$ | All the other states |

results are summarized in Table 1, with a corresponding geographic visualization shown in Figure 3. As shown by the results, the method identifies three distinct clusters of state-level price elasticities. All estimated elasticities are negative, consistent with standard demand theory. Cluster 0 isolates Virginia, which exhibits a highly elastic response $(-1.138 \pm 0.189)$. Cluster 1 groups four neighboring Southern states, Kentucky, Alabama, Oklahoma, and Tennessee, with moderately large elasticities around $-0.788 \pm 0.051$. States in Clusters 0 and 1 are hotter and more cooling-intensive, so households spend more on electricity and can adjust usage in the short run (e.g., switching between air conditioning and fans), leading to stronger reductions in consumption when prices rise. The remaining 46 states are assigned to Cluster 2, with a substantially smaller elasticity estimate of $-0.221 \pm 0.009$, indicating comparatively inelastic electricity demand across most of the U.S. The resulting spatial pattern (Figure 3) reveals that heightened price sensitivity is concentrated in warmer Southern regions, while demand responses elsewhere are considerably flatter, broadly aligning with the geographic distribution of climate zones. Overall, this application illustrates that adaptive fusion effectively pools structurally similar tasks while preserving meaningful regional heterogeneity in electricity price responsiveness.

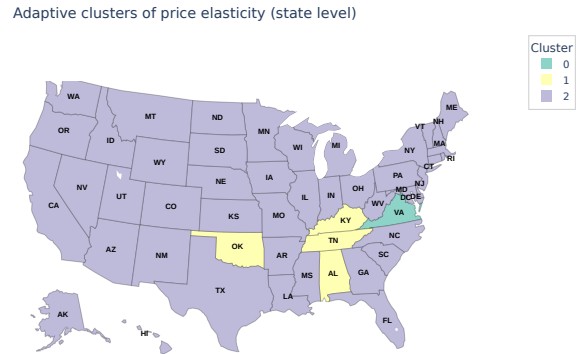

Adaptive clusters of price elasticity (state level)

*Figure 3.* Geographic visualization of state-level elasticity clusters identified by the proposed adaptive fusion method.

# 6. Conclusion

In this paper, we introduce an adaptive semiparametric multitask learning framework that integrates Neyman orthogonality with adaptive pairwise fusion. Our method enables efficient estimation of shared targets under heterogeneous nuisance structures, achieving exact cluster recovery, pooled-rate accuracy, and asymptotic normality at a pooled $\sqrt{N_k}$ rate. Empirically, the adaptive estimator demonstrates superior performance relative to strong baselines across various models and yields interpretable, statistically significant clustering in real data, illustrating its utility as both a predictive and inferential tool. However, several open directions remain. Our theory characterizes performance in terms of sample size, but a finer analysis of the dependence on the cluster separation would be valuable. Extending the framework to high-dimensional regimes, where the target dimension grows with or exceeds the sample size, is another promising direction for future work.

## Impact Statement

This work provides a framework for borrowing strength across heterogeneous tasks while preserving task-specific nuisance structure. The learned clusters can serve as useful data-driven summaries of similarity across environments and may support more stable estimation and inference in applications such as causal inference and policy analysis. At the same time, their interpretation should account for the strength of the clustering signal: when within-cluster heterogeneity is relatively large, cluster separation is weak, or pilot and nuisance estimates are noisy, the estimated partition may be sensitive to preprocessing choices, nuisance learners, and tuning parameters. Our theory partially addresses this in a near-homogeneous setting, showing that the inference remains valid when heterogeneity is sufficiently small. By pooling only low-dimensional target parameters while keeping nuisance components task-specific and fully heterogeneous, the proposed adaptive fusion mitigates negative transfer compared with naive global pooling, although some caution is still warranted when the clustering assumption is weakly supported by the data. In practice, we recommend complementing the estimated clusters with sensitivity analyses over nuisance learners and tuning parameters, as well as validation based on domain knowledge (if available), before drawing substantive conclusions.

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

## A. Roadmap on the proof

Since all the losses mentioned in the proof are Neyman orthogonal, we drop their superscript in Equation (2.2) for readability and use shorthand notation $f_j := f_j^\dagger$. Define $F_k(\beta, \boldsymbol{\eta}_k) = \sum_{j \in S_k} f_j(\beta, \eta_j)$ with $\boldsymbol{\eta}_k = \{\eta_j\}_{j \in S_k}$, $N_k = \sum_{j \in S_k} n_j$, and $\Lambda_{kk'} = \sum_{j \in S_k} \sum_{j' \in S_{k'}} \lambda_{jj'}$. Recall that the proposed adaptive fusion estimator is

$$(\hat{\theta}_1, \ldots, \hat{\theta}_m) = \arg\min_{\theta_1, \ldots, \theta_m} \sum_{j \in [m]} \left\{ f_j(\theta_j, \hat{\eta}_j) + \sum_{j' \neq j, j' \in [m]} \lambda_{jj'} \|\theta_j - \theta_{j'}\|_2 / 2 \right\};$$

To facilitate our analysis, we define two other sets of oracle-type estimators; the first one is a collection of reference estimators, defined as:

$$(\hat{\beta}_1, \ldots, \hat{\beta}_K) = \arg\min_{\beta_1, \ldots, \beta_K} \sum_{k \in [K]} \left\{ F_k(\beta_k, \hat{\boldsymbol{\eta}}_k) + \sum_{k' \neq k, k' \in [K]} \Lambda_{kk'} \|\beta_k - \beta_{k'}\|_2 / 2 \right\},$$

and the second one is the oracle estimators defined as:

$$(\tilde{\beta}_1, \ldots, \tilde{\beta}_K) = \arg\min_{\beta_1, \ldots, \beta_K} \sum_{k \in [K]} F_k(\beta_k, \hat{\boldsymbol{\eta}}_k).$$

Note that both estimators can be computed only if we know the clusters beforehand. In our proof, we show that with high probability, $\hat{\theta}_j = \hat{\beta}_k$ for all $j \in S_k$ (i.e., we have exact cluster recovery) and $\|\hat{\beta}_k - \tilde{\beta}_k\|_2 = o_p(N_k^{-1/2})$, implies that $\hat{\theta}_j = \hat{\beta}_k$ is very close to the oracle, which, in turn, would ensure asymptotic normality of the estimators.

We split the rest of the analysis into a *deterministic* part and a *probabilistic* part. Section B works under high-probability events $\mathcal{E}_{1n}$ and $\mathcal{E}_{2n}$, as defined in Definitions B.1 and B.2. On the event $\mathcal{E}_{1n} \cap \mathcal{E}_{2n}$, we show: (i) *exact shrinkage*: for every $j \in S_k$, the estimator $\hat{\theta}_j$ fuses to the reference $\hat{\beta}_k$; and (ii) *oracle approximation*: the reference $\hat{\beta}_k$ converges to the oracle $\tilde{\beta}_k$ at a rate $o(N_k^{-1/2})$. In Section C, we prove that $\mathbb{P}(\mathcal{E}_{1n} \cap \mathcal{E}_{2n}) \to 1$ as $n_{\min} \uparrow \infty$. Therefore, the previous conclusion holds for a set whose probability goes to 1. Finally, Section D assembles these ingredients to prove the main theorems, with auxiliary technical lemmas collected in Section E.

For readability, we adopt the following notational conventions. First, unless otherwise stated, for a loss function of the form $f(\theta, \eta)$, we use $\nabla f$ and $\nabla^2 f$ to denote the gradient and Hessian with respect to the first argument $\theta$. Gateaux derivatives with respect to the nuisance argument $\eta$ are denoted by $D_\eta$ and $D_\eta^2$. Second, the data splits used in (2.2) are written as

$$\mathcal{D}_{j,2} = \{Z_{ji}\}_{i=1}^{n_j}, \qquad j \in [m].$$

Since the nuisance estimators are assumed to be precomputed and to satisfy Assumption 3.2, the subsequent analysis does not require an explicit construction of the first split $\mathcal{D}_{j,1}$. It only relies on the sample-splitting condition that the nuisance estimator is independent of the evaluation sample:

$$\hat{\eta}_j \perp\!\!\!\perp \mathcal{D}_{j,2}.$$

## B. Deterministic analysis

In this section, we study the deterministic behavior of the adaptive estimator under the "good" events $\mathcal{E}_{1n}$ and $\mathcal{E}_{2n}$, as defined in Definitions B.1 and B.2. Throughout this section, we work conditionally on these events. Our goal is to show that, for sufficiently small $\varepsilon_n$ and properly chosen $\tau$, the minimizer $\hat{\theta}_j$ fuses exactly within each true cluster $S_k$ and remains separated across different clusters. The main statement is given in Lemmas B.6 and B.7.

The proof proceeds in several steps. We first establish in Lemma B.3 that the adaptive weights $\lambda_{jj'}$ separate the clusters at the initial stage: intra-cluster weights are large and inter-cluster weights are small. In Lemma B.4 we show that the reference estimates $(\hat{\beta}_1, \ldots, \hat{\beta}_K)$ are well-defined, and satisfies $\hat{\beta}_k - \tilde{\beta}_k = o(N_k^{-1/2})$, where $\tilde{\beta}_k$ is the oracle minimizer. Next, Lemma B.5 shows that under certain conditions/bounds on the penalties, the adaptive/proposed estimates $\{\hat{\theta}_j\}_{j \in [m]}$ collapse to the cluster minimizers, i.e. $\hat{\theta}_j = \hat{\beta}_k$ for all $j \in S_k$. Finally, Lemma B.6 verifies that those bounds on the penalty parameters indeed hold under $\mathcal{E}_{1n}$ and $\mathcal{E}_{2n}$ and Lemma B.7 builds on it to find the rate of $\tilde{\beta}_k$. These arguments rely on several technical lemmas, collected in Section E.

**Definition B.1.** *Define the event $\mathcal{E}_{1n}$ as the collection of inequalities*

$$\left\|\tfrac{1}{n_j}\nabla f_j(\theta_j^*, \hat{\eta}_j)\right\|_2 < \tfrac{\rho}{8}\{M \wedge (\delta/2)\}, \qquad\qquad \forall j \in [m],$$

$$\rho_j I := \tfrac{\rho n_j}{2} I \preceq \nabla^2 f_j(\theta, \hat{\eta}_j) \preceq \tfrac{3\kappa n_j}{2} I =: \kappa_j I, \qquad\qquad \forall \theta \in \mathcal{B}_{\theta_j^*, M}, \ \forall j \in [m],$$

$$\left\|\tfrac{1}{N_k}\nabla F_k(\beta_k^*, \hat{\boldsymbol{\eta}}_k)\right\|_2 < \tfrac{\rho}{8}\{M \wedge (\delta/2)\}, \qquad\qquad \forall k \in [K],$$

$$\left\|\hat{\eta}_j - \eta_j^*\right\|_{\mathcal{H}_j} \le M, \qquad\qquad j \in [m],$$

$$N_k^{-1/2}\left\|\sum_{j \in S_k} D_\eta \nabla f_j(\theta_j^*, \eta_j^*)[\Delta\eta_j]\right\|_2 \le C\sqrt{K}\max_{j \in S_k} s(n_j)^{-1}t \qquad\qquad k \in [K];$$

$$N_k^{-1/2}\left\|\sum_{j \in S_k} D_\eta^2 \nabla f_j(\theta_j^*, \bar{\eta}_j)[\Delta\eta_j, \Delta\eta_j]\right\|_2 \le CK\, m_k^{1/2}\max_{j \in S_k} n_j^{1/2}s(n_j)^{-2}t \qquad\qquad k \in [K];$$

$$N_k^{-1/2}\left\|\sum_{j \in S_k} \nabla f_j(\theta_j^*, \eta_j^*)\right\|_2 \le CK^{1/r_1}t \qquad\qquad k \in [K],$$

*where $\Delta\eta_j = \hat{\eta}_j - \eta_j^*$ and the quantity $t$ satisfies*

$$1 < t < c_2\, m^{-1}\{n_{\min}^{\frac{r_2}{2(r_2+d)}} \wedge s(n_{\min})\} := a(n_{\min}), \qquad c_2 > 0. \tag{B.1}$$

**Definition B.2.** *Define the event $\mathcal{E}_{2n}$ as the requirement on the initial estimators*

$$n_j^\alpha\left\|\hat{\theta}_j^{\text{init}} - \theta_j^*\right\|_2 < \delta/4 \qquad \forall j \in [m],$$

*for some $\alpha \in (0, 1/2]$.*

**Lemma B.3.** *Under event $\mathcal{E}_{2n}$, for $\tau \in \left(c_w\{\tfrac{\delta}{2}\}^{-\gamma}, c_w\{\tfrac{\delta}{2}\}^{-\gamma}n_{\min}^{\alpha\gamma}\right)$, the following hold:*

$$\lambda_{jj'} = \varepsilon_n \qquad\qquad \forall j \in S_k, \forall j' \in S_{k'}, k \neq k' \in [K];$$

$$\lambda_{jj'} = w_{jj'} > c_w\left\{\tfrac{\delta}{2}\, n_{\min}^{-\alpha}\right\}^{-\gamma} \qquad\qquad \forall j \neq j' \in S_k, \forall k \in [K],$$

*where $\lambda_{jj'}, w_{jj'}$ are defined as in Equation (2.4),*

*Proof.* Recall definitions $w_{jj'} = c_w\|\hat{\theta}_j^{\text{init}} - \hat{\theta}_{j'}^{\text{init}}\|_2^{-\gamma}$, and

$$\lambda_{jj'} = \begin{cases} \varepsilon_n, & \text{if } w_{jj'} \le \tau, \\ w_{jj'}, & \text{if } w_{jj'} > \tau, \end{cases} \tag{B.2}$$

$j, j'$ **belong to different clusters.** It follows directly from definition of $\mathcal{E}_{2n}$ that

$$n_j^\alpha\left\|\hat{\theta}_j^{\text{init}} - \theta_j^*\right\|_2 \vee n_{j'}^\alpha\left\|\hat{\theta}_{j'}^{\text{init}} - \theta_{j'}^*\right\|_2 < \frac{\delta}{4}.$$

Recall that $\left\|\theta_j^* - \theta_{j'}^*\right\|_2 \ge \delta$ since they belong to different clusters. We apply triangle inequality and obtain

$$\left\|\hat{\theta}_j^{\text{init}} - \hat{\theta}_{j'}^{\text{init}}\right\|_2 \ge \left\|\theta_j^* - \theta_{j'}^*\right\|_2 - \left\|\hat{\theta}_j^{\text{init}} - \theta_j^*\right\|_2 - \left\|\hat{\theta}_{j'}^{\text{init}} - \theta_{j'}^*\right\|_2 > \delta/2.$$

The definition $w_{jj'} = c_w\|\hat{\theta}_j^{\text{init}} - \hat{\theta}_{j'}^{\text{init}}\|_2^{-\gamma}$ yields the bound $w_{jj'} < c_w(\delta/2)^{-\gamma}$, which further implies that $w_{jj'} < c_w(\delta/2)^{-\gamma} < \tau$. Thus $\lambda_{jj'} = \varepsilon_n$ by definition.

$j, j'$ **belong to the same cluster.** Secondly, consider case when $j, j'$ belong to the same cluster. Triangle inequality and definition of $\mathcal{E}_{2n}$ yield

$$\left\|\hat{\theta}_j^{\text{init}} - \hat{\theta}_{j'}^{\text{init}}\right\|_2 \le \left\|\hat{\theta}_j^{\text{init}} - \theta_j^*\right\|_2 + \left\|\hat{\theta}_{j'}^{\text{init}} - \theta_{j'}^*\right\|_2 < \frac{\delta}{4}\left(n_j^{-\alpha} + n_{j'}^{-\alpha}\right).$$

and thus

$$w_{jj'} > c_w \left\{\frac{\delta}{4}\left(n_j^{-\alpha} + n_{j'}^{-\alpha}\right)\right\}^{-\gamma} \ge c_w \left\{\frac{\delta}{2}n_{\min}^{-\alpha}\right\}^{-\gamma}.$$

Again, this implies that $w_{jj'} > \tau$ in this case, and the resulting $\lambda_{jj'} = w_{jj'}$. □

**Lemma B.4.** *By definition of $\mathcal{E}_{1n}$, for any $k \in [K]$,*

$$\nabla^2 F_k(\beta, \hat{\boldsymbol{\eta}}_k) \succeq \sum_{j \in S_k} \rho_j I, \quad \forall \beta \in \mathcal{B}_{\beta_k^*, M}.$$

*If $\{\Lambda_{kk'}\}_{k \ne k'}$ additionally satisfy*

$$\frac{2\|\nabla F_k(\beta_k^*, \hat{\boldsymbol{\eta}}_k)\|_2 + \sum_{k' \ne k} \Lambda_{kk'}}{\sum_{j \in S_k} \rho_j} < \min\{M, \delta/2\}, \quad \forall k \in [K]. \tag{B.3}$$

*Then we have*

$$\left\|\tilde{\beta}_k - \beta_k^*\right\|_2 \le \frac{\|\nabla F_k(\beta_k^*, \hat{\boldsymbol{\eta}}_k)\|_2}{\sum_{j \in S_k} \rho_j}, \quad \left\|\hat{\beta}_k - \tilde{\beta}_k\right\|_2 \le \frac{\sum_{k' \ne k, k' \in [K]} \Lambda_{kk'}}{\sum_{j \in S_k} \rho_j} \quad \forall k \in [K],$$

*and*

$$\hat{\beta}_k \ne \hat{\beta}_{k'} \quad \forall k \ne k' \in [K].$$

*Proof.* We use the shorthand notation

$$\tilde{\rho}_k := \sum_{j \in S_k} \rho_j.$$

As $F_k(\cdot, \hat{\boldsymbol{\eta}}_k)$ is $\tilde{\rho}_k$-strongly convex on $\mathcal{B}_{\beta_k^*, M}$, by Lemma E.1, we conclude that the oracle minimizer $\tilde{\beta}_k$ satisfies:

$$\left\|\tilde{\beta}_k - \beta_k^*\right\|_2 \le \frac{\|\nabla F_k(\beta_k^*, \hat{\boldsymbol{\eta}}_k)\|_2}{\tilde{\rho}_k}.$$

Let $r_k$ be such that

$$\sum_{k' \in [K] \setminus \{k\}} \frac{\Lambda_{kk'}}{\tilde{\rho}_k} < r_k < M - \frac{2\|\nabla F_k(\beta_k^*, \hat{\boldsymbol{\eta}}_k)\|_2}{\tilde{\rho}_k}$$

and let $x \in \mathcal{B}_{\tilde{\beta}_k, r_k}$. Then

$$\|x - \beta_k^*\|_2 \le \|x - \tilde{\beta}_k\|_2 + \|\tilde{\beta}_k - \beta_k^*\|_2 \le r_k + \frac{\|\nabla F_k(\beta_k^*, \hat{\boldsymbol{\eta}}_k)\|_2}{\tilde{\rho}_k} < M - \frac{\|\nabla F_k(\beta_k^*, \hat{\boldsymbol{\eta}}_k)\|_2}{\tilde{\rho}_k} \le M.$$

This means $\mathcal{B}_{\tilde{\beta}_k, r_k} \subseteq \mathcal{B}_{\beta_k^*, M}$ and thus $F_k$ is strongly convex in $\mathcal{B}_{\tilde{\beta}_k, r_k}$.

Because the penalty part $\sum_{k \in [K]} \sum_{k' \ne k, k' \in [K]} \Lambda_{kk'}\|\beta_k - \beta_{k'}\|_2/2$ is $\sum_{k' \ne k, k' \in [K]} \Lambda_{kk'}$-Lipschitz in $\beta_k$, we can apply Lemma E.2 to conclude that

$$\|\hat{\beta}_k - \tilde{\beta}_k\|_2 \le \sum_{k' \ne k, k' \in [K]} \frac{\Lambda_{kk'}}{\tilde{\rho}_k}.$$

Define

$$R_k = \frac{\|\nabla F_k(\beta_k^*, \hat{\boldsymbol{\eta}}_k)\|_2 + \sum_{k' \ne k, k' \in [K]} \Lambda_{kk'}}{\tilde{\rho}_k}$$

then it holds for any $k$ that $\hat{\beta}_k \in \mathcal{B}_{\beta_k^*, R_k}$. Condition (B.3) implies that $R_k < \delta/2$ and this means $\hat{\beta}_k \in \text{int } \mathcal{B}_{\beta_k^*, \delta/2}$. Moreover, for any $k, k' \in [K]$, because $\|\beta_k^* - \beta_{k'}^*\|_2 \ge \delta$, it follows that $\text{int } \mathcal{B}_{\beta_k^*, \delta/2} \cap \text{int } \mathcal{B}_{\beta_{k'}^*, \delta/2} = \varnothing$. Therefore, $\hat{\beta}_k \ne \hat{\beta}_{k'}$.

□

**Lemma B.5.** *Assume events $\mathcal{E}_{1n}$ and $\mathcal{E}_{2n}$ hold and let $\{\lambda_{jj'}\}_{j\neq j'\in[m]}$ satisfy the property in Lemma B.3. If for any $k \in [K]$ and any index $j_k \in S_k$ with $|S_k| > 1$, it holds that*

$$\frac{2\|\nabla F_k(\beta_k^*, \hat{\boldsymbol{\eta}}_k)\|_2 + \sum_{k'\neq k, k'\in[K]} \Lambda_{kk'}}{\sum_{j\in S_k} \rho_j} < \min\left\{M, \frac{\delta}{2}, \frac{\lambda_{j_k, j} - \|\nabla f_j(\beta_k^*, \hat{\eta}_j)\|_2}{\kappa_j}\right\}, \quad \forall j \in S_k \setminus \{j_k\}, \qquad \text{(B.4)}$$

*then the following hold:*

$$\hat{\beta}_k \neq \hat{\beta}_{k'} \quad \forall k, k' \in [K], k \neq k';$$
$$\hat{\theta}_j = \hat{\beta}_k, \quad j \in S_k.$$

*Proof.* By condition (B.4) and Lemma B.4, we know that

$$\hat{\beta}_k \neq \hat{\beta}_{k'} \quad \forall k \neq k' \in [K]. \qquad \text{(B.5)}$$

For each ordered pair $(k, k')$ with $k \neq k'$, define

$$g_{kk'} \in \partial_{\beta_k} \|\beta_k - \hat{\beta}_{k'}\| \Big|_{\beta_k = \hat{\beta}_k}.$$

Then (B.5) implies $g_{kk'}$ is the unit vector

$$g_{kk'} = \frac{\hat{\beta}_k - \hat{\beta}_{k'}}{\|\hat{\beta}_k - \hat{\beta}_{k'}\|_2}. \qquad \text{(B.6)}$$

The first-order condition for the reference objective in $(\beta_1, \ldots, \beta_K)$ gives

$$\nabla F_k(\hat{\beta}_k, \hat{\boldsymbol{\eta}}_k) + \sum_{k'\neq k} \Lambda_{kk'} g_{kk'} = 0, \qquad \forall k \in [K]. \qquad \text{(B.7)}$$

Summing (B.7) over $k$ and using the symmetry $\Lambda_{kk'} = \Lambda_{k'k}$ and $g_{kk'} = -g_{k'k}$, we obtain

$$\sum_{k\in[K]} \nabla F_k(\hat{\beta}_k, \hat{\boldsymbol{\eta}}_k) = -\sum_{k\in[K]} \sum_{k'\neq k} \Lambda_{kk'} g_{kk'} = 0. \qquad \text{(B.8)}$$

**1) Lower bound on the loss.** For each $k$ and $j \in S_k$, by convexity of $f_j(\cdot, \hat{\eta}_j)$,

$$f_j(\theta_j, \hat{\eta}_j) \geq f_j(\hat{\beta}_k, \hat{\eta}_j) + \langle \theta_j - \hat{\beta}_k, \nabla f_j(\hat{\beta}_k, \hat{\eta}_j) \rangle.$$

Summing over $k$ and $j \in S_k$ gives

$$\sum_{k\in[K]} \sum_{j\in S_k} f_j(\theta_j, \hat{\eta}_j) \geq \sum_{k\in[K]} \sum_{j\in S_k} f_j(\hat{\beta}_k, \hat{\eta}_j) + \sum_{k\in[K]} \sum_{j\in S_k} \langle \theta_j - \hat{\beta}_k, \nabla f_j(\hat{\beta}_k, \hat{\eta}_j) \rangle. \qquad \text{(B.9)}$$

Choose indices

$$(j_1, \ldots, j_K) \in \arg\min_{z_k \in S_k, k\in[K]} \sum_{k\in[K]} \sum_{k'\neq k} \Lambda_{kk'} \|\theta_{z_k} - \theta_{z_{k'}}\|_2, \qquad \text{(B.10)}$$

and decompose the inner-product term as

$$\sum_{k\in[K]} \sum_{j\in S_k} \langle \theta_j - \hat{\beta}_k, \nabla f_j(\hat{\beta}_k, \hat{\eta}_j) \rangle$$
$$= \sum_{k\in[K]} \left[ \langle \theta_{j_k} - \hat{\beta}_k, \nabla f_{j_k}(\hat{\beta}_k, \hat{\eta}_{j_k}) \rangle + \sum_{j\in S_k \setminus \{j_k\}} \langle \theta_j - \hat{\beta}_k, \nabla f_j(\hat{\beta}_k, \hat{\eta}_j) \rangle \right]. \qquad \text{(B.11)}$$

From (B.8),

$$\nabla f_{j_k}(\hat{\beta}_k, \hat{\eta}_{j_k}) = \nabla F_k(\hat{\beta}_k, \hat{\boldsymbol{\eta}}_k) - \sum_{j\in S_k \setminus \{j_k\}} \nabla f_j(\hat{\beta}_k, \hat{\eta}_j) = -\sum_{k'\neq k} \nabla F_{k'}(\hat{\beta}_{k'}, \hat{\boldsymbol{\eta}}_{k'}) - \sum_{j\in S_k \setminus \{j_k\}} \nabla f_j(\hat{\beta}_k, \hat{\eta}_j).$$

Substituting into (B.11) and rearranging,

$$\sum_{k\in[K]}\sum_{j\in S_k}\langle\theta_j-\hat{\beta}_k,\nabla f_j(\hat{\beta}_k,\hat{\eta}_j)\rangle$$

$$=\sum_{k\in[K]}\left[\sum_{j\in S_k\setminus\{j_k\}}\langle\theta_j-\theta_{j_k},\nabla f_j(\hat{\beta}_k,\hat{\eta}_j)\rangle+\sum_{k'\neq k}\langle\hat{\beta}_k-\theta_{j_k},\nabla F_{k'}(\hat{\beta}_{k'},\hat{\boldsymbol{\eta}}_{k'})\rangle\right]$$

$$=:A_1+A_2.$$

**2) Bounds on $A_1$ and $A_2$.** By Lemma B.4 and condition (B.4),

$$\hat{\beta}_k\in\mathcal{B}\left(\beta_k^*,\frac{\|\nabla F_k(\beta_k^*,\hat{\boldsymbol{\eta}}_k)\|_2+\sum_{k'\neq k}\Lambda_{kk'}}{\sum_{j\in S_k}\rho_j}\right)\subseteq\mathcal{B}_{\beta_k^*,M}.$$

Since $\nabla f_j(\cdot,\hat{\eta}_j)$ is $\kappa_j$–Lipschitz on $\mathcal{B}_{\beta_k^*,M}$ due to event $\mathcal{E}_{1n}$, and using again Lemma B.4, for any $j\in S_k$,

$$\|\nabla f_j(\hat{\beta}_k,\hat{\eta}_j)\|_2\leq\|\nabla f_j(\hat{\beta}_k,\hat{\eta}_j)-\nabla f_j(\beta_k^*,\hat{\eta}_j)\|_2+\|\nabla f_j(\beta_k^*,\hat{\eta}_j)\|_2$$

$$\leq\kappa_j\|\hat{\beta}_k-\beta_k^*\|_2+\|\nabla f_j(\beta_k^*,\hat{\eta}_j)\|_2$$

$$\leq\kappa_j\frac{\|\nabla F_k(\beta_k^*,\hat{\boldsymbol{\eta}}_k)\|_2+\sum_{k'\neq k}\Lambda_{kk'}}{\sum_{j\in S_k}\rho_j}+\|\nabla f_j(\beta_k^*,\hat{\eta}_j)\|_2$$

$$<\lambda_{j_k,j},$$

where the last inequality uses (B.4). By Cauchy–Schwarz,

$$A_1=\sum_{k\in[K]}\sum_{j\in S_k\setminus\{j_k\}}\langle\theta_j-\theta_{j_k},\nabla f_j(\hat{\beta}_k,\hat{\eta}_j)\rangle\geq-\sum_{k\in[K]}\sum_{j\in S_k\setminus\{j_k\}}\|\theta_j-\theta_{j_k}\|_2\|\nabla f_j(\hat{\beta}_k,\hat{\eta}_j)\|_2$$

$$>-\sum_{k\in[K]}\sum_{j\in S_k\setminus\{j_k\}}\lambda_{j_k,j}\|\theta_j-\theta_{j_k}\|_2.$$

(B.12)

For $A_2$, use (B.8) and (B.7):

$$A_2=\sum_{k\in[K]}\langle\hat{\beta}_k-\theta_{j_k},-\nabla F_k(\hat{\beta}_k,\hat{\boldsymbol{\eta}}_k)\rangle=\sum_{k\in[K]}\left\langle\hat{\beta}_k-\theta_{j_k},\sum_{k'\neq k}\Lambda_{kk'}g_{kk'}\right\rangle$$

$$=\sum_{k\in[K]}\sum_{k'\neq k}\langle\hat{\beta}_k-\theta_{j_k},\Lambda_{kk'}g_{kk'}\rangle.$$

Using $\Lambda_{kk'}g_{kk'}=-\Lambda_{k'k}g_{k'k}$,

$$A_2=\sum_{k\in[K]}\sum_{k'\neq k}\langle\theta_{j_{k'}}-\theta_{j_k}+\hat{\beta}_k-\hat{\beta}_{k'},\Lambda_{kk'}g_{kk'}\rangle/2$$

$$=\sum_{k\in[K]}\sum_{k'\neq k}\langle\theta_{j_{k'}}-\theta_{j_k},\Lambda_{kk'}g_{kk'}/2\rangle+\sum_{k\in[K]}\sum_{k'\neq k}\Lambda_{kk'}\|\hat{\beta}_k-\hat{\beta}_{k'}\|_2/2$$

$$\geq-\sum_{k\in[K]}\sum_{k'\neq k}\Lambda_{kk'}\|\theta_{j_{k'}}-\theta_{j_k}\|_2/2+\sum_{k\in[K]}\sum_{k'\neq k}\Lambda_{kk'}\|\hat{\beta}_k-\hat{\beta}_{k'}\|_2/2.$$

By the choice of $(j_1,\ldots,j_K)$ in (B.10) and Lemma E.3,

$$\sum_{k\in[K]}\sum_{k'\neq k}\Lambda_{kk'}\|\theta_{j_{k'}}-\theta_{j_k}\|_2\leq\sum_{k\in[K]}\sum_{k'\neq k}\sum_{j\in S_k}\sum_{j'\in S_{k'}}\varepsilon_n\|\theta_j-\theta_{j'}\|_2,$$

and using $\Lambda_{kk'} = m_k m_{k'} \varepsilon_n$ with $m_k = |S_k|$, this gives

$$A_2 \geq -\sum_{k \in [K]} \sum_{k' \neq k} \sum_{j \in S_k} \sum_{j' \in S_{k'}} \lambda_{jj'} \|\theta_j - \theta_{j'}\|_2/2 + \sum_{k \in [K]} \sum_{k' \neq k} \Lambda_{kk'} \|\hat{\beta}_k - \hat{\beta}_{k'}\|_2/2. \qquad \text{(B.13)}$$

**3) Combine with the penalty and identify the minimizer.** Define the full objective

$$L(\theta) := \sum_{j \in [m]} f_j(\theta_j, \hat{\eta}_j) + \sum_{j \in [m]} \sum_{j' \neq j} \lambda_{jj'} \|\theta_j - \theta_{j'}\|_2/2.$$

From (B.9), (B.12) and (B.13),

$$L(\theta) = \sum_{k \in [K]} \sum_{j \in S_k} f_j(\theta_j, \hat{\eta}_j) + \sum_{j \in [m]} \sum_{j' \neq j} \lambda_{jj'} \|\theta_j - \theta_{j'}\|_2/2$$

$$\geq \sum_{k \in [K]} \sum_{j \in S_k} f_j(\hat{\beta}_k, \hat{\eta}_j) + A_1 + A_2 + \sum_{j \in [m]} \sum_{j' \neq j} \lambda_{jj'} \|\theta_j - \theta_{j'}\|_2/2.$$

Insert the bounds on $A_1$ and $A_2$:

$$L(\theta) \geq \sum_{k \in [K]} \sum_{j \in S_k} f_j(\hat{\beta}_k, \hat{\eta}_j) - \sum_{k} \sum_{j \in S_k \setminus \{j_k\}} \lambda_{j_k, j} \|\theta_j - \theta_{j_k}\|_2$$

$$- \sum_{k} \sum_{k' \neq k} \sum_{j \in S_k} \sum_{j' \in S_{k'}} \lambda_{jj'} \|\theta_j - \theta_{j'}\|_2/2 + \sum_{k} \sum_{k' \neq k} \Lambda_{kk'} \|\hat{\beta}_k - \hat{\beta}_{k'}\|_2/2$$

$$+ \sum_{j \in [m]} \sum_{j' \neq j} \lambda_{jj'} \|\theta_j - \theta_{j'}\|_2/2.$$

Now rewrite the last two penalty terms explicitly. The total penalty is

$$\sum_{j \in [m]} \sum_{j' \neq j} \lambda_{jj'} \|\theta_j - \theta_{j'}\|_2/2 = \sum_{k} \sum_{j \in S_k} \sum_{j' \in S_k, j' \neq j} \lambda_{jj'} \|\theta_j - \theta_{j'}\|_2/2 + \sum_{k} \sum_{k' \neq k} \sum_{j \in S_k} \sum_{j' \in S_{k'}} \lambda_{jj'} \|\theta_j - \theta_{j'}\|_2/2.$$

Therefore

$$-\sum_{k} \sum_{k' \neq k} \sum_{j \in S_k} \sum_{j' \in S_{k'}} \lambda_{jj'} \|\theta_j - \theta_{j'}\|_2/2 + \sum_{j \in [m]} \sum_{j' \neq j} \lambda_{jj'} \|\theta_j - \theta_{j'}\|_2/2$$

$$= \sum_{k} \sum_{j \in S_k} \sum_{j' \in S_k, j' \neq j} \lambda_{jj'} \|\theta_j - \theta_{j'}\|_2/2.$$

Hence

$$L(\theta) \geq \sum_{k \in [K]} \sum_{j \in S_k} f_j(\hat{\beta}_k, \hat{\eta}_j) + \sum_{k} \sum_{k' \neq k} \Lambda_{kk'} \|\hat{\beta}_k - \hat{\beta}_{k'}\|_2/2$$

$$- \sum_{k} \sum_{j \in S_k \setminus \{j_k\}} \lambda_{j_k, j} \|\theta_j - \theta_{j_k}\|_2$$

$$+ \sum_{k} \sum_{j \in S_k} \sum_{j' \in S_k, j' \neq j} \lambda_{jj'} \|\theta_j - \theta_{j'}\|_2/2$$

$$= \sum_{k \in [K]} \sum_{j \in S_k} f_j(\hat{\beta}_k, \hat{\eta}_j) + \sum_{k} \sum_{k' \neq k} \Lambda_{kk'} \|\hat{\beta}_k - \hat{\beta}_{k'}\|_2/2$$

$$+ \sum_{k} \sum_{j, j' \in S_k \setminus \{j_k\}, j' \neq j} \lambda_{jj'} \|\theta_j - \theta_{j'}\|_2/2$$

$$\geq \sum_{k \in [K]} \sum_{j \in S_k} f_j(\hat{\beta}_k, \hat{\eta}_j) + \sum_{k} \sum_{k' \neq k} \Lambda_{kk'} \|\hat{\beta}_k - \hat{\beta}_{k'}\|_2/2$$

Now consider the configuration $\hat{\theta}$ defined by $\hat{\theta}_j = \hat{\beta}_k$ for $j \in S_k$. For this configuration, all intra-cluster distances $\|\hat{\theta}_j - \hat{\theta}_{j'}\|_2$ with $j, j' \in S_k$ vanish, and for $j \in S_k$, $j' \in S_{k'}$ with $k \neq k'$ we have $\|\hat{\theta}_j - \hat{\theta}_{j'}\|_2 = \|\hat{\beta}_k - \hat{\beta}_{k'}\|_2$. Then one can easily verify that

$$L(\hat{\theta}) = \sum_{k \in [K]} \sum_{j \in S_k} f_j(\hat{\beta}_k, \hat{\eta}_j) + \sum_k \sum_{k' \neq k} \Lambda_{kk'} \|\hat{\beta}_k - \hat{\beta}_{k'}\|_2 / 2$$

and the lower bound is attained. Therefore, we conclude that $\hat{\theta}_j = \hat{\beta}_k$ for all $j \in S_k$. $\qquad \square$

**Lemma B.6.** *Let events $\mathcal{E}_{1n}$ and $\mathcal{E}_{2n}$ hold and assume $n_{\min}^{\alpha\gamma}/n_{\max} > c_0$ with $c_0 = c_w^{-1} (\delta/2)^\gamma (\rho/8 + 3\kappa/2)\{M \wedge (\delta/2)\}$. If $\varepsilon_n$ is set to be $\varepsilon_n < \min_{k \in [K]} N_k^\zeta \{M \wedge (\delta/2)\} m^{-2} \rho/4$ for $\zeta < 1/2$, then for $\tau \in \left( c_w \{\frac{\delta}{2}\}^{-\gamma}, c_w \{\frac{\delta}{2}\}^{-\gamma} n_{\min}^{\alpha\gamma} \right)$ the following hold*

$$\hat{\beta}_k \neq \hat{\beta}_{k'} \quad \forall k, k' \in [K], k \neq k';$$
$$\hat{\theta}_j = \hat{\beta}_k, \quad j \in S_k;$$
$$\left\| \tilde{\beta}_k - \beta_k^* \right\|_2 \leq \frac{1}{4}\{M \wedge (\delta/2)\}, \quad \forall k \in [K];$$
$$\left\| \hat{\beta}_k - \tilde{\beta}_k \right\|_2 < \tilde{C} N_k^{\zeta-1}, \quad \forall k \in [K],$$

*where $\tilde{C} > 0$ is a constant.*

*Proof.* Recall by definition of $\mathcal{E}_{1n}$ that $\rho_j = \rho n_j/2$, $\kappa_j = 3\kappa n_j/2$, hence $\sum_{j \in S_k} \rho_j = N_k \rho/2$. By Lemma B.5 and definition of $\mathcal{E}_{1n}$, it suffices to show the following statement: for any $k \in [K]$ and any index $j_k \in S_k$, it holds that

$$\frac{2 \left\| \sum_{j \in S_k} \nabla f_j(\beta_k^*, \hat{\eta}_j) \right\|_2 + \sum_{k' \neq k, k' \in [K]} \Lambda_{kk'}}{N_k \rho/2} < \min \left\{ M, \frac{\delta}{2}, \frac{\lambda_{j_k, j} - \|\nabla f_j(\beta_k^*, \hat{\eta}_j)\|_2}{3\kappa n_j/2} \right\}, \quad \forall j \in S_k \setminus \{j_k\}. \quad \text{(B.14)}$$

First consider the RHS of (B.14). By Lemma B.3 and definition of $\mathcal{E}_{1n}$, we have $\lambda_{j_k, j} > c_w \left\{ \frac{\delta}{2} n_{\min}^{-\alpha} \right\}^{-\gamma}$ and $\|\nabla f_j(\beta_k^*, \hat{\eta}_j)\|_2/n_j \leq \frac{\rho}{8}\{M \wedge (\delta/2)\}$. Therefore

$$\frac{\lambda_{j_k, j} - \|\nabla f_j(\beta_k^*, \hat{\eta}_j)\|_2}{3\kappa n_j/2} \geq \frac{2}{3\kappa} \left( c_w \left\{ \frac{\delta}{2} n_{\min}^{-\alpha} \right\}^{-\gamma} / n_j - \frac{\rho}{8}\{M \wedge (\delta/2)\} \right)$$
$$\geq \frac{2}{3\kappa} \left( c_w \left\{ \frac{\delta}{2} \right\}^{-\gamma} n_{\min}^{\alpha\gamma} / n_{\max} - \frac{\rho}{8}\{M \wedge (\delta/2)\} \right).$$

If $n_{\min}^{\alpha\gamma}/n_{\max} > c_0$ with $c_0 = c_w^{-1} (\delta/2)^\gamma (\rho/8 + 3\kappa/2)\{M \wedge (\delta/2)\}$, then it follows that

$$\frac{\lambda_{j_k, j} - \|\nabla f_j(\beta_k^*, \hat{\eta}_j)\|_2}{3\kappa n_j/2} > M \wedge (\delta/2).$$

Therefore we have shown that RHS $= M \wedge (\delta/2)$ under the given condition.

Consider LHS of (B.14). By the condition $\varepsilon_n < \min_{k \in [K]} N_k^\zeta \{M \wedge (\delta/2)\} m^{-2} \rho/4$, it follows from Lemma B.3 that

$$\frac{\sum_{k' \neq k, k' \in [K]} \Lambda_{kk'}}{N_k \rho/2} \leq \frac{m^2 \varepsilon_n}{N_k \rho/2} < \frac{N_k^{\zeta-1}}{2}\{M \wedge (\delta/2)\} \leq \frac{1}{2}\{M \wedge (\delta/2)\}.$$

Also, definition of $\mathcal{E}_{1n}$ implies

$$\frac{2 \|\nabla F_k(\beta_k^*, \hat{\boldsymbol{\eta}}_k)\|_2}{N_k \rho/2} \leq \frac{4}{\rho} \frac{\|\nabla F_k(\beta_k^*, \hat{\boldsymbol{\eta}}_k)\|_2}{N_k} \leq \frac{1}{2}\{M \wedge (\delta/2)\}$$

and thus LHS of (B.14) $< M \wedge (\delta/2) =$ RHS of (B.14). Now we have shown LHS $<$ RHS, and we can apply Lemma B.5 to prove the shrinkage

$$\hat{\beta}_k \neq \hat{\beta}_{k'} \quad \forall k, k' \in [K], k \neq k',$$
$$\hat{\theta}_j = \hat{\beta}_k, \quad j \in S_k.$$

Moreover, Lemma B.4 yields

$$\left\| \hat{\beta}_k - \tilde{\beta}_k \right\|_2 \leq \frac{\sum_{k' \neq k, k' \in [K]} \Lambda_{kk'}}{N_k \rho/2} < \frac{N_k^{\zeta-1}}{2} \{ M \wedge (\delta/2) \} \quad \forall k \in [K].$$

Now we apply Lemma B.4 and definition of $\mathcal{E}_{1n}$, and conclude that

$$\left\| \tilde{\beta}_k - \beta_k^* \right\|_2 \leq \frac{\| \nabla F_k(\beta_k^*, \hat{\boldsymbol{\eta}}_k) \|_2}{N_k \rho/2} \leq \frac{1}{4} \{ M \wedge (\delta/2) \} \quad \forall k \in [K].$$

$\square$

**Lemma B.7.** *Under the same conditions as in Lemma B.6, the following bound holds*

$$\| \tilde{\beta}_k - \beta_k^* \|_2 \leq C \left\{ K^{1/r_1} + \sqrt{K} \max_{j \in S_k} s(n_j)^{-1} + K m_k^{1/2} \max_{j \in S_k} n_j^{1/2} s(n_j)^{-2} \right\} N_k^{-1/2} t,$$

*where $C > 0$ is a constant.*

*Proof.* The first order condition gives $\nabla F_k(\tilde{\beta}_k, \hat{\boldsymbol{\eta}}_k) = 0$. A first order Taylor expansion around $\beta_k^*$ yields

$$\sqrt{N_k}(\tilde{\beta}_k - \beta_k^*) = - \left\{ N_k^{-1} \nabla^2 F_k(\bar{\beta}_k, \hat{\boldsymbol{\eta}}_k) \right\}^{-1} \left\{ N_k^{-1/2} \nabla F_k(\beta_k^*, \hat{\boldsymbol{\eta}}_k) \right\},$$

for some $\bar{\beta}_k$ on the segment between $\beta_k^*$ and $\tilde{\beta}_k$. Consider the Hessian part. Conditions in Lemma B.6 implies $\| \tilde{\beta}_k - \beta_k^* \|_2 \leq M$, and thus by definition of $\mathcal{E}_{1n}$ we have

$$N_k^{-1} \nabla^2 F_k(\bar{\beta}_k, \hat{\boldsymbol{\eta}}_k) \succeq \frac{\rho}{2} I.$$

Consider the score part. Write

$$N_k^{-1/2} \nabla F_k(\beta_k^*, \hat{\boldsymbol{\eta}}_k) = S_k^0 + R_k,$$
$$S_k^0 := N_k^{-1/2} \nabla F_k(\beta_k^*, \boldsymbol{\eta}_k^*),$$
$$R_k := N_k^{-1/2} \{ \nabla F_k(\beta_k^*, \hat{\boldsymbol{\eta}}_k) - \nabla F_k(\beta_k^*, \boldsymbol{\eta}_k^*) \}.$$

For each summand in $R_k$, a second-order expansion in $\eta_j^*$ gives for $j \in S_k$,

$$\nabla \ell_j(\beta_k^*, \hat{\eta}_j, Z_{ji}) - \nabla \ell_j(\beta_k^*, \eta_j^*, Z_{ji}) = D_\eta \nabla \ell_j(\beta_k^*, \eta_j^*, Z_{ji})[\Delta \eta_j] + \tfrac{1}{2} D_\eta^2 \nabla \ell_j(\beta_k^*, \bar{\eta}_j, Z_{ji})[\Delta \eta_j, \Delta \eta_j],$$

with $\Delta \eta_j := \hat{\eta}_j - \eta_j^*$ and some $\bar{\eta}_j$ on the segment between $\hat{\eta}_j$ and $\eta_j^*$. Hence $R_k$ is decomposed as

$$R_k = R_{k1} + R_{k2}$$
$$R_{k1} = N_k^{-1/2} \sum_{j \in S_k} \sum_{i=1}^{n_j} D_\eta \nabla \ell_j(\beta_k^*, \eta_j^*, Z_{ji})[\Delta \eta_j]$$
$$R_{k2} = N_k^{-1/2} \sum_{j \in S_k} \sum_{i=1}^{n_j} \tfrac{1}{2} D_\eta^2 \nabla \ell_j(\beta_k^*, \bar{\eta}_j, Z_{ji})[\Delta \eta_j, \Delta \eta_j].$$

(B.15)

Now by definition of event $\mathcal{E}_{1n}$ in Definition B.1, it follows that

$$\|S_k^0\|_2 \leq C_1 K^{1/r_1} t$$

$$\|R_{k1}\|_2 \leq C_2 \sqrt{K} \max_{j \in S_k} s(n_j)^{-1} t$$

$$\|R_{k2}\|_2 \leq C_3 K m_k^{1/2} \max_{j \in S_k} n_j^{1/2} s(n_j)^{-2} t$$

Hence, we have for some constant $C > 0$

$$\sqrt{N_k}\|\tilde{\beta}_k - \beta_k^*\|_2 \leq (\rho/2)^{-1} \left\{ C_1 K^{1/r_1} t + C_2 \sqrt{K} \max_{j \in S_k} s(n_j)^{-1} t + C_3 K m_k^{1/2} \max_{j \in S_k} n_j^{1/2} s(n_j)^{-2} t \right\}$$

$$\leq C \left\{ K^{1/r_1} + \sqrt{K} \max_{j \in S_k} s(n_j)^{-1} + K m_k^{1/2} \max_{j \in S_k} n_j^{1/2} s(n_j)^{-2} \right\} \cdot t \,.$$

$\square$

## C. High-probability bounds for events $\mathcal{E}_{1n}$ and $\mathcal{E}_{2n}$

In this section we show that the regularity events $\mathcal{E}_{1n}$ and $\mathcal{E}_{2n}$ in Definitions B.1 and B.2 hold with high probability. The initialization event $\mathcal{E}_{2n}$ follows directly from Assumption 3.3, so we record its probability bound first. We then devote the rest of the section to proving a high-probability bound for $\mathcal{E}_{1n}$, collected in Lemma C.6. The proof of Lemma C.6 is based on auxiliary concentration results for gradients and Hessians, stated in Lemmas C.2 – C.5.

**Lemma C.1** (Probability control of the initialization event $\mathcal{E}_{2n}$). *Under Assumption 3.3, the event $\mathcal{E}_{2n}$ holds with probability at least $1 - p_{\delta/4}(n_{\min})$.*

*Proof.* This is immediate from Assumption 3.3. Setting $\varepsilon = \delta/4$ yields

$$\mathbb{P}\left( \forall j \in [m], \ n_j^\alpha \|\hat{\theta}_j^{\text{init}} - \theta_j^*\|_2 \leq \delta/4 \right) \geq 1 - p_{\delta/4}(n_{\min}) \,, \tag{C.1}$$

and this completes the proof. $\square$

We now turn to the gradient–Hessian regularity event $\mathcal{E}_{1n}$. Recall that $\mathcal{E}_{1n}$ in Definition B.1 requires simultaneous control of empirical gradients and Hessians at $(\theta_j^*, \hat{\eta}_j)$ across all tasks. The following lemmas provide such a bound.

**Lemma C.2.** *Under Assumptions 3.1 and 3.2, there exists a constant $C > 0$ such that, for $t > 0$,*

$$\mathbb{P}\left( \frac{1}{n_j} \|\nabla f_j(\theta_j^*, \hat{\eta}_j) - \nabla f_j(\theta_j^*, \eta_j^*)\|_2 \geq C n_j^{-1/2} t \right) \leq t^{-1}.$$

*Proof.* Write $\Delta\eta_j := \hat{\eta}_j - \eta_j^*$. A second-order expansion in $\eta$ gives

$$\frac{1}{n_j} \|\nabla f_j(\theta_j^*, \hat{\eta}_j) - \nabla f_j(\theta_j^*, \eta_j^*)\|_2 \leq \frac{1}{n_j} \|D_\eta \nabla f_j(\theta_j^*, \eta_j^*)[\Delta\eta_j]\|_2 + \frac{1}{2n_j} \|D_\eta^2 \nabla f_j(\theta_j^*, \bar{\eta}_j)[\Delta\eta_j, \Delta\eta_j]\|_2,$$

for some $\bar{\eta}_j$ on the segment between $\eta_j^*$ and $\hat{\eta}_j$.

*First term.* Let

$$X_i := D_\eta \nabla \ell_j(\theta_j^*, \eta_j^*, Z_{ji})[\Delta\eta_j]$$

for $Z_{ji} \in \mathcal{D}_{j,2}$. By sample splitting, $\Delta\eta_j$ is independent of $\{Z_{ji}\}$. By orthogonality, $\mathbb{E}[X_i \mid \Delta\eta_j] = 0$, and by the second-moment bound in Assumption 3.1(6), $\mathbb{E}\|X_i\|_2^2 \leq \sigma_5 \|\Delta\eta_j\|_{\mathcal{H}_j}^2$. The Marcinkiewicz–Zygmund inequality yields that there exists constant $C_2$

$$\mathbb{E}\left[ \frac{1}{n_j} \|\textstyle\sum_i X_i\|_2 \mid \Delta\eta_j \right] \leq C_2 n_j^{-1/2} \|\Delta\eta_j\|_{\mathcal{H}_j}.$$

Taking expectation and using $\mathbb{E}\|\Delta\eta_j\|_{\mathcal{H}_j} \leq \{\mathbb{E}\|\Delta\eta_j\|_{\mathcal{H}_j}^2\}^{1/2} \leq \sigma_4^{1/2} s(n_j)^{-1}$ from Assumption 3.2, there exists constant $C_3$

$$\mathbb{E}\left[\frac{1}{n_j}\big\|D_\eta\nabla f_j(\theta_j^*, \eta_j^*)[\Delta\eta_j]\big\|_2\right] \leq C_3\, n_j^{-1/2}.$$

*Second term.* By independence of splits and Assumption 3.1(6),

$$\mathbb{E}\left[\frac{1}{2n_j}\big\|D_\eta^2\nabla f_j(\theta_j^*, \bar{\eta}_j)[\Delta\eta_j, \Delta\eta_j]\big\|_2\right] \leq \tfrac{1}{2}\sigma_5\,\mathbb{E}\|\Delta\eta_j\|_{\mathcal{H}_j}^2 \leq \tfrac{1}{2}\sigma_5\,\sigma_4\, s(n_j)^{-2} = o(n_j^{-1/2}),$$

since $s(n_j) \to \infty$ and $n_j^{1/4}/s(n_j) \to 0$.

*Tail bound.* Combining the two displays,

$$\mathbb{E}\left[\frac{1}{n_j}\big\|\nabla f_j(\theta_j^*, \hat{\eta}_j) - \nabla f_j(\theta_j^*, \eta_j^*)\big\|_2\right] \leq C'\, n_j^{-1/2}.$$

Markov's inequality gives

$$\mathbb{P}\left(\frac{1}{n_j}\big\|\nabla f_j(\theta_j^*, \hat{\eta}_j) - \nabla f_j(\theta_j^*, \eta_j^*)\big\|_2 \geq C\, n_j^{-1/2}\, t\right) \leq t^{-1}.$$

$\square$

**Lemma C.3.** *Let $\Delta\eta_j := \hat{\eta}_j - \eta_j^*$, and let $\bar{\eta}_j$ be a point on the line segment between $\eta_j^*$ and $\hat{\eta}_j$. Under Assumptions 3.1 and 3.2, there exist constants $C, \tilde{C} > 0$ such that for any $t > 0$,*

$$\mathbb{P}\left(N_k^{-1/2}\left\|\sum_{j\in S_k} D_\eta\nabla f_j(\theta_j^*, \eta_j^*)[\Delta\eta_j]\right\|_2 > C\max_{j\in S_k} s(n_j)^{-1}t\right) \leq t^{-2},$$

$$\mathbb{P}\left(N_k^{-1/2}\left\|\sum_{j\in S_k} D_\eta^2\nabla f_j(\theta_j^*, \bar{\eta}_j)[\Delta\eta_j, \Delta\eta_j]\right\|_2 > \tilde{C}\, m_k^{1/2}\max_{j\in S_k} n_j^{1/2} s(n_j)^{-2}t\right) \leq t^{-1},$$

*where $m_k := |S_k|$.*

*Proof.* We prove the two bounds separately.

*First term.* For $j \in S_k$ and $i \in [n_j]$, define

$$X_{ji} := D_\eta\nabla\ell_j(\theta_j^*, \eta_j^*, Z_{ji})[\Delta\eta_j].$$

By sample splitting, $\Delta\eta_j$ is independent of $\{Z_{ji} : i \in [n_j]\}$. Hence, conditional on $\Delta\eta_j$, the variables $\{X_{ji}\}_{i=1}^{n_j}$ are independent. By orthogonality,

$$\mathbb{E}[X_{ji} \mid \Delta\eta_j] = 0.$$

Moreover, by the second-moment bound in Assumption 3.1,

$$\mathbb{E}\big[\|X_{ji}\|_2^2 \mid \Delta\eta_j\big] \leq \sigma_5\|\Delta\eta_j\|_{\mathcal{H}_j}^2.$$

Therefore,

$$\mathbb{E}\left[\left\|N_k^{-1/2}\sum_{j\in S_k}\sum_{i=1}^{n_j} X_{ji}\right\|_2^2 \,\Bigg|\, \{\Delta\eta_j\}_{j\in S_k}\right] = N_k^{-1}\sum_{j\in S_k}\sum_{i=1}^{n_j}\mathbb{E}\big[\|X_{ji}\|_2^2 \mid \Delta\eta_j\big]$$

$$\leq \sigma_5 N_k^{-1}\sum_{j\in S_k} n_j\|\Delta\eta_j\|_{\mathcal{H}_j}^2.$$

Taking expectation and using Assumption 3.2,

$$\mathbb{E}\|\Delta\eta_j\|_{\mathcal{H}_j}^2 \leq \sigma_4 s(n_j)^{-2},$$

we obtain

$$\mathbb{E}\left[\left\|N_k^{-1/2}\sum_{j\in S_k}\sum_{i=1}^{n_j}X_{ji}\right\|_2^2\right] \leq \sigma_4\sigma_5\,N_k^{-1}\sum_{j\in S_k}n_j s(n_j)^{-2}$$

$$\leq C\max_{j\in S_k}s(n_j)^{-2}$$

for some constant $C > 0$. By Markov's inequality,

$$\mathbb{P}\left(\left\|N_k^{-1/2}\sum_{j\in S_k}\sum_{i=1}^{n_j}X_{ji}\right\|_2 > C^{1/2}\max_{j\in S_k}s(n_j)^{-1}t\right) \leq t^{-2}.$$

*Second term.* By Assumption 3.1,

$$\frac{1}{n_j}D_\eta^2\nabla f_j(\theta_j^*,\bar{\eta}_j)[\Delta\eta_j,\Delta\eta_j] = \frac{1}{n_j}\sum_{i=1}^{n_j}D_\eta^2\nabla\ell_j(\theta_j^*,\bar{\eta}_j,Z_{ji})[\Delta\eta_j,\Delta\eta_j].$$

Using the moment bound in Assumption 3.1 together with Assumption 3.2, we get

$$\mathbb{E}\left[\frac{1}{n_j}\left\|D_\eta^2\nabla f_j(\theta_j^*,\bar{\eta}_j)[\Delta\eta_j,\Delta\eta_j]\right\|_2\right] \leq \sigma_5\mathbb{E}\|\Delta\eta_j\|_{\mathcal{H}_j}^2 \leq \sigma_4\sigma_5\,s(n_j)^{-2}.$$

Hence,

$$\mathbb{E}\left[\left\|D_\eta^2\nabla f_j(\theta_j^*,\bar{\eta}_j)[\Delta\eta_j,\Delta\eta_j]\right\|_2\right] \leq C'\,n_j s(n_j)^{-2}$$

for some constant $C' > 0$. Summing over $j \in S_k$ and using the triangle inequality,

$$\mathbb{E}\left[N_k^{-1/2}\left\|\sum_{j\in S_k}D_\eta^2\nabla f_j(\theta_j^*,\bar{\eta}_j)[\Delta\eta_j,\Delta\eta_j]\right\|_2\right] \leq C'N_k^{-1/2}\sum_{j\in S_k}n_j s(n_j)^{-2}$$

$$= C'N_k^{-1/2}\sum_{j\in S_k}n_j^{1/2}\left(n_j^{1/2}s(n_j)^{-2}\right)$$

$$\leq C'\left(N_k^{-1/2}\sum_{j\in S_k}n_j^{1/2}\right)\max_{j\in S_k}n_j^{1/2}s(n_j)^{-2}.$$

By Cauchy–Schwarz,

$$\sum_{j\in S_k}n_j^{1/2} \leq m_k^{1/2}\left(\sum_{j\in S_k}n_j\right)^{1/2} = m_k^{1/2}N_k^{1/2},$$

so

$$\mathbb{E}\left[N_k^{-1/2}\left\|\sum_{j\in S_k}D_\eta^2\nabla f_j(\theta_j^*,\bar{\eta}_j)[\Delta\eta_j,\Delta\eta_j]\right\|_2\right] \leq C'\,m_k^{1/2}\max_{j\in S_k}n_j^{1/2}s(n_j)^{-2}.$$

Another application of Markov's inequality yields

$$\mathbb{P}\left(N_k^{-1/2}\left\|\sum_{j\in S_k}D_\eta^2\nabla f_j(\theta_j^*,\bar{\eta}_j)[\Delta\eta_j,\Delta\eta_j]\right\|_2 > C'\,m_k^{1/2}\max_{j\in S_k}n_j^{1/2}s(n_j)^{-2}t\right) \leq t^{-1}.$$

This proves the second bound. $\qquad\square$

**Lemma C.4.** *Under Assumption 3.1, if $t > c_0 n_j^{-r_2/(r_2+d)}$ for a constant $c_0 > 0$, then it holds for any $j \in [m]$ that*

$$\mathbb{P}\left(\sup_{\theta \in \mathcal{B}_{\theta_j^*, M}} \left\| \frac{1}{n_j} \left[ \nabla^2 f_j(\theta, \eta_j^*) - \mathbb{E}\nabla^2 f_j(\theta, \eta_j^*) \right] \right\|_2 \geq C n_j^{-\frac{r_2}{2(r_2+d)}} t \right) \leq t^{-\frac{d+r_2}{d+1}}$$

*Proof.* Since we only consider a fixed $j$, in this proof, all the dependencies on $j$ are suppressed for better readability and notational convenience. For example, $\theta_j^*, f_j, \ell_j, n_j, Z_{ji}, \mathcal{B}_{\theta_j^*, M}$ are denoted by $\theta^*, f, \ell, n, Z_i, \mathcal{B}_\theta$. Moreover, since the coordinate $\eta_j^*$ is fixed, it is also suppressed in the proof, and we use shorthand notations $f(\theta, \eta_j^*) := f(\theta)$ and $\ell(\theta, \eta_j^*, Z) := \ell(\theta, Z)$ here.

Let $\mathcal{N}_\varepsilon$ be the $\varepsilon$-covering number of $\mathcal{B}_\theta$ and $\Theta_\varepsilon = \{\theta_j\}_{j \in [\mathcal{N}_\varepsilon]}$ be the corresponding $\varepsilon$-net. Define the map $q : \mathcal{B}_\theta \to \Theta_\varepsilon$ as $q(\theta) = \arg\min_{j \in [\mathcal{N}_\varepsilon]} \|\theta - \theta_j\|_2$, then $\|\theta - \theta_{q(\theta)}\|_2 \leq \varepsilon, \forall \theta \in \mathcal{B}_\theta$. Recall $\mathcal{B}_\theta$ is a $d$-dimensional Euclidean ball of radius $M$, then it follows that $\mathcal{N}_\varepsilon \leq (3M/\varepsilon)^d$, see (Vershynin, 2010).

For any $\theta \in \mathcal{B}_\theta$, the quantity of interest is decomposed into:

$$\left\| \frac{1}{n} \left[ \nabla^2 f(\theta) - \mathbb{E}\nabla^2 f(\theta) \right] \right\|_2 \leq \underbrace{\left\| \frac{1}{n} \sum_{i=1}^n \left[ \nabla^2 \ell(\theta, Z_i) - \nabla^2 \ell(\theta_{q(\theta)}, Z_i) \right] \right\|_2}_{:=T_1(\theta)}$$

$$+ \underbrace{\left\| \frac{1}{n} \sum_{i=1}^n \nabla^2 \ell(\theta_{q(\theta)}, Z_i) - \mathbb{E}\left[ \nabla^2 \ell(\theta_{q(\theta)}, Z) \right] \right\|_2}_{:=T_2} + \underbrace{\left\| \mathbb{E}\left[ \nabla^2 \ell(\theta_{q(\theta)}, Z) \right] - \mathbb{E}\left[ \nabla^2 \ell(\theta, Z) \right] \right\|_2}_{:=T_3(\theta)}.$$

Note that the second term $T_2$ is independent of $\theta$. Our target probability has upper bound

$$\mathbb{P}\left( \sup_{\theta \in \mathcal{B}_\theta} \left\| \frac{1}{n} \left[ \nabla^2 f(\theta) - \mathbb{E}\nabla^2 f(\theta) \right] \right\|_2 \geq t \right)$$

$$\leq \mathbb{P}\left( \sup_{\theta \in \mathcal{B}_\theta} T_1(\theta) \geq t/3 \right) + \mathbb{P}\left( \sup_{\theta \in \mathcal{B}_\theta} T_2(\theta) \geq t/3 \right) + \mathbb{P}\left( \sup_{\theta \in \mathcal{B}_\theta} T_3(\theta) \geq t/3 \right). \tag{C.2}$$

Now we start to bound each term above.

**Bound on $T_1$.** By Markov inequality,

$$\mathbb{P}\left( \sup_{\theta \in \mathcal{B}_\theta} T_1(\theta) \geq t/3 \right) \leq \frac{3}{t} \mathbb{E}\left[ \sup_{\theta \in \mathcal{B}_\theta} \left\| \frac{1}{n} \sum_{i=1}^n \left[ \nabla^2 \ell(\theta, Z_i) - \nabla^2 \ell(\theta_{q(\theta)}, Z_i) \right] \right\|_2 \right].$$

By Jensen's inequality and subadditivity of the supremum, the term inside the expectation is such that

$$\sup_{\theta \in \mathcal{B}_\theta} \left\| \frac{1}{n} \sum_{i=1}^n \left[ \nabla^2 \ell(\theta, Z_i) - \nabla^2 \ell(\theta_{q(\theta)}, Z_i) \right] \right\|_2 \leq \sup_{\theta \in \mathcal{B}_\theta} \frac{1}{n} \sum_{i=1}^n \left\| \nabla^2 \ell(\theta, Z_i) - \nabla^2 \ell(\theta_{q(\theta)}, Z_i) \right\|_2$$

$$\leq \frac{1}{n} \sum_{i=1}^n \sup_{\theta \in \mathcal{B}_\theta} \left\| \nabla^2 \ell(\theta, Z_i) - \nabla^2 \ell(\theta_{q(\theta)}, Z_i) \right\|_2.$$

Since $Z_i$ are i.i.d., it follows that

$$
\begin{aligned}
\mathbb{P}\left(\sup_{\theta \in \mathcal{B}_\theta} T_1(\theta) \geq t/3\right) &\leq \frac{3}{t} \mathbb{E}\left[\sup_{\theta \in \mathcal{B}_\theta} \left\|\nabla^2 \ell(\theta, Z) - \nabla^2 \ell\left(\theta_{q(\theta)}, Z\right)\right\|_2\right] \\
&\leq \frac{3}{t} \mathbb{E}\left[\sup_{\theta \in \mathcal{B}_\theta} \frac{\left\|\nabla^2 \ell(\theta, Z) - \nabla^2 \ell\left(\theta_{q(\theta)}, Z\right)\right\|_2}{\left\|\theta - \theta_{q(\theta)}\right\|_2}\right] \cdot \sup_{\theta \in \mathcal{B}_\theta}\left\|\theta - \theta_{q(\theta)}\right\|_2 \\
&\leq \frac{3}{t} \mathbb{E}\left[\sup_{\theta_1 \neq \theta_2 \in \mathcal{B}_\theta} \frac{\left\|\nabla^2 \ell\left(\theta_1, Z\right) - \nabla^2 \ell\left(\theta_2, Z\right)\right\|_2}{\left\|\theta_1 - \theta_2\right\|_2}\right] \cdot \varepsilon \\
&\leq \frac{3\sigma_3 \varepsilon}{t}. \qquad \text{[Assumption 3.1(4)]}
\end{aligned}
$$

**Bound on $T_2$.** Let $j$ be an arbitrary element in $[\mathcal{N}_\varepsilon]$. Define $\tilde{\Theta}_{1/4}$ be a $(1/4)$-net of $d$-dimensional unit ball $\{x : \|x\|_2 \leq 1\}$ and its covering number $\tilde{\mathcal{N}}_{1/4} \leq 12^d$. By Lemma 5.4 of (Vershynin, 2010), we have

$$
\left\|\frac{1}{n}\sum_{i=1}^n \nabla^2 \ell\left(\theta_j, Z_i\right) - \mathbb{E}\left[\nabla^2 \ell\left(\theta_j, Z\right)\right]\right\|_2 \leq 2 \sup_{v \in \tilde{\Theta}_{1/4}}\left|\left\langle v, \left(\frac{1}{n}\sum_{i=1}^n \nabla^2 \ell\left(\theta_j, Z_i\right) - \mathbb{E}\left[\nabla^2 \ell\left(\theta_j, Z\right)\right]\right) v\right\rangle\right|,
$$

which implies

$$
\mathbb{P}\left(\sup_{\theta \in \mathcal{B}_\theta} T_2(\theta) \geq \frac{t}{3}\right) \leq \mathbb{P}\left(2 \sup_{j \in [\mathcal{N}_\varepsilon]} \sup_{v \in \tilde{\Theta}_{1/4}}\left|\left\langle v, \left(\frac{1}{n}\sum_{i=1}^n \nabla^2 \ell\left(\theta_j, Z_i\right) - \mathbb{E}\left[\nabla^2 \ell\left(\theta_j, Z\right)\right]\right) v\right\rangle\right| \geq \frac{t}{3}\right).
$$

Now we apply union bounds over $\Theta_\varepsilon$ and $\tilde{\Theta}_{1/4}$, whose covering numbers are $(3M/\varepsilon)^d$ and $12^d$, to get

$$
\mathbb{P}\left(\sup_{\theta \in \mathcal{B}_\theta} T_2(\theta) \geq t/3\right) \leq (36M/\varepsilon)^d \cdot \mathbb{P}\left(\left|\frac{1}{n}\sum_{i=1}^n \left\langle v, \left(\nabla^2 \ell\left(\theta_j, Z_i\right) - \mathbb{E}\left[\nabla^2 \ell\left(\theta_j, Z\right)\right]\right) v\right\rangle\right| \geq \frac{t}{6}\right).
$$

Since $\left\langle v, \left(\nabla^2 \ell\left(\theta_j, Z_i\right) - \mathbb{E}\left[\nabla \ell^2\left(\theta_j, Z\right)\right]\right) v\right\rangle$ has bounded $r_2$ moments by Assumption 3.1(3), applying Markov inequality yields

$$
\mathbb{P}\left(\left|\frac{1}{n}\sum_{i=1}^n \left\langle v, \left(\nabla^2 \ell\left(\theta_j, Z_i\right) - \mathbb{E}\left[\nabla^2 \ell\left(\theta_j, Z\right)\right]\right) v\right\rangle\right| \geq \frac{t}{6}\right) \leq \frac{C_1}{n^{r_2/2} t^{r_2}}
$$

for some constant $C_1 > 0$. Therefore, we have an upper bound

$$
\mathbb{P}\left(\sup_{\theta \in \mathcal{B}_\theta} T_2(\theta) \geq \frac{t}{3}\right) \leq \frac{C_1}{n^{r_2/2} t^{r_2}}\left(\frac{36M}{\varepsilon}\right)^d.
$$

**Bound on $T_3$.** By Assumption 3.1(4), it holds that

$$
\begin{aligned}
\sup_{\theta \in \mathcal{B}_\theta} T_3(\theta) &\leq \sup_{\theta \in \mathcal{B}_\theta} \frac{\left\|\mathbb{E}\left[\nabla^2 \ell(\theta, Z) - \nabla^2 \ell\left(\theta_{q(\theta)}, Z\right)\right]\right\|_2}{\left\|\theta - \theta_{q(\theta)}\right\|_2} \sup_{\theta \in \mathcal{B}_\theta}\left\|\theta - \theta_{q(\theta)}\right\|_2 \\
&\leq \mathbb{E}\left[\sup_{\theta_1 \neq \theta_2 \in \mathcal{B}_\theta} \frac{\left\|\nabla^2 \ell\left(\theta_1, Z\right) - \nabla^2 \ell\left(\theta_2, Z\right)\right\|_2}{\left\|\theta_1 - \theta_2\right\|_2}\right] \cdot \varepsilon \\
&\leq \sigma_3 \cdot \varepsilon.
\end{aligned}
$$

Therefore, we have

$$
\mathbb{P}\left(\sup_{\theta \in \mathcal{B}_\theta} T_3(\theta) \geq t/3\right) \leq \mathbb{1}(t/3 \leq \sigma_3 \cdot \varepsilon),
$$

which means 0 probability for sufficiently large $t$.

**Combining 3 bounds.** Collecting the bounds on $T_1(\theta), T_2(\theta)$, and $T_3(\theta)$ above, and going back to the decomposition in Equation (C.2), it follows that

$$\mathbb{P}\left(\sup_{\theta \in \mathcal{B}_\theta} \left\| \frac{1}{n} \left[ \nabla^2 f(\theta) - \mathbb{E}\nabla^2 f(\theta) \right] \right\|_2 \geq t\right) \leq \frac{3\sigma_3 \varepsilon}{t} + \frac{C_1}{n^{r_2/2} t^{r_2}} \left( \frac{36M}{\varepsilon} \right)^d + \mathbb{1}(t \leq 3\sigma_3 \cdot \varepsilon).$$

Here we choose $\varepsilon$ to be

$$\varepsilon^* = \left( \frac{C_1 d(36M)^d}{3\sigma_3} \cdot \frac{1}{n^{r_2/2} t^{r_2-1}} \right)^{1/(d+1)}.$$

Because the given condition $t > c_0 n^{-r_2/2(r_2+d)}$, as long as $c_0, C_1$ are chosen suitably, the following inequality holds:

$$t > 3\sigma_3 \varepsilon^* = 3\sigma_3 \left( \frac{C_1 d(36M)^d}{3\sigma_3} \cdot \frac{1}{n^{r_2/2} t^{r_2-1}} \right)^{1/(d+1)}.$$

This inequality implies $\mathbb{1}(t \leq 3\sigma_3 \cdot \varepsilon^*) = 0$ when $\varepsilon = \varepsilon^*$. Hence, with this $\varepsilon^*$, we have upper bounds for the terms

$$\frac{3\sigma_3 \varepsilon}{t} \leq C_1^{\frac{1}{d+1}} d^{\frac{1}{d+1}} (108M\sigma_3)^{\frac{d}{d+1}} n^{-r_2/(2(d+1))} t^{-(d+r_2)/(d+1)};$$

$$\frac{C_1}{n^{r_2/2} t^{r_2}} \left( \frac{36M}{\varepsilon} \right)^d \leq C_1^{\frac{1}{d+1}} d^{-\frac{d}{d+1}} (108M\sigma_3)^{\frac{d}{d+1}} n^{-r_2/(2(d+1))} t^{-(d+r_2)/(d+1)}.$$

The facts that $2 > d^{1/(d+1)} \geq d^{-d/(d+1)}$ yields the final upper bound

$$\mathbb{P}\left(\sup_{\theta \in \mathcal{B}_\theta} \left\| \frac{1}{n} \left[ \nabla^2 f(\theta) - \mathbb{E}\nabla^2 f(\theta) \right] \right\|_2 \geq t\right) \leq Cn^{-r_2/(2(d+1))} t^{-(d+r_2)/(d+1)}$$

for some large enough $C$. Recall that we have shown if $t > c_0 n^{-r_2/2(r_2+d)}$ then the above concentration holds. Substituting $t$ with $C^{-\frac{d+1}{d+r_2}} n^{\frac{r_2}{2(d+r_2)}} t$ proves the lemma. $\qquad \square$

**Lemma C.5.** *Let Assumptions 3.1 and 3.2 hold. There exist constants $C_1, C_2 > 0$ such that*

$$\mathbb{P}\left(\sup_{\theta \in \mathcal{B}_{\theta_j^*, M}} \frac{1}{n_j} \left\| \nabla^2 f_j(\theta, \hat{\eta}_j) - \nabla^2 f_j(\theta, \eta_j^*) \right\|_2 \geq C_1 s^{-1}(n_j) t \text{ or } \left\| \hat{\eta}_j - \eta_j^* \right\|_{\mathcal{H}_j} > M\right) \leq t^{-1} + C_2 s^{-1}(n_j).$$

*Proof.* Define the event

$$\mathcal{E} = \left\{ \left\| \hat{\eta}_j - \eta_j^* \right\|_{\mathcal{H}_j} \leq M \right\},$$

then it has the following bound

$$\begin{aligned} \mathbb{P}(\mathcal{E}^c) &\leq \frac{1}{M} \mathbb{E}_{\mathcal{D}_{j,1}} \left\{ \left\| \hat{\eta}_j - \eta_j^* \right\|_{\mathcal{H}_j} \right\} && \text{[Markov's inequality]} \\ &\leq \sigma_4^{1/2} M^{-1} s^{-1}(n_j). && \text{[Assumption 3.2]} \end{aligned} \quad (C.3)$$

Consider the decomposition:

$$\begin{aligned} &\mathbb{P}\left(\sup_{\theta \in \mathcal{B}_{\theta_j^*, M}} \frac{1}{n_j} \left\| \nabla^2 f_j(\theta, \hat{\eta}_j) - \nabla^2 f_j(\theta, \eta_j^*) \right\|_2 \geq t\right) \\ &= \mathbb{P}\left(\sup_{\theta \in \mathcal{B}_{\theta_j^*, M}} \frac{1}{n_j} \left\| \nabla^2 f_j(\theta, \hat{\eta}_j) - \nabla^2 f_j(\theta, \eta_j^*) \right\|_2 \geq t \middle| \mathcal{E}\right) \mathbb{P}(\mathcal{E}) \\ &\quad + \mathbb{P}\left(\sup_{\theta \in \mathcal{B}_{\theta_j^*, M}} \frac{1}{n_j} \left\| \nabla^2 f_j(\theta, \hat{\eta}_j) - \nabla^2 f_j(\theta, \eta_j^*) \right\|_2 \geq t \middle| \mathcal{E}^c\right) \mathbb{P}(\mathcal{E}^c). \end{aligned} \quad (C.4)$$

By Jensen's inequality and subadditivity of supremum,

$$
\mathbb{E}_{\mathcal{D}_{j,2}} \left[ \sup_{\theta \in \mathcal{B}_{\theta_j^*, M}} \frac{1}{n_j} \left\| \nabla^2 f_j(\theta, \hat{\eta}_j) - \nabla^2 f_j(\theta, \eta_j^*) \right\|_2 \,\middle|\, \mathcal{E} \right]
$$

$$
= \mathbb{E}_{\mathcal{D}_{j,2}} \left[ \sup_{\theta \in \mathcal{B}_{\theta_j^*, M}} \left\| \frac{1}{n_j} \sum_{Z \in \mathcal{D}_{j,2}} \left[ \nabla^2 \ell_j(\theta, \hat{\eta}_j, Z) - \nabla^2 \ell_j(\theta, \eta_j^*, Z) \right] \right\|_2 \,\middle|\, \mathcal{E} \right]
$$

$$
\leq \mathbb{E}_{\mathcal{D}_{j,2}} \left[ \frac{1}{n_j} \sum_{Z \in \mathcal{D}_{j,2}} \sup_{\theta \in \mathcal{B}_{\theta_j^*, M}} \left\| \left[ \nabla^2 \ell_j(\theta, \hat{\eta}_j, Z) - \nabla^2 \ell_j(\theta, \eta_j^*, Z) \right] \right\|_2 \,\middle|\, \mathcal{E} \right]
$$

$$
= \mathbb{E}_{\mathcal{D}_{j,2}} \left[ \sup_{\theta \in \mathcal{B}_{\theta_j^*, M}} \left\| \left[ \nabla^2 \ell_j(\theta, \hat{\eta}_j, Z) - \nabla^2 \ell_j(\theta, \eta_j^*, Z) \right] \right\|_2 \,\middle|\, \mathcal{E} \right] .
$$

This term has upper bound

$$
\mathbb{E}_{\mathcal{D}_{j,2}} \left[ \sup_{\theta \in \mathcal{B}_{\theta_j^*, M}} \left\| \nabla^2 \ell_j(\theta, \hat{\eta}_j, Z) - \nabla^2 \ell_j(\theta, \eta_j^*, Z) \right\|_2 \,\middle|\, \mathcal{E} \right]
$$

$$
= \mathbb{E}_{\mathcal{D}_{j,2}} \left[ \sup_{\theta \in \mathcal{B}_{\theta_j^*, M}} \frac{\left\| \nabla^2 \ell_j(\theta, \hat{\eta}_j, Z) - \nabla^2 \ell_j(\theta, \eta_j^*, Z) \right\|_2}{\left\| \hat{\eta}_j - \eta_j^* \right\|_{\mathcal{H}_j}} \left\| \hat{\eta}_j - \eta_j^* \right\|_{\mathcal{H}_j} \,\middle|\, \mathcal{E} \right] \tag{C.5}
$$

$$
\leq \mathbb{E}_{\mathcal{D}_{j,2}} \left[ \sup_{(\theta, \eta_1) \neq (\theta, \eta_2) \in \mathcal{B}_{\theta_j^*, M} \times \mathcal{B}_{\eta_j^*, M}} \frac{\left\| \nabla^2 \ell_j(\theta, \eta_1, Z) - \nabla^2 \ell_j(\theta, \eta_2, Z) \right\|_2}{\left\| \eta_1 - \eta_2 \right\|_{\mathcal{H}_j}} \right] \left\| \hat{\eta}_j - \eta_j^* \right\|_{\mathcal{H}_j}
$$

$$
\leq \sigma_3 \left\| \hat{\eta}_j - \eta_j^* \right\|_{\mathcal{H}_j} . \quad \text{[Assumption 3.1(4)]}
$$

Here, the penultimate inequality follows from the independence between the splits, i.e., $\hat{\eta}_j$ is independent of the data used to construct our adaptive estimators. Now we use this result to bound in first term in (C.4),

$$
\mathbb{P} \left( \sup_{\theta \in \mathcal{B}_{\theta_j^*, M}} \frac{1}{n_j} \left\| \nabla^2 f_j(\theta, \hat{\eta}_j) - \nabla^2 f_j(\theta, \eta_j^*) \right\|_2 \geq t \,\middle|\, \mathcal{E} \right) \mathbb{P}(\mathcal{E})
$$

$$
\leq \frac{1}{t} \mathbb{E} \left[ \sup_{\theta \in \mathcal{B}_{\theta_j^*, M}} \frac{1}{n_j} \left\| \nabla^2 f_j(\theta, \hat{\eta}_j) - \nabla^2 f_j(\theta, \eta_j^*) \right\|_2 \,\middle|\, \mathcal{E} \right] \mathbb{P}(\mathcal{E})
$$

$$
= \frac{1}{t} \mathbb{E}_{\mathcal{D}_{j,1}} \left\{ \mathbb{E}_{\mathcal{D}_{j,2}} \left[ \sup_{\theta \in \mathcal{B}_{\theta_j^*, M}} \frac{1}{n_j} \left\| \nabla^2 f_j(\theta, \hat{\eta}_j) - \nabla^2 f_j(\theta, \eta_j^*) \right\|_2 \,\middle|\, \mathcal{E} \right] \,\middle|\, \mathcal{E} \right\} \mathbb{P}(\mathcal{E})
$$

$$
\leq \frac{\sigma_3}{t} \mathbb{E}_{\mathcal{D}_{j,1}} \left\{ \left\| \hat{\eta}_j - \eta_j^* \right\|_{\mathcal{H}_j} \,\middle|\, \mathcal{E} \right\} \mathbb{P}(\mathcal{E}) \quad \text{[upper bound in (C.5)]}
$$

$$
\leq \frac{\sigma_3}{t} \mathbb{E}_{\mathcal{D}_{j,1}} \left\{ \left\| \hat{\eta}_j - \eta_j^* \right\|_{\mathcal{H}_j} \right\}
$$

$$
\leq \frac{\sigma_3 \sigma_4^{1/2}}{t} s^{-1}(n_j) . \quad \text{[Assumption 3.2]}
$$

Now we bound the second term in (C.4) using Equation (C.3),

$$
\mathbb{P} \left( \sup_{\theta \in \mathcal{B}_{\theta_j^*, M}} \frac{1}{n_j} \left\| \nabla^2 f_j(\theta, \hat{\eta}_j) - \nabla^2 f_j(\theta, \eta_j^*) \right\|_2 \geq t \,\middle|\, \mathcal{E}^c \right) \mathbb{P}(\mathcal{E}^c) \leq \mathbb{P}(\mathcal{E}^c)
$$

$$
\leq \sigma_4^{1/2} M^{-1} s^{-1}(n_j) .
$$

Collecting the two upper bounds, we go back to Inequality (C.4), and it follows that

$$\mathbb{P}\left(\sup_{\theta\in\mathcal{B}_{\theta_j^*,M}}\frac{1}{n_j}\left\|\nabla^2 f_j(\theta,\hat\eta_j)-\nabla^2 f_j(\theta,\eta_j^*)\right\|_2\geq t\right)\leq\frac{\sigma_3\sigma_4^{1/2}}{t}s^{-1}(n_j)+\sigma_4^{1/2}M^{-1}s^{-1}(n_j).\qquad\text{(C.6)}$$

Applying the union bound to combine (C.6) and (C.3) we have

$$\mathbb{P}\left(\sup_{\theta\in\mathcal{B}_{\theta_j^*,M}}\frac{1}{n_j}\left\|\nabla^2 f_j(\theta,\hat\eta_j)-\nabla^2 f_j(\theta,\eta_j^*)\right\|_2\geq t\ \textbf{ or }\ \mathcal{E}^c\right)\leq\frac{\sigma_3\sigma_4^{1/2}}{t}s^{-1}(n_j)+2\sigma_4^{1/2}M^{-1}s^{-1}(n_j).$$

Rescaling $t$ yields the result stated in the lemma. $\qquad\square$

**Lemma C.6.** *Under Assumptions 3.1 - 3.2, if*

$$1<t<c_2 m^{-1}\left\{n_{\min}^{\frac{r_2}{2(r_2+d)}}\wedge s(n_{\min})\right\}$$

*then it holds with probability at least $1-t^{-1}$ that*

$$\frac{1}{n_j}\left\|\nabla f_j(\theta_j^*,\hat\eta_j)\right\|_2\leq\frac{\rho}{8}\{M\wedge(\delta/2)\}\qquad\qquad\forall j\in[m];$$

$$\frac{\rho}{2}I\preceq\frac{1}{n_j}\nabla^2 f_j(\theta,\hat\eta_j)\preceq\frac{3}{2}\kappa\qquad\qquad\forall\theta\in\mathcal{B}_{\theta_j^*,M},\ \forall j\in[m];$$

$$\left\|\tfrac{1}{N_k}\nabla F_k(\beta_k^*,\hat{\boldsymbol\eta}_k)\right\|_2\leq\frac{\rho}{8}\{M\wedge(\delta/2)\}\qquad\qquad\forall k\in[K];$$

$$\left\|\hat\eta_j-\eta_j^*\right\|_{\mathcal{H}_j}\leq M\qquad\qquad j\in[m];$$

$$N_k^{-1/2}\left\|\sum_{j\in S_k}D_\eta\nabla f_j(\theta_j^*,\eta_j^*)[\Delta\eta_j]\right\|_2\leq C\sqrt{K}\max_{j\in S_k}s(n_j)^{-1}t\qquad\qquad k\in[K];$$

$$N_k^{-1/2}\left\|\sum_{j\in S_k}D_\eta^2\nabla f_j(\theta_j^*,\bar\eta_j)[\Delta\eta_j,\Delta\eta_j]\right\|_2\leq CK\,m_k^{1/2}\max_{j\in S_k}n_j^{1/2}s(n_j)^{-2}t\qquad\qquad k\in[K];$$

$$N_k^{-1/2}\left\|\sum_{j\in S_k}\nabla f_j(\theta_j^*,\eta_j^*)\right\|_2\leq CK^{1/r_1}t\qquad\qquad k\in[K],$$

*where $\Delta\eta_j=\hat\eta_j-\eta_j^*$ and $c_1,c_2,C>0$ are some constants.*

*Proof.* We begin by collecting the previous tail bounds for gradients and Hessians. Under the stated assumptions, there exist

constants $C_0, C_1$ such that, with $m_k := |S_k|$, for any $t \geq 1$,

$$\mathbb{P}\left(\left\|\frac{1}{n_j}\nabla f_j(\theta_j^*, \eta_j^*)\right\|_2 \geq C_1 n_j^{-1/2}t\right) \leq t^{-r_1} \qquad\qquad j \in [m]; \quad \text{(C.7)}$$

$$\mathbb{P}\left(\frac{1}{n_j}\left\|\nabla f_j(\theta_j^*, \hat{\eta}_j) - \nabla f_j(\theta_j^*, \eta_j^*)\right\|_2 \geq C_1 m n_j^{-1/2}t\right) \leq t^{-1} \qquad\qquad j \in [m]; \quad \text{(C.8)}$$

$$\mathbb{P}\left(N_k^{-1/2}\left\|\sum_{j \in S_k} D_\eta \nabla f_j(\theta_j^*, \eta_j^*)[\Delta\eta_j]\right\|_2 > C_1 \max_{j \in S_k} s(n_j)^{-1}t\right) \leq t^{-2} \qquad\qquad k \in [K]; \quad \text{(C.9)}$$

$$\mathbb{P}\left(N_k^{-1/2}\left\|\sum_{j \in S_k} D_\eta^2 \nabla f_j(\theta_j^*, \bar{\eta}_j)[\Delta\eta_j, \Delta\eta_j]\right\|_2 > C_1 m_k^{1/2} \max_{j \in S_k} n_j^{1/2} s(n_j)^{-2}t\right) \leq t^{-1} \qquad k \in [K]; \quad \text{(C.10)}$$

$$\mathbb{P}\left(N_k^{-1/2}\left\|\sum_{j \in S_k} \nabla f_j(\theta_j^*, \eta_j^*)\right\|_2 \geq C_1 t\right) \leq t^{-r_1} \qquad\qquad k \in [K] \quad \text{(C.11)}$$

$$\mathbb{P}\left(\sup_{\theta \in \mathcal{B}_{\theta_j^*, M}} \frac{1}{n_j}\left\|\nabla^2 f_j(\theta, \eta_j^*) - \mathbb{E}\nabla^2 f_j(\theta, \eta_j^*)\right\|_2 \geq C_1 n_j^{-\frac{r_2}{2(r_2+d)}}t\right) \leq t^{-\frac{d+r_2}{d+1}} \qquad j \in [m]; \quad \text{(C.12)}$$

$$\mathbb{P}\left(\sup_{\theta \in \mathcal{B}_{\theta_j^*, M}} \frac{1}{n_j}\left\|\nabla^2 f_j(\theta, \hat{\eta}_j) - \nabla^2 f_j(\theta, \eta_j^*)\right\|_2 \geq C_1 s^{-1}(n_j)t \text{ or } \left\|\hat{\eta}_j - \eta_j^*\right\|_{\mathcal{H}_j} > M\right) \leq t^{-1} + C_0 s^{-1}(n_j) \quad j \in [m], \quad \text{(C.13)}$$

where Equations (C.7) and (C.11) are due to Assumption 3.1(2) on the first-order condition, combined with the standard Markov's inequality argument; Equation (C.8) is shown in Lemma C.2; Equations (C.9) and (C.10) have been shown in Lemma C.3; Equation (C.12) is proved in Lemma C.4; Equation (C.13) is verified by Lemma C.5.

Rescaling all $t$, the inequalities above are equivalent to

$$\mathbb{P}\left(\left\|\frac{1}{n_j}\nabla f_j(\theta_j^*, \eta_j^*)\right\|_2 \geq C_1' m^{1/r_1} n_j^{-1/2}t\right) \leq \frac{t^{-r_1}}{7m} \qquad\qquad j \in [m];$$

$$\text{(C.14)}$$

$$\mathbb{P}\left(\frac{1}{n_j}\left\|\nabla f_j(\theta_j^*, \hat{\eta}_j) - \nabla f_j(\theta_j^*, \eta_j^*)\right\|_2 \geq C_1' m n_j^{-1/2}t\right) \leq \frac{t^{-1}}{7m} \qquad\qquad j \in [m];$$

$$\text{(C.15)}$$

$$\mathbb{P}\left(N_k^{-1/2}\left\|\sum_{j \in S_k} D_\eta \nabla f_j(\theta_j^*, \eta_j^*)[\Delta\eta_j]\right\|_2 > C_1' \sqrt{K} \max_{j \in S_k} s(n_j)^{-1}t\right) \leq \frac{t^{-2}}{7K} \qquad\qquad k \in [K];$$

$$\text{(C.16)}$$

$$\mathbb{P}\left(N_k^{-1/2}\left\|\sum_{j \in S_k} D_\eta^2 \nabla f_j(\theta_j^*, \bar{\eta}_j)[\Delta\eta_j, \Delta\eta_j]\right\|_2 > C_1' K m_k^{1/2} \max_{j \in S_k} n_j^{1/2} s(n_j)^{-2}t\right) \leq \frac{t^{-1}}{7K} \qquad k \in [K];$$

$$\text{(C.17)}$$

$$\mathbb{P}\left(N_k^{-1/2}\left\|\sum_{j \in S_k} \nabla f_j(\theta_j^*, \eta_j^*)\right\|_2 \geq C_1' K^{1/r_1}t\right) \leq \frac{t^{-r_1}}{7K} \qquad\qquad k \in [K];$$

$$\text{(C.18)}$$

$$\mathbb{P}\left(\sup_{\theta \in \mathcal{B}_{\theta_j^*, M}} \frac{1}{n_j}\left\|\nabla^2 f_j(\theta, \eta_j^*) - \mathbb{E}\nabla^2 f_j(\theta, \eta_j^*)\right\|_2 \geq C_1' m^{\frac{d+1}{d+r_2}} n_j^{-\frac{r_2}{2(r_2+d)}}t\right) \leq \frac{t^{-\frac{d+r_2}{d+1}}}{7m} \qquad j \in [m];$$

$$\text{(C.19)}$$

$$\mathbb{P}\left(\sup_{\theta \in \mathcal{B}_{\theta_j^*, M}} \frac{1}{n_j}\left\|\nabla^2 f_j(\theta, \hat{\eta}_j) - \nabla^2 f_j(\theta, \eta_j^*)\right\|_2 \geq C_1' m s^{-1}(n_j)t \text{ or } \left\|\hat{\eta}_j - \eta_j^*\right\|_{\mathcal{H}_j} > M\right) \leq \frac{t^{-1}}{14m} + C_0 s^{-1}(n_j) \quad j \in [m].$$

$$\text{(C.20)}$$

Let $t$ be such that

$$t < c_2 m^{-1}\{n_{\min}^{\frac{r_2}{2(r_2+d)}} \wedge s(n_{\min})\}, \qquad c_2 = \min\left(\frac{1}{14C_0}, \frac{\rho}{16C_1'}\{4 \wedge M \wedge (\delta/2)\}\right). \tag{C.21}$$

Then it can be easily verified that such a $t$ in (C.21) leads the following:

$$C_1' \cdot t \cdot \max\left\{m^{1/r_1}n_j^{-1/2}, mn_j^{-1/2}, m^{\frac{d+1}{d+r_2}}n_j^{-\frac{r_2}{2(r_2+d)}}, ms^{-1}(n_j)\right\} \leq \frac{\rho}{16}\{4 \wedge M \wedge (\delta/2)\} \tag{C.22}$$

$$\frac{t^{-1}}{14m} \geq \frac{c_2^{-1}}{14}s^{-1}(n_{\min}) \geq C_0 s^{-1}(n_{\min}) \tag{C.23}$$

Now, we can plug the results in (C.22) inside the LHS of Equations (C.14)-(C.20), and plug (C.23) into the RHS of (C.20). This yields the following

$$\mathbb{P}\left(\left\|\frac{1}{n_j}\nabla f_j(\theta_j^*, \eta_j^*)\right\|_2 \geq \frac{\rho}{16}\{M \wedge (\delta/2)\}\right) \leq \frac{t^{-r_1}}{7m} \qquad j \in [m];$$

$$\mathbb{P}\left(\frac{1}{n_j}\left\|\nabla f_j(\theta_j^*, \hat{\eta}_j) - \nabla f_j(\theta_j^*, \eta_j^*)\right\|_2 \geq \frac{\rho}{16}\{M \wedge (\delta/2)\}\right) \leq \frac{t^{-1}}{7m} \qquad j \in [m];$$

$$\mathbb{P}\left(N_k^{-1/2}\left\|\sum_{j \in S_k} D_\eta \nabla f_j(\theta_j^*, \eta_j^*)[\Delta\eta_j]\right\|_2 > C_1'\sqrt{K}\max_{j \in S_k} s(n_j)^{-1}t\right) \leq \frac{t^{-2}}{7K} \qquad k \in [K];$$

$$\mathbb{P}\left(N_k^{-1/2}\left\|\sum_{j \in S_k} D_\eta^2 \nabla f_j(\theta_j^*, \bar{\eta}_j)[\Delta\eta_j, \Delta\eta_j]\right\|_2 > C_1'K\,m_k^{1/2}\max_{j \in S_k} n_j^{1/2}s(n_j)^{-2}t\right) \leq \frac{t^{-1}}{7K} \qquad k \in [K];$$

$$\mathbb{P}\left(N_k^{-1/2}\left\|\sum_{j \in S_k}\nabla f_j(\theta_j^*, \eta_j^*)\right\|_2 \geq C_1'K^{1/r_1}t\right) \leq \frac{t^{-r_1}}{7K} \qquad k \in [K];$$

$$\mathbb{P}\left(\sup_{\theta \in \mathcal{B}_{\theta_j^*, M}}\frac{1}{n_j}\left\|\nabla^2 f_j(\theta, \eta_j^*) - \mathbb{E}\nabla^2 f_j(\theta, \eta_j^*)\right\|_2 \geq \frac{\rho}{4}\right) \leq \frac{t^{-\frac{d+r_2}{d+1}}}{7m} \qquad j \in [m];$$

$$\mathbb{P}\left(\sup_{\theta \in \mathcal{B}_{\theta_j^*, M}}\frac{1}{n_j}\left\|\nabla^2 f_j(\theta, \hat{\eta}_j) - \nabla^2 f_j(\theta, \eta_j^*)\right\|_2 \geq \frac{\rho}{4} \text{ or } \left\|\hat{\eta}_j - \eta_j^*\right\|_{\mathcal{H}_j} > M\right) \leq \frac{t^{-1}}{7m} \qquad j \in [m].$$

By the conditions $r_1, r_2 \geq 1$ in Assumption 3.1(2)(3) and $t \geq 1$, it follows that $t^{-1} \geq t^{-\frac{d+r_2}{d+1}} \vee t^{-r_1}$. By the union bound,

it holds with probability at least $1 - t^{-1}$ that the simultaneous inequalities hold:

$$\frac{1}{n_j} \left\| \nabla f_j(\theta_j^*, \eta_j^*) \right\|_2 \leq \frac{\rho}{16} \{ M \wedge (\delta/2) \} \qquad j \in [m]; \qquad \text{(C.24)}$$

$$\frac{1}{n_j} \left\| \nabla f_j(\theta_j^*, \hat{\eta}_j) - \nabla f_j(\theta_j^*, \eta_j^*) \right\|_2 \leq \frac{\rho}{16} \{ M \wedge (\delta/2) \} \qquad j \in [m]; \qquad \text{(C.25)}$$

$$N_k^{-1/2} \left\| \sum_{j \in S_k} D_\eta \nabla f_j(\theta_j^*, \eta_j^*)[\Delta \eta_j] \right\|_2 \leq C_1' \sqrt{K} \max_{j \in S_k} s(n_j)^{-1} t \qquad k \in [K]; \qquad \text{(C.26)}$$

$$N_k^{-1/2} \left\| \sum_{j \in S_k} D_\eta^2 \nabla f_j(\theta_j^*, \bar{\eta}_j)[\Delta \eta_j, \Delta \eta_j] \right\|_2 \leq C_1' K \, m_k^{1/2} \max_{j \in S_k} n_j^{1/2} s(n_j)^{-2} t \qquad k \in [K]; \qquad \text{(C.27)}$$

$$N_k^{-1/2} \left\| \sum_{j \in S_k} \nabla f_j(\theta_j^*, \eta_j^*) \right\|_2 \leq C_1' K^{1/r_1} t \qquad k \in [K]; \qquad \text{(C.28)}$$

$$\sup_{\theta \in \mathcal{B}_{\theta_j^*, M}} \frac{1}{n_j} \left\| \nabla^2 f_j(\theta, \eta_j^*) - \mathbb{E} \nabla^2 f_j(\theta, \eta_j^*) \right\|_2 \leq \frac{\rho}{4} \qquad j \in [m]; \qquad \text{(C.29)}$$

$$\sup_{\theta \in \mathcal{B}_{\theta_j^*, M}} \frac{1}{n_j} \left\| \nabla^2 f_j(\theta, \hat{\eta}_j) - \nabla^2 f_j(\theta, \eta_j^*) \right\|_2 \leq \frac{\rho}{4} \qquad j \in [m]; \qquad \text{(C.30)}$$

$$\left\| \hat{\eta}_j - \eta_j^* \right\|_{\mathcal{H}_j} \leq M \qquad j \in [m], \qquad \text{(C.31)}$$

which, by triangle inequality, implies

$$\frac{1}{n_j} \left\| \nabla f_j(\theta_j^*, \hat{\eta}_j) \right\|_2 \leq \frac{\rho}{8} \{ M \wedge (\delta/2) \} \qquad j \in [m];$$

$$\sup_{\theta \in \mathcal{B}_{\theta_j^*, M}} \frac{1}{n_j} \left\| \nabla^2 f_j(\theta, \hat{\eta}_j) - \mathbb{E} \nabla^2 f_j(\theta, \eta_j^*) \right\|_2 \leq \frac{\rho}{2} \qquad j \in [m].$$

Using the geometry of losses in Assumption 3.1(1) that $\rho I \preceq \frac{1}{n_j} \mathbb{E} \nabla^2 f_j(\theta, \eta_j^*) \preceq \kappa I$ for $\theta \in \mathcal{B}_{\theta_j^*, M}$, we have $\rho/2 \cdot I \preceq \frac{1}{n_j} \nabla^2 f_j(\theta, \hat{\eta}_j) \preceq (\kappa + \rho/2) I \preceq 3\kappa/2 \cdot I$ for $\theta \in \mathcal{B}_{\theta_j^*, M}$. Finally, define weights $w_j^{(k)} = n_j / N_k$. Then it follows from Jensen's inequality and upper bound for $\frac{1}{n_j} \| \nabla f_j(\theta_j^*, \hat{\eta}_j) \|_2$ that for $k \in [K]$,

$$\frac{1}{N_k} \left\| \sum_{j \in S_k} \nabla f_j(\theta_j^*, \hat{\eta}_j) \right\|_2 \leq \sum_{j \in S_k} w_j^{(k)} \frac{1}{n_j} \left\| \nabla f_j(\theta_j^*, \hat{\eta}_j) \right\|_2 \leq \frac{\rho}{8} \{ M \wedge (\delta/2) \}.$$

Replacing $\theta_j^*$ with $\beta_k^*$ leads to the presented inequality. $\qquad \square$

## D. Proof of main results

We first establish consistency of the cluster representatives $\hat{\beta}_k$ (Theorem D.1). We then show that this implies exact cluster recovery and an $\ell_2$ bound on $\hat{\theta}_j$ (Theorem 3.5). Finally, we prove the asymptotic normality of $\hat{\theta}_j$ within each recovered cluster (Theorem 3.6).

**Theorem D.1.** *Under Assumptions 3.1 - 3.3, if i) $n_{\min}^{\alpha\gamma}/n_{\max} > c_0$, ii) $\varepsilon_n < \min_{k \in [K]} N_k^\zeta \{ M \wedge (\delta/2) \} m^{-2} \rho/4$ for $\zeta < 1/2$, iii) $1 < t < c_2 m^{-1} \{ n_{\min}^{\frac{r_2}{2(r_2+d)}} \wedge s(n_{\min}) \}$, and iv) $\tau \in \left( c_w \{ \frac{\delta}{2} \}^{-\gamma}, c_w \{ \frac{\delta}{2} \}^{-\gamma} n_{\min}^{\alpha\gamma} \right)$, then it holds with probability*

*at least* $1 - t^{-1} - p_{\delta/4}(n_{\min})$ *that*

$$
\begin{aligned}
&\hat{\beta}_k \neq \hat{\beta}_{k'} \quad \forall k, k' \in [K], k \neq k'; \\
&\hat{\theta}_j = \hat{\beta}_k, \quad j \in S_k; \\
&\left\| \tilde{\beta}_k - \beta_k^* \right\|_2 \leq C \left\{ K^{1/r_1} + \sqrt{K} \max_{j \in S_k} s(n_j)^{-1} + K m_k^{1/2} \max_{j \in S_k} n_j^{1/2} s(n_j)^{-2} \right\} N_k^{-1/2} t, \quad \forall k \in [K]; \\
&\left\| \hat{\beta}_k - \tilde{\beta}_k \right\|_2 < \tilde{C} N_k^{\zeta - 1} \quad \forall k \in [K].
\end{aligned}
$$
(D.1)

*where* $c_0, c_2, C, \tilde{C} > 0$ *are some constants.*

*Proof.* This theorem is built upon Lemmas B.6, B.7 and C.6. From Lemmas B.6 and B.7, we know that the inequality system (D.1) holds as long as $\mathcal{E}_{1n} \cap \mathcal{E}_{2n}$ holds. In terms of probability, this means

$$
\mathbb{P}\big((\text{D.1}) \text{ holds}\big) \geq \mathbb{P}\big(\mathcal{E}_{1n} \cap \mathcal{E}_{2n}\big).
$$

Now we only need to find the probability lower bound on the event $\mathcal{E}_{1n} \cap \mathcal{E}_{2n}$ to prove the Theorem. From Lemma C.1, we get

$$
\mathbb{P}\left(\mathcal{E}_{2n}\right) \geq 1 - p_{\delta/4}(n_{\min}),
$$

while it follows from Lemma C.6 that

$$
\mathbb{P}\left(\mathcal{E}_{1n}\right) \geq 1 - t^{-1}.
$$

Thus, the joint probability is such that

$$
\mathbb{P}(\mathcal{E}_{1n} \cap \mathcal{E}_{2n}) \geq \mathbb{P}\left(\mathcal{E}_{1n}\right) + \mathbb{P}\left(\mathcal{E}_{2n}\right) - 1 \geq 1 - t^{-1} - p_{\delta/4}(n_{\min}).
$$

$\square$

### D.1. Proof of Theorem 3.5

Our proof of Theorem 3.5 is a direct application of Theorem D.1. Firstly, the first part of Theorem D.1 says

$$
\begin{aligned}
&\hat{\beta}_k \neq \hat{\beta}_{k'} \quad \forall k, k' \in [K], k \neq k'; \\
&\hat{\theta}_j = \hat{\beta}_k, \quad j \in S_k.
\end{aligned}
$$

This implies that $\hat{\theta}_j = \hat{\theta}_{j'}$ for all $j, j' \in S_k$ and $\hat{\theta}_j \neq \hat{\theta}_{j'}$ for $j \in S_k$, $j' \in S_{k'}$, $k \neq k'$.

Secondly, the second part of Theorem D.1 gives

$$
\begin{aligned}
&\left\| \tilde{\beta}_k - \beta_k^* \right\|_2 \leq C \left\{ K^{1/r_1} + \sqrt{K} \max_{j \in S_k} s(n_j)^{-1} + K m_k^{1/2} \max_{j \in S_k} n_j^{1/2} s(n_j)^{-2} \right\} N_k^{-1/2} t, \quad \forall k \in [K]; \\
&\left\| \hat{\beta}_k - \tilde{\beta}_k \right\|_2 < \tilde{C} N_k^{\zeta - 1} \quad \forall k \in [K],
\end{aligned}
$$

which, by triangle inequality, implies

$$
\|\hat{\theta}_j - \theta_j^*\|_2 \leq C \left\{ K^{1/r_1} + K^{1/2} \max_{j \in S_k} s(n_j)^{-1} + K m_k^{1/2} \max_{j \in S_k} n_j^{1/2} s(n_j)^{-2} \right\} N_k^{-1/2} t + \tilde{C} N_k^{\zeta - 1}.
$$

Now since the rate function $s(n_j) \lesssim n_j^{1/2}$ (defined in Assumption 3.2), we have

$$
\frac{n_j^{1/2} s(n_j)^{-2}}{s(n_j)^{-1}} = n_j^{1/2} s(n_j)^{-1} \gtrsim 1.
$$

Therefore, the sum of second and third term is upper bounded by

$$
K^{1/2} \max_{j \in S_k} s(n_j)^{-1} + K m_k^{1/2} \max_{j \in S_k} n_j^{1/2} s(n_j)^{-2} \lesssim K m_k^{1/2} \max_{j \in S_k} n_j^{1/2} s(n_j)^{-2} = b_{k,n}.
$$

### D.2. Proof of Theorem 3.6

Fix $k$ and any $j \in S_k$. Since $\theta_j^* = \beta_k^*$,

$$\hat{\theta}_j - \theta_j^* = (\hat{\theta}_j - \tilde{\beta}_k) + (\tilde{\beta}_k - \beta_k^*).$$

By Theorem D.1, $\hat{\theta}_j - \tilde{\beta}_k = O_p(N_k^{\zeta-1}) = o_p(N_k^{-1/2})$ since $\zeta < 1/2$. It remains to study $\tilde{\beta}_k - \beta_k^*$.

**First-order expansion.** The first order condition gives $\nabla F_k(\tilde{\beta}_k, \hat{\boldsymbol{\eta}}_k) = 0$. A Taylor expansion around $\beta_k^*$ yields

$$\sqrt{N_k}(\tilde{\beta}_k - \beta_k^*) = -\left\{N_k^{-1}\nabla^2 F_k(\bar{\beta}_k, \hat{\boldsymbol{\eta}}_k)\right\}^{-1}\left\{N_k^{-1/2}\nabla F_k(\beta_k^*, \hat{\boldsymbol{\eta}}_k)\right\},$$

for some $\bar{\beta}_k$ on the segment between $\beta_k^*$ and $\tilde{\beta}_k$.

**Hessian consistency.** Decompose

$$N_k^{-1}\nabla^2 F_k(\bar{\beta}_k, \hat{\boldsymbol{\eta}}_k) = B_0 + B_1 + B_2,$$
$$B_0 := N_k^{-1}\nabla^2 F_k(\beta_k^*, \boldsymbol{\eta}_k^*),$$
$$B_1 := N_k^{-1}\left\{\nabla^2 F_k(\bar{\beta}_k, \hat{\boldsymbol{\eta}}_k) - \nabla^2 F_k(\beta_k^*, \hat{\boldsymbol{\eta}}_k)\right\},$$
$$B_2 := N_k^{-1}\left\{\nabla^2 F_k(\beta_k^*, \hat{\boldsymbol{\eta}}_k) - \nabla^2 F_k(\beta_k^*, \boldsymbol{\eta}_k^*)\right\}.$$

By the Law of Large Number and Assumption 3.1(1), $B_0 \to_p \Psi_k$. For $B_1$, with definition

$$\Xi_j(\theta, \theta', \eta, \eta', z) = \frac{\|\nabla^2\ell_j(\theta, \eta, z) - \nabla^2\ell_j(\theta', \eta', z)\|_2}{\|\theta - \theta'\|_2 + \|\eta - \eta'\|_{\mathcal{H}_j}}$$

by the triangle inequality and the local Lipschitz control in Assumption 3.1(4),

$$\|B_1\|_2 \leq N_k^{-1}\sum_{j \in S_k}\sum_{i=1}^{n_j}\Xi_j(\bar{\beta}_k, \beta_k^*, \hat{\eta}_j, \hat{\eta}_j, Z_{ji})\,\|\bar{\beta}_k - \beta_k^*\|_2 = O_p(1) \cdot o_p(1) = o_p(1),$$

since $\|\bar{\beta}_k - \beta_k^*\| = o_p(1)$ by consistency of $\tilde{\beta}_k$. For $B_2$, similarly, by Assumption 3.2 on consistency of $\hat{\eta}_j$,

$$\|B_2\|_2 \leq N_k^{-1}\sum_{j \in S_k}\sum_{i=1}^{n_j}\Xi_j(\beta_k^*, \beta_k^*, \hat{\eta}_j, \eta_j^*, Z_{ji})\,\|\hat{\eta}_j - \eta_j^*\|_{\mathcal{H}_j} = O_p(1) \cdot o_p(1) = o_p(1).$$

Therefore,

$$N_k^{-1}\nabla^2 F_k(\bar{\beta}_k, \hat{\boldsymbol{\eta}}_k) \to_p \Psi_k. \tag{D.2}$$

**Score limit.** Write

$$N_k^{-1/2}\nabla F_k(\beta_k^*, \hat{\boldsymbol{\eta}}_k) = S_k^0 + R_k,$$
$$S_k^0 := N_k^{-1/2}\nabla F_k(\beta_k^*, \boldsymbol{\eta}_k^*), \tag{D.3}$$
$$R_k := N_k^{-1/2}\left\{\nabla F_k(\beta_k^*, \hat{\boldsymbol{\eta}}_k) - \nabla F_k(\beta_k^*, \boldsymbol{\eta}_k^*)\right\}. \tag{D.4}$$

It is immediate by central limit theorem that $S_k^0 \to_d \mathcal{N}(0, \Omega_k)$.

For each summand in $R_k$, a second-order expansion in $\eta_j^*$ gives for $j \in S_k$,

$$\nabla\ell_j(\beta_k^*, \hat{\eta}_j, Z_{ji}) - \nabla\ell_j(\beta_k^*, \eta_j^*, Z_{ji}) = D_\eta\nabla\ell_j(\beta_k^*, \eta_j^*, Z_{ji})[\Delta\eta_j] + \tfrac{1}{2}D_\eta^2\nabla\ell_j(\beta_k^*, \bar{\eta}_j, Z_{ji})[\Delta\eta_j, \Delta\eta_j],$$

with $\Delta\eta_j := \hat{\eta}_j - \eta_j^*$ and some $\bar{\eta}_j$ on the segment between $\hat{\eta}_j$ and $\eta_j^*$. Hence

$$R_k = N_k^{-1/2}\sum_{j \in S_k}\sum_{i=1}^{n_j}\left\{D_\eta\nabla\ell_j(\theta_j^*, \eta_j^*, Z_{ji})[\Delta\eta_j] + \tfrac{1}{2}D_\eta^2\nabla\ell_j(\theta_j^*, \bar{\eta}_j, Z_{ji})[\Delta\eta_j, \Delta\eta_j]\right\}. \tag{D.5}$$

From Lemma C.6

$$R_k = O_p\left(\sqrt{K}\max_{j \in S_k} s(n_j)^{-1} + Km_k^{1/2}\max_{j \in S_k} n_j^{1/2}s(n_j)^{-2}\right)$$

The condition $b_{k,n} = Km_k^{1/2}\max_{j \in S_k} n_j^{1/2}s(n_j)^{-2} = o(1)$ implies

$$R_k = o_p(1).$$

**Slutsky.** Combine (D.2), $S_k^0 \to_d \mathcal{N}(0, \Omega_k)$, and $R_k = o_p(1)$:

$$\sqrt{N_k}(\tilde{\beta}_k - \beta_k^*) \Rightarrow \mathcal{N}(0, \Psi_k^{-1}\Omega_k\Psi_k^{-1}).$$

Because $\hat{\theta}_j - \tilde{\beta}_k = o_p(N_k^{-1/2})$, we also have

$$\sqrt{N_k}(\hat{\theta}_j - \theta_j^*) \Rightarrow \mathcal{N}(0, \Psi_k^{-1}\Omega_k\Psi_k^{-1}).$$

## E. Technical lemmas

We put all the technical lemma in this section. Note that the notations in the technical lemmas are self-contained. For example, $f$ could be a function in general, instead of a loss function as in the main body.

**Lemma E.1.** *Let $f : \mathbb{R}^d \to \mathbb{R}$ be convex. Suppose there exist $x_0 \in \mathbb{R}^d$, $g \in \partial f(x_0)$, and constants $\rho > 0$, $r > 0$ such that $\|g\|_2 < (\rho r)/2$ and, for all $x \in \mathcal{B}_{x_0, r}$,*

$$f(x) \geq f(x_0) + \langle g, x - x_0 \rangle + \tfrac{\rho}{2}\|x - x_0\|_2^2.$$

*Then every global minimizer lies in $\mathcal{B}_{x_0, 2\|g\|_2/\rho}$, i.e. $\arg\min_{x \in \mathbb{R}^d} f(x) \subseteq \mathcal{B}_{x_0, 2\|g\|_2/\rho}$.*

*Moreover, if $f$ is twice differentiable and $\nabla^2 f(x) \succeq \rho I$ for all $x \in \mathcal{B}_{x_0, r}$, then $f$ has a unique minimizer $x^\star$ and*

$$\|x^\star - x_0\|_2 \leq \|\nabla f(x_0)\|_2/\rho.$$

*Proof.* Proof can be found in Lemma F.1 of (Duan & Wang, 2023). ☐

**Lemma E.2.** *For any $j \in [m]$, let $f_j : \mathbb{R}^d \to \mathbb{R}$ be a convex function whose minimizer is $\tilde{x}_j = \arg\min_x f_j(x)$. Suppose $\forall j \in [m]$, there exist $\rho_j, r > 0$ such that $\nabla^2 f_j(x) \succeq \rho_j I$, $\forall x \in B(\tilde{x}_j, r)$. Then it follows that*

$$\|\boldsymbol{f}_j\|_2 \geq \min\{\rho_j\|x - \tilde{x}_j\|, \rho_j r\}, \quad \forall \boldsymbol{f}_j \in \partial f_j(x), \quad x \in \mathbb{R}^d. \tag{E.1}$$

*Moreover, let $h(x_1, \ldots, x_m)$ be a convex function that is $\lambda_j$-Lipschitz in $x_j$, $\forall j \in [m]$. Define minimizer*

$$(\hat{x}_1, \ldots, \hat{x}_m) = \arg\min_{x_1, \ldots, x_m} \sum_{j \in [m]} f_j(x_j) + h(x_1, \ldots, x_m).$$

*For any $j \in [m]$, if $\lambda_j < \rho_j r$, then $\hat{x}_j \in \mathcal{B}_{\tilde{x}_j, \lambda_j/\rho_j}$.*

*Proof.* Proof of Equation (E.1) can be found in Lemma F.2 of (Duan & Wang, 2023). Now we focus on the proof of the second part. For any $j \in [m]$, by the first-order condition, there exist $\boldsymbol{f}_j \in \partial f_j(\hat{x}_j)$ and $\boldsymbol{h}_j \in \partial_{x_j} h(\hat{x}_1, \ldots, \hat{x}_m)$ such that $\boldsymbol{f}_j + \boldsymbol{h}_j = 0$. Because $h$ is $\lambda_j$-Lipschitz in $x_j$ and the condition that $\lambda_j < \rho_j r$, we have $\|\boldsymbol{f}_j\| = \|\boldsymbol{h}_j\| \leq \lambda_j < \rho_j r$. Moreover, it has been shown in (E.1) that $\|\boldsymbol{f}_j\|_2 \geq \min\{\rho_j\|x - \tilde{x}_j\|, \rho_j r\}$, so it can only hold that $\lambda_j \geq \|\boldsymbol{f}_j\|_2 \geq \rho_j\|\hat{x}_j - \tilde{x}_j\|$, and thus $\hat{x}_j \in \mathcal{B}_{\tilde{x}_j, \lambda_j/\rho_j}$. ☐

**Lemma E.3.** *Let $\{S_1, \ldots, S_K\}$ be a partition of $[m]$, $m_k = |S_k|$, and $\Lambda_{kk'} = m_k m_{k'}\varepsilon_n$ for some constant $\varepsilon_n \geq 0$. For arbitrary $\theta_j \in \mathbb{R}^d$ with $j \in [m]$, define*

$$(z_1^*, \ldots, z_K^*) = \arg\min_{z_k \in S_k, \forall k \in [K]} \sum_{k \in [K]} \sum_{k' \neq k, k' \in [K]} \Lambda_{kk'}\|\theta_{z_k} - \theta_{z_{k'}}\|_2.$$

*Then it follows that*

$$\sum_{k \in [K]} \sum_{k' \neq k, k' \in [K]} \sum_{j \in S_k} \sum_{j' \in S_{k'}} \varepsilon_n\|\theta_j - \theta_{j'}\|_2 \geq \sum_{k \in [K]} \sum_{k' \neq k, k' \in [K]} \Lambda_{kk'}\|\theta_{z_k^*} - \theta_{z_{k'}^*}\|_2.$$

*Proof.* Due to the optimality of $z_k^*$, it holds for arbitrary $z_k \in S_k$ for $k \in [K]$ that

$$\sum_{k \in [K]} \sum_{k' \neq k, k' \in [K]} \Lambda_{kk'} \|\theta_{z_k^*} - \theta_{z_{k'}^*}\|_2 \leq \sum_{k \in [K]} \sum_{k' \neq k, k' \in [K]} \Lambda_{kk'} \|\theta_{z_k} - \theta_{z_{k'}}\|_2. \tag{E.2}$$

Now, we consider a uniform probability distribution $P$ over $(z_1, \ldots, z_K)$ such that

$$P(z_1 = j_1, \ldots, z_K = j_K) = 1 / \prod_{k \in [K]} m_k.$$

Note that this distribution has pairwise marginal

$$P(z_k = j_1, z_{k'} = j_{k'}) = 1/(m_k m_{k'}).$$

Then it follows that

$$\mathbb{E}_P \|\theta_{z_k} - \theta_{z_{k'}}\|_2 = \Lambda_{kk'}^{-1} \sum_{j \in S_k} \sum_{j' \in S_{k'}} \varepsilon_n \|\theta_j - \theta_{j'}\|_2.$$

Hence, applying $\mathbb{E}_P$ to both sides of Equation (E.2), whose LHS is independent of $z_k$, yields

$$\sum_{k \in [K]} \sum_{k' \neq k, k' \in [K]} \Lambda_{kk'} \|\theta_{z_k^*} - \theta_{z_{k'}^*}\|_2 \leq \sum_{k \in [K]} \sum_{k' \neq k, k' \in [K]} \sum_{j \in S_k} \sum_{j' \in S_{k'}} \varepsilon_n \|\theta_j - \theta_{j'}\|_2.$$

$\square$

# F. Construction of Orthogonal Loss in DID Model

We estimate the task-specific treatment effect under a Difference-in-Differences (DID) design using the doubly robust orthogonal score of (Sant'Anna & Zhao, 2020). For unit $i$ in task $j$, let us define the difference between the post-treatment and pre-treatment individual-level outcomes as

$$\Delta Y_{ji} = Y_{ji1} - Y_{ji0}.$$

First, using the first subsample $\mathcal{D}_{j,1}$, we estimate the nuisance functions using LightGBM to obtain $\hat{\pi}_j(x)$ and $\hat{m}_j(x)$, where $\hat{\pi}_j(x)$ estimates $\mathbb{P}(D_j = 1 \mid X_j = x)$ and $\hat{m}_j(x)$ estimates $\mathbb{E}[\Delta Y_j \mid D = 0, X = x]$. Using these estimates, define the following:

$$\widehat{\overline{D}}_j = \frac{1}{|\mathcal{D}_j^{(1)}|} \sum_{i \in \mathcal{D}_j^{(1)}} D_{ji}, \qquad \widehat{\overline{v}}_j = \frac{1}{|\mathcal{D}_j^{(1)}|} \sum_{i \in \mathcal{D}_j^{(1)}} \frac{\hat{\pi}_j(X_{ji})(1 - D_{ji})}{1 - \hat{\pi}_j(X_{ji})}.$$

Then, on the second subsample $\mathcal{D}_{j,2}$, we construct the normalized weights

$$\hat{w}_{1,ji} = \frac{D_{ji}}{\widehat{\overline{D}}_j}, \qquad \hat{w}_{0,ji} = \frac{1}{\widehat{\overline{v}}_j} \left( \frac{\hat{\pi}_j(X_{ji})(1 - D_{ji})}{1 - \hat{\pi}_j(X_{ji})} \right),$$

and set

$$\hat{A}_{ji} = (\hat{w}_{1,ji} - \hat{w}_{0,ji})(\Delta Y_{ji} - \hat{m}_j(X_{ji})).$$

The orthogonal loss for task $j$ takes the quadratic form

$$f_j^\dagger(\theta_j) = a_j(\theta_j - b_j)^2,$$

where

$$a_j = \frac{1}{|\mathcal{D}_j^{(2)}|} \sum_{i \in \mathcal{D}_j^{(2)}} \hat{w}_{1,ji}, \qquad b_j = \frac{\sum_{i \in \mathcal{D}_j^{(2)}} \hat{A}_{ji}}{\sum_{i \in \mathcal{D}_j^{(2)}} \hat{w}_{1,ji}}.$$

This is the loss we use for the estimation of DID model in Section 4.

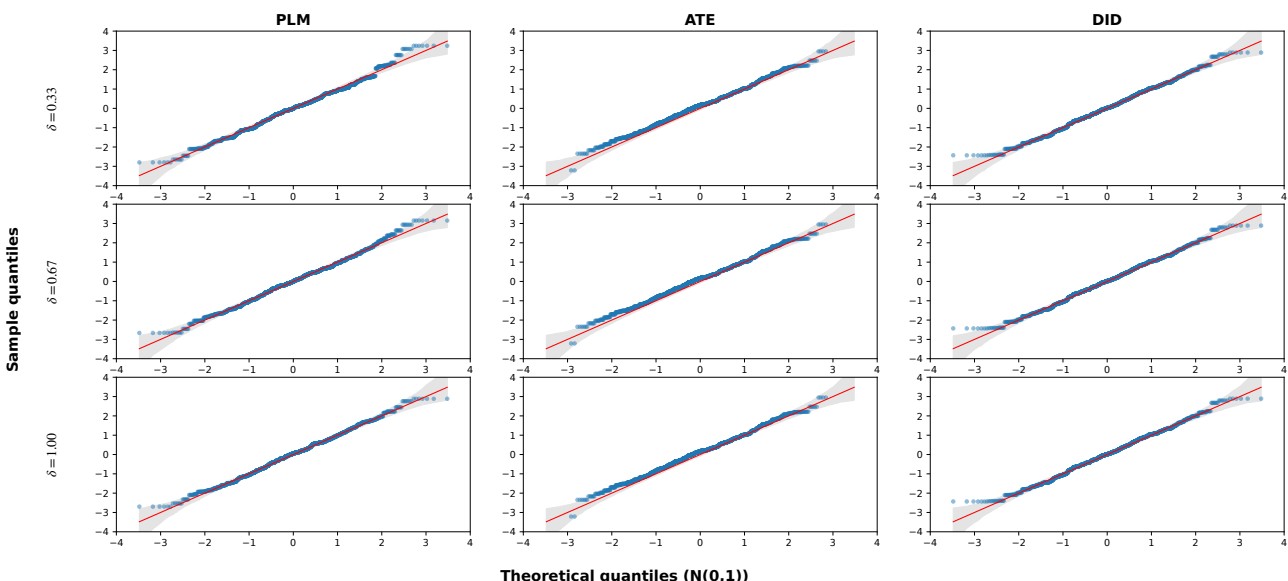

*Figure 4.* Normality diagnostics via QQ-plots with 99% confidence bands for standardized adaptive fusion estimators $Z_j = (\hat{\theta}_j - \theta_j^*)/\widehat{\mathrm{SE}}(\hat{\theta}_j)$.

## G. Normality Check

To assess the distributional behavior of the proposed estimator, we examine quantile–quantile (QQ) plots of the standardized estimators $Z_j = (\hat{\theta}_j - \theta_j^*)/\widehat{\mathrm{SE}}(\hat{\theta}_j)$. Figure 4 displays $3 \times 3$ QQ-plots corresponding to three simulation models (PLM, ATE, DID) and three signal strengths $\delta \in \{1/3, 2/3, 1\}$. Each panel compares the empirical quantiles of $\{Z_j\}$ to the theoretical quantiles of the $\mathcal{N}(0,1)$ and along with 99% confidence bands.

Across all models and values of $\delta$, the QQ-plots show that the standardized estimators align closely with the $\mathcal{N}(0,1)$ reference line and lie within the confidence bands. This indicates that the asymptotic normal approximation holds well. Overall, the QQ-plots provide strong empirical support for the claim that the adaptive fusion estimators are approximately normal in finite samples, corroborating the theoretical result in Theorem 3.6.

## H. Additional Simulation Results

Using the same simulation setup as in Section 4, we evaluate estimation accuracy using the *cluster-size weighted RMSE*

$$\mathrm{wRMSE} = \sqrt{\frac{1}{B} \sum_{j=1}^{m} N_{q(j)} (\hat{\theta}_j - \theta_j^*)^2},$$

where $q : [m] \to [K]$ maps each task $j$ to its true cluster index $k$ and the normalizing constant $B = \sum_{j=1}^{m} N_{q(j)}$. Intuitively, clusters with more data contribute proportionally more to the error.

To assess recovery of the latent task groups, we use the *Adjusted Rand Index (ARI)*. Let $\{S_k\}_{k=1}^{K}$ and $\{\hat{S}_\ell\}_{\ell=1}^{\hat{K}}$ denote the true and estimated task partitions, respectively. The ARI between the two partitions is defined as

$$\mathrm{ARI} = \frac{\sum_{k,\ell} \binom{|S_k \cap \hat{S}_\ell|}{2} - \frac{\sum_k \binom{|S_k|}{2} \sum_\ell \binom{|\hat{S}_\ell|}{2}}{\binom{m}{2}}}{\frac{1}{2}\left[\sum_k \binom{|S_k|}{2} + \sum_\ell \binom{|\hat{S}_\ell|}{2}\right] - \frac{\sum_k \binom{|S_k|}{2} \sum_\ell \binom{|\hat{S}_\ell|}{2}}{\binom{m}{2}}},$$

which satisfies $\mathrm{ARI} \in [-1, 1]$, with 1 indicating perfect agreement and values near 0 corresponding to random cluster assignments.

*Table 2.* Weighted RMSE and ARI under PLM, ATE, and DID.

| | $\delta$ | Per | ARMUL(K-1) | ARMUL(K) | ARMUL(K+1) | CN | FC | MeTaG | Ada |
|---|---|---|---|---|---|---|---|---|---|
| **(a) Weighted RMSE** | | | | | | | | | |
| **PLM** | 1/3 | 11.75 | 14.13 | 6.60 | 8.64 | 17.41 | 19.59 | 55.44 | **4.46** |
| | 2/3 | 12.15 | 14.13 | 7.21 | 9.11 | 29.23 | 12.40 | 57.01 | **5.27** |
| | 1 | 12.67 | 14.52 | 8.01 | 9.71 | 41.83 | 11.22 | 58.35 | **6.25** |
| **ATE** | 1/3 | 23.86 | 70.88 | 10.02 | 16.83 | 25.07 | 26.25 | 56.41 | **8.82** |
| | 2/3 | 23.86 | 76.66 | 9.80 | 16.83 | 33.02 | 24.03 | 56.41 | **8.82** |
| | 1 | 23.86 | 74.09 | 9.80 | 16.83 | 43.13 | 20.51 | 56.41 | **8.82** |
| **DID** | 1/3 | 14.75 | 77.88 | 6.14 | 10.19 | 17.85 | 19.77 | 54.84 | **5.27** |
| | 2/3 | 14.75 | 164.66 | 6.14 | 10.22 | 27.62 | 14.11 | 54.84 | **5.27** |
| | 1 | 14.75 | 254.27 | 6.14 | 10.23 | 38.93 | 12.68 | 54.84 | **5.27** |
| **(b) ARI** | | | | | | | | | |
| **PLM** | 1/3 | 0.03 | 0.55 | **1.00** | 0.88 | 0.04 | 0.04 | 0.98 | **1.00** |
| | 2/3 | 0.04 | 0.56 | **1.00** | 0.88 | 0.04 | 0.04 | 0.98 | **1.00** |
| | 1 | 0.03 | 0.56 | **1.00** | 0.88 | 0.04 | 0.04 | 0.98 | **1.00** |
| **ATE** | 1/3 | 0.02 | 0.52 | **1.00** | 0.87 | 0.02 | 0.02 | 0.67 | **0.99** |
| | 2/3 | 0.02 | 0.55 | **1.00** | 0.87 | 0.02 | 0.02 | 0.67 | **0.99** |
| | 1 | 0.02 | 0.55 | **1.00** | 0.87 | 0.02 | 0.02 | 0.67 | **0.99** |
| **DID** | 1/3 | 0.03 | 0.54 | **1.00** | 0.88 | 0.03 | 0.03 | 0.93 | **1.00** |
| | 2/3 | 0.03 | 0.56 | **1.00** | 0.88 | 0.03 | 0.03 | 0.93 | **1.00** |
| | 1 | 0.03 | 0.56 | **1.00** | 0.88 | 0.03 | 0.03 | 0.93 | **1.00** |

Table 2 reports cluster-size weighted RMSE and ARI for the PLM, ATE, and DID settings across $\delta \in \{1/3, 2/3, 1\}$. As in Section 4, we compare the estimators: Personalized, ARMUL with $K - 1$, $K$, and $K + 1$ many clusters, CN, FC, MeTaG, and the proposed Adaptive Fusion method. It is immediate from Table 2 that our proposed estimators achieve uniformly smaller RMSE compared to all other estimators, along with achieving near-perfect clustering. This indicates that our method simultaneously recovers the true task clusters and effectively pools information within them. *ARMUL* with the oracle number of clusters $(K)$ attains a competitive ARI but has a higher RMSE than ours. Furthermore, when $K$ is misspecified in ARMUL, it exhibits substantial degradation: using $K - 1$ underestimates the true cluster count as expected and forces heterogeneous tasks to merge, producing large pooling bias, high RMSE, and lower ARI; using $K + 1$ tolerates over-clustering somewhat better, but underperforms in terms of RMSE compared to the adaptive estimator. *MeTaG* effectively identifies cluster structure, but exhibits substantially larger RMSE, likely due to bias (as discussed later). Finally, the *Personalized*, *CN*, and *FC* estimators do not borrow strength across tasks, yielding the largest RMSE and near-zero ARI throughout. Overall, the proposed adaptive estimator achieves the best trade-off between bias and variance.

## I. Fixed vs. Adaptive Fusion

We conduct an additional study comparing the proposed *Adaptive* fusion penalty to a *Fixed* baseline under the same experimental setup as the PLM model in Section 4. The fixed estimator replaces the adaptive weights $\lambda_{jj'}$ in Equation (2.4) with a constant $\lambda > 0$, while the adaptive scheme learns heterogeneous pairwise penalties from the data.

Figure 5 reports RMSE and ARI at $\delta = 1/3$ for three configurations: adaptive $\lambda_{jj'}$, fixed $\lambda = 10^{-3}$, and fixed $\lambda = 10^{-2}$. The adaptive method achieves both the lowest RMSE and the highest ARI, indicating that it simultaneously improves estimation accuracy and recovers the latent cluster structure. In contrast, the fixed baseline exhibits a clear bias–variance tradeoff. A moderate penalty ($\lambda = 10^{-3}$) yields the smallest RMSE among the two fixed choices but suffers from near-zero ARI, implying accurate point estimation but almost no clustering. As the penalty increases ($\lambda = 10^{-2}$), the ARI improves substantially due to stronger pooling, but at the cost of larger RMSE from over-shrinkage.

To further diagnose this effect, Figure 6 plots the empirical distribution of all the $\hat{\theta}_j$ at $\delta = 0.33$. The adaptive estimator produces three tight, approximately Gaussian clusters centered at the true values. The fixed baseline again reveals the same bias–variance tradeoff. When $\lambda = 10^{-3}$, the distributions remain wide and separated, reflecting low bias but weak pooling. When $\lambda = 10^{-2}$, the variance contracts to a level similar to the adaptive method, resulting in better clustering; however, the

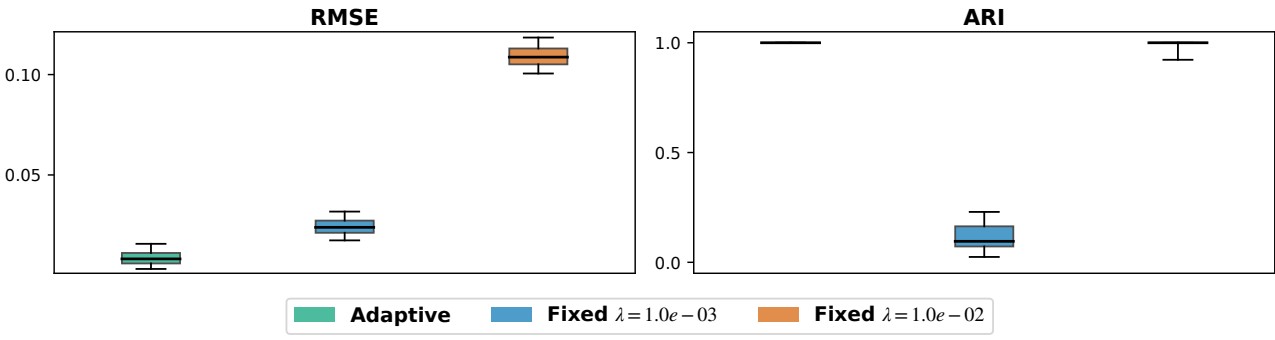

*Figure 5.* RMSE and ARI at $\delta = 1/3$ for adaptive versus fixed fusion under PLM model. Adaptive achieves both the lowest RMSE and highest ARI, whereas fixed penalties exhibit a tradeoff: $\lambda = 10^{-3}$ gives lower RMSE but near-zero ARI, while $\lambda = 10^{-2}$ improves ARI at the cost of higher RMSE.

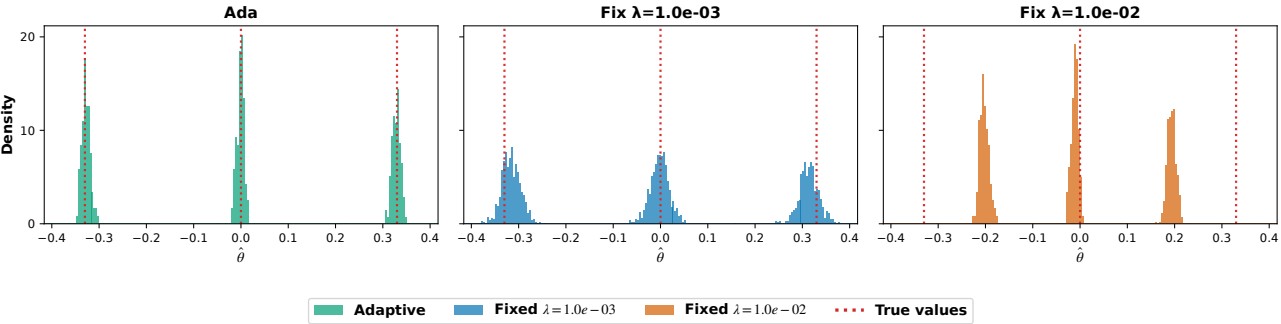

*Figure 6.* Distribution of $\hat{\theta}_j$ at $\delta = 1/3$ under PLM model. Adaptive produces three tight clusters aligned with truth (red lines). Fixed $\lambda = 10^{-3}$ shows weak pooling (low bias, high variance), while $\lambda = 10^{-2}$ increases pooling (lower variance) but introduces bias by shifting cluster centers toward the global mean.

estimated centers are shifted toward the global mean, producing a visible bias relative to the true cluster centers.

## J. General $R$-fold Cross-Fitting

Our main analysis is based on simple sample splitting: for each task $j$, we estimate the nuisance functions on one half $\mathcal{D}_{j,1}$ and evaluate the orthogonal loss on the other half $\mathcal{D}_{j,2}$. A natural refinement is to use $R$-fold cross-fitting at the loss level.

Specifically, split task $\mathcal{D}_j$ equally into $R$ folds $\mathcal{D}_{j,1}, \ldots, \mathcal{D}_{j,R}$. For each fold $r$, estimate the nuisance functions on the complement $\mathcal{D}_j^{(-r)} = \bigcup_{s \neq r} \mathcal{D}_{j,s}$ to obtain $\hat{\eta}_j^{(-r)}$, and form the fold-wise orthogonal loss $f_j^{\dagger,(r)}(\theta_j; \hat{\eta}_j^{(-r)})$. We then define the cross-fitted loss

$$\bar{f}_j^{\dagger}(\theta_j) = \frac{1}{R} \sum_{r=1}^{R} f_j^{\dagger,(r)}(\theta_j; \hat{\eta}_j^{(-r)}),$$

and the adaptive fusion objective:

$$\hat{\theta} = \arg\min_{\theta} \sum_{j=1}^{m} \bar{f}_j^{\dagger}(\theta_j) + \sum_{1 \leq j' < j \leq m} \lambda_{jj'} \|\theta_j - \theta_{j'}\|_2,$$

where $\lambda_{jj'}$ is same as Equation (2.4). Using standard analysis for cross-fitted estimators, along with the fact that our estimators are asymptotically linear, it is immediate that the theoretical guarantees will continue to hold for $R$-fold cross-fitted estimators.

## K. Data Preprocessing of RECS 2020

**Control of data leakage.** Using the RECS 2020 codebook `eia.gov/consumption/residential/data/2020`, we retain a single geographic key, `State`, as the task identifier and construct a set of predictors by *excluding* variables that could leak the outcome $Y$ or the main regressor $T$: (i) variables directly related to outcome/regressor (`DOLLAR`, `COST`, etc.); (ii) direct fuel-volume measures (`CUFEETNG`, `GALLONLP`, `GALLONFO`); (iii) redundant geography (task identifiers like `REGIONC`, `STATE_FIPS`, `state_postal`, `state_name`).

**Cleaning and encoding.** Starting from the constructed dataset, we clean the dataset by: (i) dropping columns with $> 40\%$ missing values; (ii) removing near-constant numeric features (variance $< 10^{-8}$); (iii) one-hot encoding categorical variables. (iv) normalize all the numerical variables.

## L. Extension to within-cluster heterogeneity

We consider an extended setup allowing for within-cluster heterogeneity. Let $\{S_k\}_{k=1}^K$ be an unknown partition of $[m]$. For each cluster $k \in [K]$, there exists a centroid $\beta_k^*$ such that the target parameter $\theta_j^*$ lies in a $\xi_k$-neighborhood of $\beta_k^*$, i.e.,

$$\|\theta_j^* - \beta_k^*\|_2 \le \xi_k. \tag{L.1}$$

Define

$$\xi_{\max} := \max_{k \in [K]} \xi_k.$$

Throughout this section, we assume that $n_{\min}$ is sufficiently large and the within-cluster perturbation is bounded by a constant,

$$\xi_{\max} \le \frac{\rho}{24\kappa}\{(M/2) \wedge (\delta/2)\}, \qquad n_{\min}^{-\alpha} < \frac{1}{2}. \tag{L.2}$$

Across clusters, we impose the same separation condition as before: for $k \ne k'$,

$$\|\beta_k^* - \beta_{k'}^*\|_2 \ge \delta.$$

Under the above setup, Theorems 3.7 and 3.8 follow as extensions of Theorems 3.5 and 3.6, respectively. The proof can be found in Section L.4.

### L.1. Roadmap on proof of the extended results

Our proof follows the same roadmap outlined in Section A. In the deterministic analysis (Section L.2), we show that if both events $\mathcal{E}_{1n}$ and $\mathcal{E}_{2n}$, defined in B.1 and B.2, hold, then: (i) *exact shrinkage*, where for every $j \in S_k$, the estimator $\hat{\theta}_j$ fuses to the reference $\hat{\beta}_k$; and (ii) *oracle approximation*, where the reference $\hat{\beta}_k$ converges to the oracle $\tilde{\beta}_k$ at rate $o(N_k^{-1/2})$ under certain conditions.

In Section L.3, we show that $\mathbb{P}(\mathcal{E}_{1n} \cap \mathcal{E}_{2n}) \to 1$, so the above properties hold with high probability. Finally, Section L.4 combines these results to establish Theorems 3.7 and 3.8.

**Note:** most arguments under the "perturbed" model follow the same lines of proof as those for the "clean" model discussed earlier in Section A. We therefore highlight only the key differences to avoid unnecessary repetition.

### L.2. Deterministic Analysis

Throughout this section, we work conditionally on the event $\mathcal{E}_{1n} \cap \mathcal{E}_{2n}$, where $\mathcal{E}_{2n}$ is defined in Definition B.2. The event $\mathcal{E}_{1n}$ is stated as in Definition B.1, except that the gradient bounds are slightly tightened to account for the smaller neighborhood used in the perturbed setting:

$$\left\|\tfrac{1}{n_j}\nabla f_j(\theta_j^*, \hat{\eta}_j)\right\|_2 < \tfrac{\rho}{8}\{(M/2) \wedge (\delta/2)\}, \qquad\qquad \forall j \in [m],$$

$$\left\|\tfrac{1}{N_k}\nabla F_k(\beta_k^*, \hat{\boldsymbol{\eta}}_k)\right\|_2 < \tfrac{\rho}{8}\{(M/2) \wedge (\delta/2)\}, \qquad\qquad \forall k \in [K]. \tag{L.3}$$

This modification only changes the constants in the definition of $\mathcal{E}_{1n}$. Consequently, the high-probability verification of $\mathcal{E}_{1n}$ remains unchanged up to constants.

The perturbed model extends the clean model by allowing task-specific deviations within each cluster, that is, $\theta_j^* \neq \beta_k^*$ for some $j \in S_k$. The arguments therefore follow the same deterministic strategy developed in Section B, with only the modifications needed to account for these within-cluster perturbations. Specifically, Lemmas B.3–B.7 are replaced by their perturbed counterparts, Lemmas L.1–L.5, respectively.

**Lemma L.1.** *Assume event $\mathcal{E}_{2n}$ holds. Then, for any*

$$\tau \in \left( c_w \left\{ \frac{\delta}{2} \right\}^{-\gamma}, \ c_w \left\{ \frac{\delta}{2} n_{\min}^{-\alpha} + 2\xi_{\max} \right\}^{-\gamma} \right),$$

*the following hold:*

$$\lambda_{jj'} = \varepsilon_n, \qquad\qquad \forall j \in S_k, \ \forall j' \in S_{k'}, \ k \neq k',$$

$$\lambda_{jj'} = w_{jj'} > c_w \left\{ \frac{\delta}{2} n_{\min}^{-\alpha} + 2\xi_{\max} \right\}^{-\gamma}, \qquad\qquad \forall j \neq j' \in S_k, \ \forall k \in [K],$$

*where $\lambda_{jj'}$ and $w_{jj'}$ are defined as in Equation (2.4).*

*Proof.* Recall that

$$w_{jj'} = c_w \|\hat{\theta}_j^{\text{init}} - \hat{\theta}_{j'}^{\text{init}}\|_2^{-\gamma},$$

and

$$\lambda_{jj'} = \begin{cases} \varepsilon_n, & \text{if } w_{jj'} \leq \tau, \\ w_{jj'}, & \text{if } w_{jj'} > \tau. \end{cases}$$

Equation (L.2) implies that

$$\xi_{\max} \leq \frac{\delta}{8}, \qquad n_{\min}^{-\alpha} < \frac{1}{2},$$

which further implies

$$\frac{\delta}{2} n_{\min}^{-\alpha} + 2\xi_{\max} < \frac{\delta}{2}.$$

$j, j'$ **belong to different clusters.** Let $j \in S_k$ and $j' \in S_{k'}$ with $k \neq k'$. By centroid separation and the perturbation bound,

$$\|\theta_j^* - \theta_{j'}^*\|_2 \geq \|\beta_k^* - \beta_{k'}^*\|_2 - \|\theta_j^* - \beta_k^*\|_2 - \|\theta_{j'}^* - \beta_{k'}^*\|_2$$
$$\geq \delta - 2\xi_{\max}.$$

By the definition of $\mathcal{E}_{2n}$,

$$\|\hat{\theta}_j^{\text{init}} - \theta_j^*\|_2 < \frac{\delta}{4} n_j^{-\alpha} \leq \frac{\delta}{4} n_{\min}^{-\alpha},$$

and similarly,

$$\|\hat{\theta}_{j'}^{\text{init}} - \theta_{j'}^*\|_2 < \frac{\delta}{4} n_{j'}^{-\alpha} \leq \frac{\delta}{4} n_{\min}^{-\alpha}.$$

Therefore, by the triangle inequality,

$$\|\hat{\theta}_j^{\text{init}} - \hat{\theta}_{j'}^{\text{init}}\|_2 \geq \|\theta_j^* - \theta_{j'}^*\|_2 - \|\hat{\theta}_j^{\text{init}} - \theta_j^*\|_2 - \|\hat{\theta}_{j'}^{\text{init}} - \theta_{j'}^*\|_2$$
$$> \delta - 2\xi_{\max} - \frac{\delta}{2} n_{\min}^{-\alpha}.$$

Since

$$\frac{\delta}{2} n_{\min}^{-\alpha} + 2\xi_{\max} < \frac{\delta}{2},$$

we have

$$\|\hat{\theta}_j^{\text{init}} - \hat{\theta}_{j'}^{\text{init}}\|_2 > \frac{\delta}{2}.$$

Thus,

$$w_{jj'} = c_w \|\hat\theta_j^{\text{init}} - \hat\theta_{j'}^{\text{init}}\|_2^{-\gamma} < c_w \left\{\frac{\delta}{2}\right\}^{-\gamma} < \tau.$$

Hence $\lambda_{jj'} = \varepsilon_n$.

$j, j'$ **belong to the same cluster.** Now suppose $j, j' \in S_k$ with $j \neq j'$. By the triangle inequality, the definition of $\mathcal{E}_{2n}$, and the perturbation bound,

$$\|\hat\theta_j^{\text{init}} - \hat\theta_{j'}^{\text{init}}\|_2 \leq \|\hat\theta_j^{\text{init}} - \theta_j^*\|_2 + \|\theta_j^* - \beta_k^*\|_2 + \|\beta_k^* - \theta_{j'}^*\|_2 + \|\theta_{j'}^* - \hat\theta_{j'}^{\text{init}}\|_2$$
$$< \frac{\delta}{4}\{n_j^{-\alpha} + n_{j'}^{-\alpha}\} + 2\xi_k$$
$$\leq \frac{\delta}{2}n_{\min}^{-\alpha} + 2\xi_{\max}.$$

Therefore,

$$w_{jj'} = c_w \|\hat\theta_j^{\text{init}} - \hat\theta_{j'}^{\text{init}}\|_2^{-\gamma} > c_w \left\{\frac{\delta}{2}n_{\min}^{-\alpha} + 2\xi_{\max}\right\}^{-\gamma} > \tau.$$

Hence $\lambda_{jj'} = w_{jj'}$, and moreover

$$\lambda_{jj'} = w_{jj'} > c_w \left\{\frac{\delta}{2}n_{\min}^{-\alpha} + 2\xi_{\max}\right\}^{-\gamma}.$$

This proves the claim. $\qquad\square$

**Lemma L.2.** *By definition of $\mathcal{E}_{1n}$, for every $j \in S_k$,*

$$\nabla^2 f_j(\beta, \hat\eta_j) \succeq \rho_j I, \qquad \forall \beta \in \mathcal{B}_{\theta_j^*, M}.$$

*Therefore, for any $k \in [K]$,*

$$\nabla^2 F_k(\beta, \hat{\boldsymbol{\eta}}_k) \succeq \sum_{j \in S_k} \rho_j I, \qquad \forall \beta \in \mathcal{B}_{\beta_k^*, M-\xi_k}.$$

*If $\{\Lambda_{kk'}\}_{k \neq k'}$ additionally satisfy*

$$\frac{2\|\nabla F_k(\beta_k^*, \hat{\boldsymbol{\eta}}_k)\|_2 + \sum_{k' \neq k} \Lambda_{kk'}}{\sum_{j \in S_k} \rho_j} < \min\{M - \xi_k, \delta/2\}, \qquad \forall k \in [K], \qquad\text{(L.4)}$$

*then we have*

$$\left\|\tilde\beta_k - \beta_k^*\right\|_2 \leq \frac{\|\nabla F_k(\beta_k^*, \hat{\boldsymbol{\eta}}_k)\|_2}{\sum_{j \in S_k} \rho_j}, \qquad \left\|\hat\beta_k - \tilde\beta_k\right\|_2, \leq \frac{\sum_{k' \neq k} \Lambda_{kk'}}{\sum_{j \in S_k} \rho_j}, \qquad \forall k \in [K].$$

*Moreover,*

$$\hat\beta_k \neq \hat\beta_{k'}, \qquad \forall k \neq k' \in [K].$$

*Proof.* We use the shorthand notation

$$\tilde\rho_k := \sum_{j \in S_k} \rho_j.$$

First, observe that for any $\beta \in \mathcal{B}_{\beta_k^*, M-\xi_k}$ and any $j \in S_k$,

$$\|\beta - \theta_j^*\|_2 \leq \|\beta - \beta_k^*\|_2 + \|\beta_k^* - \theta_j^*\|_2 < M - \xi_k + \xi_k = M.$$

Hence

$$\mathcal{B}_{\beta_k^*, M-\xi_k} \subseteq \bigcap_{j \in S_k} \mathcal{B}_{\theta_j^*, M}.$$

By the definition of $\mathcal{E}_{1n}$, this implies

$$\nabla^2 F_k(\beta, \hat{\boldsymbol{\eta}}_k) = \sum_{j \in S_k} \nabla^2 f_j(\beta, \hat{\eta}_j) \succeq \tilde{\rho}_k I, \qquad \forall \beta \in \mathcal{B}_{\beta_k^*, M-\xi_k}.$$

Therefore $F_k(\cdot, \hat{\boldsymbol{\eta}}_k)$ is $\tilde{\rho}_k$-strongly convex on $\mathcal{B}_{\beta_k^*, M-\xi_k}$.

By (L.4),

$$\frac{\|\nabla F_k(\beta_k^*, \hat{\boldsymbol{\eta}}_k)\|_2}{\tilde{\rho}_k} < M - \xi_k.$$

Thus, by Lemma E.1, the oracle minimizer $\tilde{\beta}_k$ satisfies

$$\|\tilde{\beta}_k - \beta_k^*\|_2 \leq \frac{\|\nabla F_k(\beta_k^*, \hat{\boldsymbol{\eta}}_k)\|_2}{\tilde{\rho}_k}.$$

Next, by (L.4), there exists $r_k$ such that

$$\sum_{k' \in [K] \setminus \{k\}} \frac{\Lambda_{kk'}}{\tilde{\rho}_k} < r_k < M - \xi_k - \frac{2\|\nabla F_k(\beta_k^*, \hat{\boldsymbol{\eta}}_k)\|_2}{\tilde{\rho}_k}.$$

For any $x \in \mathcal{B}_{\tilde{\beta}_k, r_k}$, we have

$$\begin{aligned}
\|x - \beta_k^*\|_2 &\leq \|x - \tilde{\beta}_k\|_2 + \|\tilde{\beta}_k - \beta_k^*\|_2 \\
&< r_k + \frac{\|\nabla F_k(\beta_k^*, \hat{\boldsymbol{\eta}}_k)\|_2}{\tilde{\rho}_k} \\
&< M - \xi_k - \frac{\|\nabla F_k(\beta_k^*, \hat{\boldsymbol{\eta}}_k)\|_2}{\tilde{\rho}_k} \leq M - \xi_k.
\end{aligned}$$

Therefore

$$\mathcal{B}_{\tilde{\beta}_k, r_k} \subseteq \mathcal{B}_{\beta_k^*, M-\xi_k} \subseteq \bigcap_{j \in S_k} \mathcal{B}_{\theta_j^*, M}.$$

Hence $F_k(\cdot, \hat{\boldsymbol{\eta}}_k)$ is $\tilde{\rho}_k$-strongly convex on $\mathcal{B}_{\tilde{\beta}_k, r_k}$.

Because the penalty part

$$\frac{1}{2} \sum_{k \in [K]} \sum_{k' \neq k} \Lambda_{kk'} \|\beta_k - \beta_{k'}\|_2$$

is $\sum_{k' \neq k} \Lambda_{kk'}$-Lipschitz in $\beta_k$, we can apply Lemma E.2 to obtain

$$\|\hat{\beta}_k - \tilde{\beta}_k\|_2 \leq \frac{\sum_{k' \neq k} \Lambda_{kk'}}{\tilde{\rho}_k}.$$

Define

$$R_k := \frac{\|\nabla F_k(\beta_k^*, \hat{\boldsymbol{\eta}}_k)\|_2 + \sum_{k' \neq k} \Lambda_{kk'}}{\tilde{\rho}_k}.$$

Combining the two preceding bounds gives

$$\|\hat{\beta}_k - \beta_k^*\|_2 \leq R_k.$$

Moreover, (L.4) implies

$$R_k \leq \frac{2\|\nabla F_k(\beta_k^*, \hat{\boldsymbol{\eta}}_k)\|_2 + \sum_{k' \neq k} \Lambda_{kk'}}{\tilde{\rho}_k} < \delta/2.$$

Therefore

$$\hat{\beta}_k \in \text{int} \, \mathcal{B}_{\beta_k^*, \delta/2}.$$

Since

$$\|\beta_k^* - \beta_{k'}^*\|_2 \geq \delta, \qquad k \neq k',$$

we have

$$\text{int}\,\mathcal{B}_{\beta_k^*,\delta/2} \cap \text{int}\,\mathcal{B}_{\beta_{k'}^*,\delta/2} = \varnothing.$$

Hence $\hat{\beta}_k \neq \hat{\beta}_{k'}$ for all $k \neq k'$. $\qquad\square$

**Lemma L.3.** *Assume events $\mathcal{E}_{1n}$ and $\mathcal{E}_{2n}$ hold and let $\{\lambda_{jj'}\}_{j\neq j'\in[m]}$ satisfy the property in Lemma L.1. If for any $k \in [K]$ and any index $j_k \in S_k$ with $|S_k| > 1$, it holds that*

$$\frac{2\,\|\nabla F_k(\beta_k^*,\hat{\boldsymbol{\eta}}_k)\|_2 + \sum_{k'\neq k,k'\in[K]} \Lambda_{kk'}}{\sum_{j\in S_k}\rho_j} < \min\left\{\frac{M}{2}, \frac{\delta}{2}, \frac{\lambda_{j_k,j} - \|\nabla f_j(\beta_k^*,\hat{\eta}_j)\|_2}{\kappa_j}\right\}, \quad \forall j \in S_k \setminus \{j_k\}, \qquad \text{(L.5)}$$

*then the following hold:*

$$\hat{\beta}_k \neq \hat{\beta}_{k'} \quad \forall k, k' \in [K], k \neq k';$$
$$\hat{\theta}_j = \hat{\beta}_k, \quad j \in S_k\,.$$

*Proof.* By (L.2) we know $\xi_k \leq M/2$, and thus a sufficient condition for (L.4) is

$$\frac{2\,\|\nabla F_k(\beta_k^*,\hat{\boldsymbol{\eta}}_k)\|_2 + \sum_{k'\neq k}\Lambda_{kk'}}{\sum_{j\in S_k}\rho_j} < \min\{M/2, \delta/2\}, \qquad \forall k \in [K].$$

Everything else follows exactly the same procedure as the proof of Lemma B.5. $\qquad\square$

**Lemma L.4.** *Let events $\mathcal{E}_{1n}$ and $\mathcal{E}_{2n}$ hold. Assume $\xi_{\max} \leq c_\xi n_{\min}^{-1/2}$ and $n_{\min}^{\alpha\gamma}/n_{\max} > c_0$ with $c_0 = c_w^{-1}\{(4c_\xi) \vee (\delta)\}^\gamma (\rho/8 + 3\kappa/2)\{(M/2) \wedge (\delta/2)\}$. If $\varepsilon_n$ is set to be $\varepsilon_n < \min_{k\in[K]} N_k^\zeta \{(M/2) \wedge (\delta/2)\}m^{-2}\rho/4$ for $\zeta < 1/2$, then for $\tau \in \left(c_w\{\frac{\delta}{2}\}^{-\gamma}, c_w\{\frac{\delta}{2}n_{\min}^{-\alpha} + 2\xi_{\max}\}^{-\gamma}\right)$, the following hold*

$$\hat{\beta}_k \neq \hat{\beta}_{k'} \quad \forall k, k' \in [K], k \neq k';$$
$$\hat{\theta}_j = \hat{\beta}_k, \quad j \in S_k;$$
$$\left\|\tilde{\beta}_k - \beta_k^*\right\|_2 \leq \frac{1}{8}\{M \wedge \delta\}, \quad \forall k \in [K];$$
$$\left\|\hat{\beta}_k - \tilde{\beta}_k\right\|_2 < \tilde{C}N_k^{\zeta-1} \quad \forall k \in [K],$$

*where $\tilde{C} > 0$ is a constant.*

*Proof.* Recall by definition of $\mathcal{E}_{1n}$ that $\rho_j = \rho n_j/2$, $\kappa_j = 3\kappa n_j/2$, hence $\sum_{j\in S_k}\rho_j = N_k\rho/2$. By Lemma L.3 and definition of $\mathcal{E}_{1n}$, it suffices to show the following statement: for any $k \in [K]$ and any index $j_k \in S_k$, it holds that

$$\frac{2\,\left\|\sum_{j\in S_k}\nabla f_j(\beta_k^*,\hat{\eta}_j)\right\|_2 + \sum_{k'\neq k,k'\in[K]}\Lambda_{kk'}}{N_k\rho/2} < \min\left\{\frac{M}{2}, \frac{\delta}{2}, \frac{\lambda_{j_k,j} - \|\nabla f_j(\beta_k^*,\hat{\eta}_j)\|_2}{3\kappa n_j/2}\right\}, \quad \forall j \in S_k \setminus \{j_k\}. \quad \text{(L.6)}$$

*Firstly consider the RHS of* (L.6). By Lemma L.1 and definition of $\mathcal{E}_{1n}$, we have $\lambda_{j_k,j} > c_w\{\frac{\delta}{2}n_{\min}^{-\alpha} + 2\xi_{\max}\}^{-\gamma}$ and $\|\nabla f_j(\beta_k^*,\hat{\eta}_j)\|_2/n_j \leq \frac{\rho}{8}\{(M/2) \wedge (\delta/2)\}$. Therefore

$$\frac{\lambda_{j_k,j} - \|\nabla f_j(\beta_k^*,\hat{\eta}_j)\|_2}{3\kappa n_j/2} \geq \frac{2}{3\kappa}\left(c_w\left\{\frac{\delta}{2}n_{\min}^{-\alpha} + 2\xi_{\max}\right\}^{-\gamma}/n_j - \frac{\rho}{8}\{(M/2) \wedge (\delta/2)\}\right)$$

$$\geq \frac{2}{3\kappa}\left(\frac{c_w}{n_{\max}}\left[\delta^{-\gamma}n_{\min}^{\alpha\gamma} \wedge \{4\xi_{\max}\}^{-\gamma}\right] - \frac{\rho}{8}\{(M/2) \wedge (\delta/2)\}\right).$$

Using the inequality that

$$\xi_{\max} \leq c_\xi n_{\min}^{-1/2}$$

the following lower bound holds:

$$\geq \frac{2}{3\kappa} \left( \frac{c_w}{n_{\max}} \left[ \delta^{-\gamma} n_{\min}^{\alpha\gamma} \wedge (4c_\xi)^{-\gamma} n_{\min}^{\gamma/2} \right] - \frac{\rho}{8} \{(M/2) \wedge (\delta/2)\} \right).$$

If

$$n_{\min}^{\alpha\gamma}/n_{\max} > c_1, \quad c_1 = c_w^{-1}(\delta)^\gamma (\rho/8 + 3\kappa/2)\{(M/2) \wedge (\delta/2)\}$$
$$n_{\min}^{\gamma/2}/n_{\max} > c_2, \quad c_2 = c_w^{-1}(4c_\xi)^\gamma (\rho/8 + 3\kappa/2)\{(M/2) \wedge (\delta/2)\}$$

(L.7)

then it follows that

$$\frac{\lambda_{j_k,j} - \|\nabla f_j(\beta_k^*, \hat{\eta}_j)\|_2}{3\kappa n_j/2} > (M/2) \wedge (\delta/2).$$

Therefore we have shown that RHS $= (M/2) \wedge (\delta/2)$ under the given condition. Lastly, we find a sufficient condition for (L.7):

$$n_{\min}^{\gamma(\alpha \wedge 1/2)}/n_{\max} \geq c_1 \vee c_2 = c_w^{-1}\{(4c_\xi) \vee (\delta)\}^\gamma (\rho/8 + 3\kappa/2)\{(M/2) \wedge (\delta/2)\} := c_0.$$

Assumption 3.3 states $\alpha \leq 1/2$, and thus this is equivalent to

$$n_{\min}^{\alpha\gamma}/n_{\max} \geq c_0.$$

*Secondly consider LHS of* (L.6). By the condition $\varepsilon_n < \min_{k \in [K]} N_k^\zeta \{(M/2) \wedge (\delta/2)\} m^{-2} \rho/4$, it follows from Lemma L.1 that

$$\frac{\sum_{k' \neq k, k' \in [K]} \Lambda_{kk'}}{N_k \rho/2} \leq \frac{m^2 \varepsilon_n}{N_k \rho/2} < \frac{N_k^{\zeta-1}}{2} \{(M/2) \wedge (\delta/2)\} \leq \frac{1}{2} \{(M/2) \wedge (\delta/2)\}.$$

Also, definition of $\mathcal{E}_{1n}$ implies

$$\frac{2\|\nabla F_k(\beta_k^*, \hat{\eta}_k)\|_2}{N_k \rho/2} \leq \frac{4}{\rho} \frac{\|\nabla F_k(\beta_k^*, \hat{\eta}_k)\|_2}{N_k} \leq \frac{1}{2} \{(M/2) \wedge (\delta/2)\}$$

and thus LHS of (L.6) $< (M/2) \wedge (\delta/2) =$ RHS of (L.6). Now we have shown LHS $<$ RHS, and we can apply Lemma L.2 and definition of $\mathcal{E}_{1n}$, and conclude that

$$\left\| \tilde{\beta}_k - \beta_k^* \right\|_2 \leq \frac{\|\nabla F_k(\beta_k^*, \hat{\eta}_k)\|_2}{N_k \rho/2} \leq \frac{1}{4} \{(M/2) \wedge (\delta/2)\} \quad \forall k \in [K];$$

$$\left\| \hat{\beta}_k - \tilde{\beta}_k \right\|_2 \leq \frac{\sum_{k' \neq k, k' \in [K]} \Lambda_{kk'}}{N_k \rho/2} < \frac{N_k^{\zeta-1}}{2} \{(M/2) \wedge (\delta/2)\} \quad \forall k \in [K].$$

$\square$

**Lemma L.5.** *Under the same conditions as in Lemma L.4, the following bound holds*

$$\|\tilde{\beta}_k - \beta_k^*\|_2 \leq C \left\{ K^{1/r_1} + \sqrt{K} \max_{j \in S_k} s(n_j)^{-1} + K m_k^{1/2} \max_{j \in S_k} n_j^{1/2} s(n_j)^{-2} \right\} N_k^{-1/2} t + C\xi_k,$$

*where $C > 0$ is a constant.*

*Proof.* The first order condition gives

$$\nabla F_k(\tilde{\beta}_k, \hat{\boldsymbol{\eta}}_k) = 0.$$

A first order Taylor expansion around $\beta_k^*$ yields

$$\sqrt{N_k}(\tilde{\beta}_k - \beta_k^*) = -\left\{N_k^{-1}\nabla^2 F_k(\bar{\beta}_k, \hat{\boldsymbol{\eta}}_k)\right\}^{-1}\left\{N_k^{-1/2}\nabla F_k(\beta_k^*, \hat{\boldsymbol{\eta}}_k)\right\},$$

for some $\bar{\beta}_k$ on the segment between $\beta_k^*$ and $\tilde{\beta}_k$.

Consider the Hessian part. Results from Lemma L.4 implies $\|\tilde{\beta}_k - \beta_k^*\|_2 \leq M/2$ and by Equation (L.2) $\|\theta_j^* - \beta_k^*\|_2 \leq M/2$, and thus

$$\|\bar{\beta}_k - \theta_j^*\|_2 \leq \|\bar{\beta}_k - \beta_k^*\|_2 + \|\beta_k^* - \theta_j^*\|_2 \leq M.$$

Thus by definition of $\mathcal{E}_{1n}$ on lower bounded Hessian inside this ball, we have

$$N_k^{-1}\nabla^2 F_k(\bar{\beta}_k, \hat{\boldsymbol{\eta}}_k) \succeq \frac{\rho}{2}I.$$

Consider the score part.

$$N_k^{-1/2}\nabla F_k(\beta_k^*, \hat{\boldsymbol{\eta}}_k) = N_k^{-1/2}\sum_{j \in S_k}\nabla f_j(\theta_j^*, \hat{\eta}_j) + N_k^{-1/2}\sum_{j \in S_k}\left(\nabla f_j(\beta_k^*, \hat{\eta}_j) - \nabla f_j(\theta_j^*, \hat{\eta}_j)\right)$$

$$:= T_1 + T_2.$$

Bounding the first term $T_1$ has been shown in the proof of Lemma B.7:

$$\|T_1\|_2 \leq C_1 K^{1/r_1}t + C_2\sqrt{K}\max_{j \in S_k}s(n_j)^{-1}t + C_3 K m_k^{1/2}\max_{j \in S_k}n_j^{1/2}s(n_j)^{-2}t.$$

As for second term $T_2$, with a first-order expansion, for some $\bar{\theta}_{jk}^*$ between $\beta_k^*$ and $\theta_j^*$,

$$T_2 = N_k^{-1/2}\sum_{j \in S_k}\nabla^2 f_j(\bar{\theta}_{jk}^*, \hat{\eta}_j)(\beta_k^* - \theta_j^*).$$

By (L.1), $\|\beta_k^* - \theta_j^*\|_2 \leq \xi_k \leq M$, so by event $\mathcal{E}_{1n}$ on the upper bounded Hessian inside this ball

$$\|T_2\|_2 \leq N_k^{1/2}\left\|N_k^{-1}\sum_{j \in S_k}\nabla^2 f_j(\bar{\theta}_{jk}^*, \hat{\eta}_j)\right\|_2\xi_k \leq \frac{3\kappa}{2}N_k^{1/2}\xi_k. \tag{L.8}$$

Hence, we have for some constant $C > 0$

$$\sqrt{N_k}\|\tilde{\beta}_k - \beta_k^*\|_2 \leq \frac{2}{\rho}\left\{C_1 K^{1/r_1}t + C_2\sqrt{K}\max_{j \in S_k}s(n_j)^{-1}t + C_3 K m_k^{1/2}\max_{j \in S_k}n_j^{1/2}s(n_j)^{-2}t + \frac{3\kappa}{2}N_k^{1/2}\xi_k\right\}$$

$$\leq C\left\{K^{1/r_1} + \sqrt{K}\max_{j \in S_k}s(n_j)^{-1} + K m_k^{1/2}\max_{j \in S_k}n_j^{1/2}s(n_j)^{-2}\right\}\cdot t + CN_k^{1/2}\xi_k.$$

$\square$

### L.3. High-probability bounds for events $\mathcal{E}_{1n}$ and $\mathcal{E}_{2n}$

In this section, we show that the regularity events $\mathcal{E}_{1n}$ and $\mathcal{E}_{2n}$, defined in Definitions B.1 and B.2, satisfy $\mathbb{P}(\mathcal{E}_{1n} \cap \mathcal{E}_{2n}) \to 1$. As stated in deterministic analysis, the event $\mathcal{E}_{1n}$ has been slightly modified according to Equation (L.3). Compared to the clean model, the perturbed model in (L.1) allows $\theta_j^* \neq \beta_k^*$ for $j \in S_k$. As a result, we only need to replace Lemma C.6 with Lemma L.6, which is the only step that relies on the condition $\theta_j^* = \beta_k^*$.

**Lemma L.6.** *Under Assumptions 3.1 - 3.2, if $1 < t < c_2 m^{-1}\left\{n_{\min}^{\frac{r_2}{2(r_2+d)}} \wedge s(n_{\min})\right\}$, then the event $\mathcal{E}_{1n}$ hold with probability at least $1 - t^{-1}$.*

*Proof.* The first part of the proof follows the same argument as in Lemma C.6. We therefore start from the intermediate step in (C.24), which states that, with probability at least $1 - t^{-1}$, the following inequalities hold simultaneously:

$$\frac{1}{n_j} \left\| \nabla f_j(\theta_j^*, \eta_j^*) \right\|_2 \leq \frac{\rho}{32} \{(M/2) \wedge (\delta/2)\} \qquad j \in [m];$$

$$\frac{1}{n_j} \left\| \nabla f_j(\theta_j^*, \hat{\eta}_j) - \nabla f_j(\theta_j^*, \eta_j^*) \right\|_2 \leq \frac{\rho}{32} \{(M/2) \wedge (\delta/2)\} \qquad j \in [m];$$

$$N_k^{-1/2} \left\| \sum_{j \in S_k} D_\eta \nabla f_j(\theta_j^*, \eta_j^*)[\Delta \eta_j] \right\|_2 \leq C_1' \sqrt{K} \max_{j \in S_k} s(n_j)^{-1} t \qquad k \in [K];$$

$$N_k^{-1/2} \left\| \sum_{j \in S_k} D_\eta^2 \nabla f_j(\theta_j^*, \bar{\eta}_j)[\Delta \eta_j, \Delta \eta_j] \right\|_2 \leq C_1' K \, m_k^{1/2} \max_{j \in S_k} n_j^{1/2} s(n_j)^{-2} t \qquad k \in [K];$$

$$N_k^{-1/2} \left\| \sum_{j \in S_k} \nabla f_j(\theta_j^*, \eta_j^*) \right\|_2 \leq C_1' K^{1/r_1} t \qquad k \in [K];$$

$$\sup_{\theta \in \mathcal{B}_{\theta_j^*, M}} \frac{1}{n_j} \left\| \nabla^2 f_j(\theta, \eta_j^*) - \mathbb{E} \nabla^2 f_j(\theta, \eta_j^*) \right\|_2 \leq \frac{\rho}{4} \qquad j \in [m];$$

$$\sup_{\theta \in \mathcal{B}_{\theta_j^*, M}} \frac{1}{n_j} \left\| \nabla^2 f_j(\theta, \hat{\eta}_j) - \nabla^2 f_j(\theta, \eta_j^*) \right\|_2 \leq \frac{\rho}{4} \qquad j \in [m];$$

$$\left\| \hat{\eta}_j - \eta_j^* \right\|_{\mathcal{H}_j} \leq M \qquad j \in [m],$$

Compared to (C.24) to (C.31), we tighten the upper bounds in gradients from $\frac{\rho}{16} \{M \wedge (\delta/2)\}$ to $\frac{\rho}{32} \{(M/2) \wedge (\delta/2)\}$. This change only affects constants and does not impact the final result. Now by the triangle inequality,

$$\frac{1}{n_j} \left\| \nabla f_j(\theta_j^*, \hat{\eta}_j) \right\|_2 \leq \frac{\rho}{16} \{(M/2) \wedge (\delta/2)\}, \qquad j \in [m], \qquad \text{(L.9)}$$

$$\sup_{\theta \in \mathcal{B}_{\theta_j^*, M}} \frac{1}{n_j} \left\| \nabla^2 f_j(\theta, \hat{\eta}_j) - \mathbb{E} \nabla^2 f_j(\theta, \eta_j^*) \right\|_2 \leq \frac{\rho}{2}, \qquad j \in [m]. \qquad \text{(L.10)}$$

By the geometric condition in Assumptions 3.1(1), the expected Hessian satisfies

$$\rho I \preceq \frac{1}{n_j} \mathbb{E} \nabla^2 f_j(\theta, \eta_j^*) \preceq \kappa I, \qquad \theta \in \mathcal{B}_{\theta_j^*, M}.$$

Combining this with (L.10) yields

$$\frac{\rho}{2} I \preceq \frac{1}{n_j} \nabla^2 f_j(\theta, \hat{\eta}_j) \preceq \left( \kappa + \frac{\rho}{2} \right) I \preceq \frac{3\kappa}{2} I, \qquad \theta \in \mathcal{B}_{\theta_j^*, M}. \qquad \text{(L.11)}$$

Let $j \in S_k$ for some cluster $k$. A first-order expansion of $\frac{1}{n_j} \nabla f_j(\beta_k^*, \hat{\eta}_j)$ around $\theta_j^*$ gives

$$\left\| \frac{1}{n_j} \nabla f_j(\beta_k^*, \hat{\eta}_j) \right\|_2 \leq \left\| \frac{1}{n_j} \nabla f_j(\theta_j^*, \hat{\eta}_j) \right\|_2 + \left\| \frac{1}{n_j} \nabla^2 f_j(\bar{\theta}_j^*, \hat{\eta}_j) \right\|_2 \|\beta_k^* - \theta_j^*\|_2,$$

where $\bar{\theta}_j^*$ lies on the line segment between $\beta_k^*$ and $\theta_j^*$.

By (L.1), $\|\beta_k^* - \theta_j^*\|_2 \leq \xi_k \leq M$, and thus by (L.11),

$$\left\| \frac{1}{n_j} \nabla^2 f_j(\bar{\theta}_j^*, \hat{\eta}_j) \right\|_2 \|\beta_k^* - \theta_j^*\|_2 \leq \frac{3\kappa}{2} \xi_k.$$

Combining this with (L.9), we obtain

$$\left\| \frac{1}{n_j} \nabla f_j(\beta_k^*, \hat{\eta}_j) \right\|_2 \leq \frac{\rho}{16} \{(M/2) \wedge (\delta/2)\} + \frac{3\kappa}{2} \xi_k.$$

Moreover, by the bound in (L.2), $\frac{3\kappa}{2} \xi_k \leq \frac{\rho}{16} \{(M/2) \wedge (\delta/2)\}$. Therefore, for some constant $C_0' > 0$,

$$\left\| \frac{1}{n_j} \nabla f_j(\beta_k^*, \hat{\eta}_j) \right\|_2 \leq \frac{\rho}{8} \{(M/2) \wedge (\delta/2)\}.$$

Finally, define weights $w_j^{(k)} = n_j/N_k$, it follows that

$$\frac{1}{N_k} \|\nabla F_k(\beta_k^*, \hat{\boldsymbol{\eta}}_k)\|_2 = \frac{1}{N_k} \left\| \sum_{j \in S_k} \nabla f_j(\beta_k^*, \hat{\eta}_j) \right\|_2 \leq \sum_{j \in S_k} w_j^{(k)} \frac{1}{n_j} \|\nabla f_j(\beta_k^*, \hat{\eta}_j)\|_2 \leq \frac{\rho}{8} \{(M/2) \wedge (\delta/2)\}.$$

$\square$

## L.4. Proof of main results

In this section, we first establish Theorem L.7, which serves as the key intermediate result. Building on this, we then derive Theorems 3.7 and 3.8 as direct consequences.

**Theorem L.7.** *Under Assumptions 3.1 - 3.3, if i) $n_{\min}^{\alpha\gamma}/n_{\max} > c_0$, ii) $\varepsilon_n < \min_{k \in [K]} N_k^\zeta \{(M/2) \wedge (\delta/2)\} m^{-2} \rho/4$ for $\zeta < 1/2$, iii) $1 < t < c_2 m^{-1} \{n_{\min}^{\frac{r_2}{2(r_2+d)}} \wedge s(n_{\min})\}$, iv) $\tau \in \left( c_w \{\frac{\delta}{2}\}^{-\gamma}, c_w \{\frac{\delta}{2} n_{\min}^{-\alpha} + 2\xi_{\max}\}^{-\gamma} \right)$, v) $\xi_{\max} \leq c_\xi n_{\min}^{-1/2}$, then it holds with probability at least $1 - t^{-1} - p_{\delta/4}(n_{\min})$ that*

$$\hat{\beta}_k \neq \hat{\beta}_{k'} \quad \forall k, k' \in [K], k \neq k';$$
$$\hat{\theta}_j = \hat{\beta}_k, \quad j \in S_k;$$
$$\left\| \tilde{\beta}_k - \beta_k^* \right\|_2 \leq C \left\{ K^{1/r_1} + \sqrt{K} \max_{j \in S_k} s(n_j)^{-1} + K m_k^{1/2} \max_{j \in S_k} n_j^{1/2} s(n_j)^{-2} \right\} N_k^{-1/2} t + C\xi_k, \quad \forall k \in [K]; \qquad \text{(L.12)}$$
$$\left\| \hat{\beta}_k - \tilde{\beta}_k \right\|_2 < \tilde{C} N_k^{\zeta-1} \quad \forall k \in [K].$$

*where $c_0, c_2, C, \tilde{C} > 0$ are some constants.*

*Proof.* This theorem is built upon Lemmas L.4, L.5 and L.6. From Lemmas L.4 and L.5, we know that the inequality system (L.12) holds as long as $\mathcal{E}_{1n} \cap \mathcal{E}_{2n}$ holds. In terms of probability, this means

$$\mathbb{P}\big((\text{L.12}) \text{ holds}\big) \geq \mathbb{P}\big(\mathcal{E}_{1n} \cap \mathcal{E}_{2n}\big).$$

Now we only need to find the probability lower bound on the event $\mathcal{E}_{1n} \cap \mathcal{E}_{2n}$ to prove the Theorem. From Lemma C.1, we get

$$\mathbb{P}\left(\mathcal{E}_{2n}\right) \geq 1 - p_{\delta/4}(n_{\min}),$$

while it follows from Lemma L.6 that

$$\mathbb{P}\left(\mathcal{E}_{1n}\right) \geq 1 - t^{-1}.$$

Thus, the joint probability is such that

$$\mathbb{P}(\mathcal{E}_{1n} \cap \mathcal{E}_{2n}) \geq \mathbb{P}\left(\mathcal{E}_{1n}\right) + \mathbb{P}\left(\mathcal{E}_{2n}\right) - 1 \geq 1 - t^{-1} - p_{\delta/4}(n_{\min}).$$

$\square$

L.4.1. PROOF OF THEOREM 3.7

Our proof of Theorem 3.7 is a direct application of Theorem L.7. Firstly, the first part of Theorem L.7 says

$$\hat{\beta}_k \neq \hat{\beta}_{k'} \quad \forall k, k' \in [K], k \neq k';$$
$$\hat{\theta}_j = \hat{\beta}_k, \quad j \in S_k.$$

This implies that $\hat{\theta}_j = \hat{\theta}_{j'}$ for all $j, j' \in S_k$ and $\hat{\theta}_j \neq \hat{\theta}_{j'}$ for $j \in S_k$, $j' \in S_{k'}$, $k \neq k'$.

Secondly, the second part of Theorem L.7

$$\left\| \tilde{\beta}_k - \beta_k^* \right\|_2 \leq C \left\{ K^{1/r_1} + \sqrt{K} \max_{j \in S_k} s(n_j)^{-1} + K m_k^{1/2} \max_{j \in S_k} n_j^{1/2} s(n_j)^{-2} \right\} N_k^{-1/2} t + C \xi_k, \quad \forall k \in [K];$$
$$\left\| \hat{\beta}_k - \tilde{\beta}_k \right\|_2 < \tilde{C} N_k^{\zeta-1} \quad \forall k \in [K]$$

and by within-cluster heterogeneity condition in (L.1),

$$\| \beta_k^* - \theta_j^* \| \leq \xi_k \,.$$

The above three inequalities imply

$$\|\hat{\theta}_j - \theta_j^*\|_2 \leq \|\hat{\theta}_j - \tilde{\beta}_k\|_2 + \|\tilde{\beta}_k - \beta_k^*\|_2 + \|\beta_k^* - \theta_j^*\|_2$$
$$\leq C \left\{ K^{1/r_1} + \sqrt{K} \max_{j \in S_k} s(n_j)^{-1} + K m_k^{1/2} \max_{j \in S_k} n_j^{1/2} s(n_j)^{-2} \right\} N_k^{-1/2} t + \tilde{C} N_k^{\zeta-1} + (C+1)\xi_k$$

by triangle inequality. Since the rate function $s(n_j) \lesssim n_j^{1/2}$ (defined in Assumption 3.2), we have

$$\frac{n_j^{1/2} s(n_j)^{-2}}{s(n_j)^{-1}} = n_j^{1/2} s(n_j)^{-1} \gtrsim 1 \,.$$

Therefore, the sum of second and third term is upper bounded by

$$K^{1/2} \max_{j \in S_k} s(n_j)^{-1} + K m_k^{1/2} \max_{j \in S_k} n_j^{1/2} s(n_j)^{-2} \lesssim K m_k^{1/2} \max_{j \in S_k} n_j^{1/2} s(n_j)^{-2} = b_{k,n} \,.$$

This completes the proof.

L.4.2. PROOF OF THEOREM 3.8

Fixing $k$ and any $j \in S_k$, the following decomposition holds:

$$\hat{\theta}_j - \theta_j^* = (\hat{\theta}_j - \tilde{\beta}_k) + (\tilde{\beta}_k - \beta_k^*) + (\beta_k^* - \theta_j^*). \tag{L.13}$$

*Firstly*, it follows from Theorem L.7 that

$$\hat{\theta}_j - \tilde{\beta}_k = O_p(N_k^{\zeta-1}), \qquad \zeta < 1/2 \,.$$

*Secondly*, to study $\tilde{\beta}_k - \beta_k^*$, a first order condition gives

$$\nabla F_k(\tilde{\beta}_k, \hat{\boldsymbol{\eta}}_k) = 0 \,.$$

A Taylor expansion around $\beta_k^*$ yields

$$\sqrt{N_k}(\tilde{\beta}_k - \beta_k^*) = -\left\{ N_k^{-1} \nabla^2 F_k(\bar{\beta}_k, \hat{\boldsymbol{\eta}}_k) \right\}^{-1} \left\{ N_k^{-1/2} \nabla F_k(\beta_k^*, \hat{\boldsymbol{\eta}}_k) \right\},$$

for some $\bar{\beta}_k$ on the segment between $\beta_k^*$ and $\tilde{\beta}_k$.

Decompose

$$N_k^{-1} \nabla^2 F_k(\bar{\beta}_k, \hat{\boldsymbol{\eta}}_k) = B_0 + B_1 + B_2,$$

$$B_0 := N_k^{-1} \sum_{j \in S_k} \nabla^2 f_j(\theta_j^*, \eta_j^*),$$

$$B_1 := N_k^{-1} \sum_{j \in S_k} \left\{ \nabla^2 f_j(\bar{\beta}_k, \hat{\eta}_j) - \nabla^2 f_j(\theta_j^*, \hat{\eta}_j) \right\},$$

$$B_2 := N_k^{-1} \sum_{j \in S_k} \left\{ \nabla^2 f_j(\theta_j^*, \hat{\eta}_j) - \nabla^2 f_j(\theta_j^*, \eta_j^*) \right\}.$$

By the Law of Large Number and Assumption 3.1(1),

$$B_0 \to_p \tilde{\Psi}_k.$$

As for $B_1$, we first show

$$\|\bar{\beta}_k - \theta_j^*\|_2 \le \|\bar{\beta}_k - \beta_k^*\|_2 + \|\beta_k^* - \theta_j^*\|_2 = o_p(1)$$

since $\|\bar{\beta}_k - \beta_k^*\| = o_p(1)$ by consistency of $\tilde{\beta}_k$ in Lemma L.7 and $\|\beta_k^* - \theta_j^*\|_2 = \xi_k = o(1)$. Now with definition

$$\Xi_j(\theta, \theta', \eta, \eta', z) = \frac{\|\nabla^2 \ell_j(\theta, \eta, z) - \nabla^2 \ell_j(\theta', \eta', z)\|_2}{\|\theta - \theta'\|_2 + \|\eta - \eta'\|_{\mathcal{H}_j}}$$

by the triangle inequality and the local Lipschitz control in Assumption 3.1(4),

$$\|B_1\|_2 \le N_k^{-1} \sum_{j \in S_k} \sum_{i=1}^{n_j} \Xi_j(\bar{\beta}_k, \theta_j^*, \hat{\eta}_j, \hat{\eta}_j, Z_{ji}) \|\bar{\beta}_k - \theta_j^*\|_2 = O_p(1) \cdot o_p(1) = o_p(1),$$

For $B_2$, similarly, by Assumption 3.2 on consistency of $\hat{\eta}_j$,

$$\|B_2\|_2 \le N_k^{-1} \sum_{j \in S_k} \sum_{i=1}^{n_j} \Xi_j(\theta_j^*, \theta_j^*, \hat{\eta}_j, \eta_j^*, Z_{ji}) \|\hat{\eta}_j - \eta_j^*\|_{\mathcal{H}_j} = O_p(1) \cdot o_p(1) = o_p(1).$$

Therefore,

$$N_k^{-1} \nabla^2 F_k(\bar{\beta}_k, \hat{\boldsymbol{\eta}}_k) \to_p \tilde{\Psi}_k.$$

Consider the score part.

$$N_k^{-1/2} \nabla F_k(\beta_k^*, \hat{\boldsymbol{\eta}}_k) = N_k^{-1/2} \sum_{j \in S_k} \nabla f_j(\theta_j^*, \hat{\eta}_j) + N_k^{-1/2} \sum_{j \in S_k} \left( \nabla f_j(\beta_k^*, \hat{\eta}_j) - \nabla f_j(\theta_j^*, \hat{\eta}_j) \right)$$

$$:= T_1 + T_2.$$

$$T_1 = N_k^{-1/2} \sum_{j \in S_k} \left\{ \nabla f_j(\theta_j^*, \eta_j^*) + D_\eta \nabla f_j(\theta_j^*, \eta_j^*)[\Delta \eta_j] + \tfrac{1}{2} D_\eta^2 \nabla f_j(\theta_j^*, \bar{\eta}_j)[\Delta \eta_j, \Delta \eta_j] \right\}$$

$$:= S_k^0 + R_{k1} + R_{k2}.$$

A direct application of CLT yields that

$$S_k^0 \implies \mathcal{N}(0, \tilde{\Omega}_k).$$

Moreover, it is shown in Lemma L.6 that

$$\|R_{k1}\|_2 = O_p(\sqrt{K} \max_{j \in S_k} s(n_j)^{-1})$$

$$\|R_{k2}\|_2 = O_p(K m_k^{1/2} \max_{j \in S_k} n_j^{1/2} s(n_j)^{-2})$$

so both $R_{k1}$ and $R_{k2}$ are of order $O_p(b_{k,n})$ due to the condition $s(n) \lesssim n^{1/2}$. Thus the condition $b_{k,n} = o(1)$ implies

$$T_1 = S_k^0 + R_{k1} + R_{k2} \implies \mathcal{N}(0, \tilde{\Omega}_k).$$

Moreover, it has been shown in Equation (L.8) that

$$\|T_2\|_2 = O_p(N_k^{1/2}\xi_k).$$

Thus the condition $\xi_k = o(N_k^{-1/2})$ gives

$$\|T_2\|_2 = o_p(1).$$

Combining this the limit of $T_1$ yields

$$N_k^{-1/2}\nabla F_k(\beta_k^*, \hat{\boldsymbol{\eta}}_k) = T_1 + T_2 \implies \mathcal{N}(0, \tilde{\Omega}_k).$$

This, combined with Hessian limit, yields

$$\sqrt{N_k}(\tilde{\beta}_k - \beta_k^*) \implies \mathcal{N}(0, \tilde{\Psi}_k^{-1}\tilde{\Omega}_k\tilde{\Psi}_k^{-1})$$

*Thirdly*, with the closeness condition between $\beta_k^*$ and $\theta_j^*$ in this theorem, we have

$$\|\beta_k^* - \theta_j^*\|_2 \leq \xi_k = o\left(N_k^{-1/2}\right).$$

Combining the three limits above, and going back to Equation (L.13), we conclude that

$$\sqrt{N_k}(\hat{\theta}_j - \theta_j^*) \Rightarrow \mathcal{N}(0, \tilde{\Psi}_k^{-1}\tilde{\Omega}_k\tilde{\Psi}_k^{-1}).$$

## M. Additional experiments on real data

Using the same RECS data and partially linear modeling framework as in Section 5, we further provide a quantitative comparison based on held-out orthogonal squared loss. We compare the proposed method with two baselines: a personalized estimator and clustered ARMUL. Since ARMUL is designed for parametric models and does not directly accommodate task-specific nuisance heterogeneity, we implement ARMUL using a full linear model rather than a partially linear specification. The results for the RECS data are reported in Table 3. For the personalized and adaptive estimators, the nuisance functions are estimated using neural networks. The proposed adaptive fusion method achieves competitive held-out performance, suggesting that it can exploit cluster structure while preserving task-specific heterogeneity. In this dataset, forced clustering by ARMUL appears to over-aggregate the treatment effects, which hurts out-of-sample prediction. By contrast, the proposed method adaptively identifies heterogeneity across tasks and achieves performance close to personalized, both in terms of mean orthogonal loss and its standard deviation.

| Method | Orthogonal loss | SD |
|---|---|---|
| **Per** | **0.818051** | **0.016061** |
| **Ada** | **0.823930** | **0.017656** |
| **ARMUL(K=2)** | 0.878603 | 0.035594 |
| **ARMUL(K=3)** | 0.913522 | 0.067865 |
| **ARMUL(K=4)** | 0.962894 | 0.083113 |

*Table 3.* Comparison of methods in terms of test orthogonal squared loss on the RECS data. The test set contains 10% of the observations within each task, and the reported standard deviation is computed across 10 random splits.

