# OpenReview forum: "Adaptive Estimation and Inference in Semi-parametric Heterogeneous Clustered Multitask Learning via Neyman Orthogonality"
_ICML.cc/2026/Conference — ICML 2026 regular_

### Official Review · Reviewer_Q1Kb · 2026-03-09

**Soundness:** 3
**Presentation:** 3
**Significance:** 4
**Originality:** 3
**Overall Recommendation:** 4
**Confidence:** 3

**Summary:**

This paper proposed a two stage Neyman Orthogonality Estimator for multitask learning. The proposed method can effectively eliminate the impact of nuisance parameters. Then the theoretical analysis showed the proposed method can establishes exact clustering and each task-level estimator attains the oracle rate. And sufficient simulation and real-world experiments show that the proposed method is effective.

**Compliance With Llm Reviewing Policy:**

Affirmed.

**Final Justification:**

The rebuttal addressed my main concerns and problems.

**Key Questions For Authors:**

1. Note that in Eq.(2.1), the initial parameter \theta^{init} is based on the \hat{\eta} and l^{init}. The author claimed that l^{init} is a possibly non-orthogonal loss. So how to set the loss function and how is the impact of different loss functions?

2. Moreover, how to estimate \hat{\eta} seems unknown in the submission (maybe I do not find).

3. Why is the first step meaning similarity estimation between tasks? \hat{\eta} and l^{init} may influence the results, I am curious about how the initial parameter \theta^{init} will influence the penalty parameters, then  influence the results.

**Limitations:**

yes

**Strengths And Weaknesses:**

Soundness: The method is soundness. First it used Neyman Orthogonality loss function to eliminate the impact of nuisance parameters. Then the penalty parameters are set by the estimation of similarity between different tasks. Then theoretical analysis and experiments are shown to validate the effectiveness of the proposed method.

Presentation: The submission is easy to follow. I understand the core idea and problem it would like to handle. However, some details in the proposed algorithm can be enhanced.

Significance: It addresses an important problem in multitask learning: potentially inffnite-dimensional nuisance components in tasks.

Originality: It attempts to use Neyman-Orthogonal Loss Function in multitask learning. The novelty is clear.

---

> ### Author Rebuttal · Authors · 2026-03-31
>
> **Impact of different loss functions** Thank you for this helpful comment. The role of $\ell_j^{\rm init}$ in Eq. (2.1) is only to produce a preliminary task-specific estimator $\hat \theta_j^{\rm init}$ for constructing the adaptive fusion weights; it is not used for final inference. Hence, $\ell_j^{\rm init}$ need not be Neyman-orthogonal. In practice, one may use any standard loss that yields a consistent pilot estimator (e.g., squared-error loss or negative log-likelihood). For example, under the partially linear model $Y_{ij} = X_{ij}^\top \theta_{0j} + f_j(Z_{ij}) + \epsilon_{ij}$, a pilot is obtained by $$(\hat \theta_j^{\rm init}, \hat f_j^{\rm init}) = \arg\min_{\theta,f} \sum_{i=1}^{n_j} (Y_{ij} - X_{ij}^\top \theta - f(Z_{ij}))^2.$$ This loss is generally not Neyman-orthogonal, so $\hat \theta_j^{\rm init}$ is consistent but may converge more slowly than $n_j^{-1/2}$ because of the bias from estimating $f_j$. This does not affect Theorems 3.5 and 3.6 directly, since our theory only requires consistency of the initial estimator (Assumption 3.3). However, the choice of $\ell_j^{\rm init}$, through $\hat \theta_j^{\rm init}$, affects the finite-sample probability of exact cluster recovery (Theorem 3.5): a more accurate pilot yields better separation between within-cluster and between-cluster pairs, so the clustering probability approaches one faster. Therefore, in practice, it is preferable to use the best available pilot estimator; if a Neyman-orthogonal loss is available and computationally feasible, it may also be used in Stage 1, but it is not required. We will highlight this in the revised version.
>
> **Estimation of $\hat \eta$.**  Thank you for pointing this out. We apologize that the estimation of the nuisance parameter $\hat \eta$ was not described clearly enough in the main draft and was only briefly mentioned in Algorithm 1. We will make this explicit in the revised manuscript. Our framework allows the nuisance parameters $\eta_j$ to differ across tasks, so we estimate them separately rather than pooling them. For example, in the partially linear model for task $j$, the nuisance components are $E[Y_{ij} \mid Z_{ij} = z] = g_j(z)$ and $E[X_{ij} \mid Z_{ij} = z] = h_j(z)$. We therefore estimate $\eta_j = (g_j,h_j)$ using only task-$j$ data, via any suitable nonparametric regression/classification method, and then use these estimators to form the orthogonal loss for that task. More generally, we leave the estimator of $\hat \eta_j$ flexible, since our theory does not rely on a specific nuisance-estimation algorithm; it only requires the rate condition in Assumption~3.2. Thus, one may use any appropriate method, including standard nonparametric regression, regularized estimation, or modern machine-learning approaches. In summary, the nuisance components are estimated locally within each task, while cross-task sharing occurs only through the target parameters in the second step. We will revise the manuscript to state this more explicitly in the main text.
>
> **Influence of initial estimates on penalties** Thank you for this important comment. As mentioned in our responses above on the choice of $\ell_j^{\rm init}$ and the estimation of $\hat\eta_j$, Stage 1 is not used for final estimation or inference, rather to produce a coarse similarity measure between tasks for calibrating the adaptive fusion penalties in Stage 2 through
> $w\_{jj'} = c\_w ||\hat\theta\_j^{\rm init} - \hat\theta\_{j'}^{\rm init}||\_2^{-\gamma}$, from which $\lambda_{jj'}$ is constructed. Thus, similar pilot estimates lead to stronger fusion. In this sense, Stage 1 is a similarity-screening step that approximates the latent cluster structure using initial estimators under the constructed penalties. The choices of $\ell_j^{\rm init}$ affect the final result only indirectly through $\hat\theta_j^{\rm init}$; they do not enter the final orthogonal loss. This is reflected in Assumption 3.3, which requires only consistency of $\hat\theta_j^{\rm init}$, not Neyman orthogonality or a $\sqrt{n_j}$-rate. Therefore, consistency is enough for asymptotically correct cluster recovery, while a more accurate pilot estimator improves finite-sample guarantees. Regarding nuisance estimation, Theorems 3.5 and 3.6 require $\hat \eta_j$ to converge faster than $n_j^{-1/4}$ (Assumption 3.2). This is the standard condition in Neyman-orthogonal estimation, ensuring that nuisance estimation is asymptotically negligible, and is satisfied for a wide range of parametric/non-parametric estimation.  To summarize, (i) $\hat\theta_j^{\rm init}$ determines the adaptive penalties and, as long as it is consistent, yields asymptotically correct clustering, while a better pilot improves finite-sample performance, as illustrated in our first response; and (ii) nuisance estimation is typically asymptotically negligible because of Neyman orthogonality of the loss function in the second stage. We will highlight this in the revised manuscript.

---

> > ### Author Rebuttal · Reviewer_Q1Kb · 2026-04-03
> >
> > Thank you for your rebuttal. My concerns have been adequately addressed.

---

> > > ### Author Response · Authors · 2026-04-04
> > >
> > > Thank you very much for your thoughtful review and for taking the time to consider our rebuttal. We are glad that our responses have addressed your concerns.
> > >
> > > We appreciate your careful consideration and would be happy to provide any further clarification if helpful.

---

### Official Review · Reviewer_jGYk · 2026-03-13

**Soundness:** 3
**Presentation:** 4
**Significance:** 2
**Originality:** 3
**Overall Recommendation:** 4
**Confidence:** 4

**Summary:**

This paper studies a semiparametric multitask setting in which each task has a low-dimensional target parameter and a heterogeneous, potentially infinite-dimensional nuisance component. The proposed method is a two-stage adaptive fusion procedure: the first stage forms taskwise pilot estimates that are used only to build pairwise fusion weights, and the second stage estimates the target parameters using task-specific Neyman-orthogonal losses with adaptive fusion across tasks. The key claimed contributions are exact recovery of the latent clustering, pooled-rate estimation within clusters, and asymptotic normality matching an oracle that knows the true clusters in advance. The empirical evaluation includes simulations in PLM, ATE, and DID settings, as well as an application to state-level electricity price elasticity using RECS data.

**Compliance With Llm Reviewing Policy:**

Affirmed.

**Final Justification:**

The rebuttal has addressed my primary concerns

**Key Questions For Authors:**

None

**Limitations:**

I do not think the paper adequately discusses limitations and potential negative impacts. The conclusion briefly notes open directions involving the dependence on the number of tasks, cluster separation, and high-dimensional targets, but this is not the same as a serious discussion of the method’s limitations in the regime the paper actually studies. In particular, the paper does not meaningfully discuss approximate clusters, within-cluster heterogeneity, cluster instability, or how sensitive the discovered partition may be to preprocessing choices and hyperparameters in the real-data analysis. Given the strength of the exact-clustering and oracle-inference claims, I think that omission is worth flagging.

The societal-impact statement is also extremely generic. Since the paper is framed partly around policy and causal applications, a more useful limitations discussion would acknowledge that overly aggressive pooling could mask real heterogeneity across environments, which in turn could matter for downstream decisions. I would encourage the authors to discuss this constructively rather than defensively. In this case, being explicit about when the clustering assumption may fail would strengthen the paper rather than weaken it.

**Strengths And Weaknesses:**

**Strengths and Weaknesses**
A major strength of the paper is that it tackles a relevant semiparametric multitask problem in a way that is technically coherent. The motivating setting is clear, the method is conceptually simple, and the choice to fuse only target parameters while keeping nuisance estimation task-specific is well aligned with the inferential goals. The simulation suite covers three standard semiparametric settings, uses flexible nuisance learners, and compares the method to personalized estimation and several ARMUL variants under misspecified and correctly specified cluster counts. The adaptive method performs very well in those experiments, and the fixed-penalty ablation strengthens the empirical case that the adaptive weighting scheme matters.

The weaknesses are mainly about the sharpness and reliability of the theoretical claims, and the empirical example that's given in the paper On the theory side, the exact-cluster assumption is strong, especially because the paper wants oracle-style asymptotic inference. The proof also appears to contain at least two issues that should be resolved before the main theorem is taken at face value. In addition, the proof explicitly uses fixed m, which narrows the scope of the asymptotic result. On the empirical side, the simulations are stronger than the real-data application. The RECS example produces an unusual clustering pattern, but the paper provides little sensitivity analysis, little comparison to simpler alternatives in the application itself, and only a brief substantive justification for why such a large pooled cluster should be believed.

**Soundness**

My hesitation comes from the assumptions and from a few proof-level issues. First, the theory effectively relies on exact equality within clusters, which also happens in other parts of the literature but it exacerbated here by the way the clusters are used. In the simulations, tasks in a cluster literally share a common parameter $\theta_j^\*= \beta_k^\* $, and the theorem claims exact cluster recovery and oracle asymptotic normality after fusion. That assumption is common in clustering theory, so it is not a fatal flaw by itself, but it is strong for the kinds of empirical settings the paper highlights, and it matters more here because the paper makes inference claims rather than only predictive claims. Second, Assumption 3.3 appears misstated relative to how Definition B.2 and Lemma B.3 use it: the assumption is written as a bound on the probability that all pilot estimators are bad simultaneously, whereas the event E2n requires all pilot estimators to be uniformly good. The appendix then seems to pass from Assumption 3.3 to P(E2n)→1 without resolving that quantifier issue. Third, the CLT proof contains a notation inconsistency. Theorem 3.6 defines one matrix as the score covariance and one as the Hessian term, but the proof later appears to swap their roles. I am not claiming the CLT is false, but this kind of issue makes it difficult to adequately assess.

A related concern is the asymptotic regime. The proof of Theorem 3.6 explicitly uses that m is fixed, and the conclusion itself notes that a finer dependence on the number of tasks and on cluster separation remains open. That makes the theoretical guarantees somewhat narrower than the informal framing might suggest. I would have liked a clearer discussion of what happens under approximate clusters, mild within-cluster heterogeneity, or growing numbers of tasks, since those seem central to whether the method is robust in practice.



**Presentation**

The paper is generally readable and well organized. The two-stage method is easy to follow, the proof roadmap in the appendix is helpful, and the simulation section is structured cleanly enough that the main empirical pattern is easy to extract. I also appreciated that the paper includes a fixed-versus-adaptive penalty comparison rather than relying only on headline comparisons to alternative multitask estimators. From a reviewer’s perspective, it is possible to understand what the method is trying to do and why orthogonality is supposed to matter.



**Significance**

The paper addresses an important problem. In many multi-environment causal or semiparametric settings, one would like to pool information across tasks without forcing the nuisance structure to be shared. The paper’s idea of fusing only the low-dimensional targets while leaving nuisance estimation entirely task-specific is appealing, and if the theory were fully convincing it would provide a useful template for combining multitask learning with valid semiparametric inference. That is a real contribution direction, especially given how common heterogeneous nuisance structure is in policy, economics, and causal ML applications.

At the same time, the significance is limited by the gap between the theory and the empirical settings where one would most want to use the method. The theoretical results are strongest under exact latent clusters, fixed or very limited task growth, and clean separation. Those conditions are sufficient for a nice oracle-style story, but they also reduce how much the results tell us about more realistic settings with approximate clusters or continuously varying effects.

The common parameter assumption also plays a role here, particularly when viewed in light of the empirical data analysis.  The resulting partition has one singleton cluster, one four-state cluster, and one 46-state cluster, which is a striking structure and shows how extreme this assumption can be.  In the context of this example it is saying that 46 states (with wildly different political leaningings, climate patterns, natural resource availablity, etc) have exactly the same elasticity.

The authors also essentially passed on writing a societal impact statement.



**Originality**

The paper lives in a very active area and does cite relevant neighboring work, including clustered multitask learning, coordinated double machine learning, and more recent semiparametric multitask references. Against that backdrop, the contribution feels more like a useful synthesis and extension than a new organizing principle. I do not view that as disqualifying, but I also do not think the current version makes a strong case for high originality.

---

> ### Author Rebuttal · Authors · 2026-03-31
>
> **Non-exact clusters and growing $m$.** Thank you for this insightful comment. During the rebuttal period, we extended Theorems 3.5 and 3.6 to a near-homogeneous clustered setting, where target parameters within a cluster need not be exactly identical. Specifically, we assume for each cluster $S_k$,
> $$
> ||\theta_j^\ast - \beta_k^\ast||\_2 \le \xi_k, \qquad j\in S_k,
> $$
> where $\xi_k$ measures within-cluster heterogeneity. This extension also allows the numbers of tasks $m$ and clusters $K$ to grow with sample size. Under Assumptions 3.1-3.3, with the same choice of $(\epsilon_n,\gamma)$ as before, an appropriate choice of $\tau$, and provided $\xi_k \le c_\xi \min_{j \in S_k}n_j^{-1/2}$, we show that: (i) the estimator still achieves exact recovery of the latent clustering, and (ii) for every $j\in S_k$,
> $$
> ||\hat\theta_j - \theta_j^\ast||\_2\le C(1+b_m)K^{1/r_1}N_k^{-1/2}t+\tilde C N_k^{\zeta-1}+2\xi_k,
> $$
> with probability at least $1-t^{-1}-p_{\delta/4}(n_{\min})$, where
> $$
> b_m=K^{1-1/r_1} m_k^{1/2}\max_{j \in S_k} n_j^{1/2} s(n_j)^{-2},
> $$
> $\zeta < 1/2$, $m_k = |S_k|$, and $s(n_j)$ is the nuisance estimation rate and $p_{\delta/4}(n_{\min})$ same as before. Thus, the error is the pooled oracle term plus a bias term from within-cluster heterogeneity. If each cluster contains roughly $m/K$ tasks, each task has sample size $n$, and the nuisance is estimated at rate $n^{-\rho}$, this becomes
> $$
> ||\hat\theta_j - \theta_j^*||_2 \lesssim K^{1/r_1} \sqrt{\frac{1 + K^{1-2/r_1} m n^{1-4\rho}}{mn/K}} + \xi_k.
> $$
> Therefore, our result extends Theorem 4.4 of Duan and Wang to the semiparametric setting with heterogeneous, possibly infinite-dimensional nuisance parameters. Notably, both the dimension and structure of the nuisance may vary from task to task. Moreover, for fixed $K$ and possibly growing $m$, if $\xi_k = o(N_k^{-1/2})$, then the asymptotic normality result of Theorem 3.6 continues to hold. We will revise the manuscript with these updated results.
>
> **Mistakes in Assumption 3.3.** Thank you for pointing this out. We apologize for this typo, as the sign was inadvertently reversed. The correct formulation is
> $$
> \Pr\left(\forall\ j \in [m],\ n\_j^{\alpha}||\hat\theta\_j^{\rm init} - \theta\_j^\ast||_2 < \epsilon\right) \ge 1 - p\_{\epsilon}(n\_{\min}).
> $$
> We have corrected this in the revised manuscript.
>
> **Notational inconsistency in CLT.** Thank you again for pointing this out. We apologize for swapping the notations for the variance of the score and the Hessian. We reread the proof carefully and corrected these typos to the best of our ability. We hope the revised manuscript is now clearer and free of such errors.
>
> **Discussion on limitations and societal impact.** Thank you for this important comment. We agree that the original draft did not discuss the practical limitations and downstream risks of the method as clearly as it should have. We will revise the paper to make these points explicit in both the limitations and societal impact. Although we have extended the theory to a near-homogeneous clustered setting, where target parameters within a cluster satisfy $||\theta_j^\ast-\beta_k^\ast||_2\le \xi_k$, this also highlights an important limitation: if within-cluster heterogeneity is too large, exact clustering and pooled-rate oracle inference may fail; in particular, pooled asymptotic normality requires $\xi_k=o(N_k^{-1/2})$. More generally, cluster recovery can be unstable when cluster separation is weak or when the pilot and nuisance estimators are noisy. In such cases, the discovered partition may be sensitive to preprocessing, nuisance learners, and hyperparameters, and should be viewed as a data-driven summary of similarity rather than definitive evidence of a unique latent grouping. This is especially important in policy and causal applications: if true effects differ meaningfully across environments, overly aggressive pooling may lead to misleading subgroup conclusions or inappropriate policy decisions. Our adaptive fusion helps mitigate this risk relative to naive global pooling, since only the low-dimensional target parameters are pooled, while nuisance components remain task-specific and may differ in both dimension and structure. However, this does not rule out negative transfer when the clustering assumption is only weakly satisfied. We will revise the manuscript to state these limitations more explicitly and to recommend sensitivity analyses before drawing substantive conclusions from the estimated clusters.
>
> **Concern regarding the empirical study.** Thank you for your comment. As detailed in our response to reviewer **bVQb**, we strengthened the empirical study by adding quantitative evaluation via MSE on held-out data, direct comparison with three other baselines, a second real-data study (CBECS), and additional discussion of sensitivity and interpretation, so that the RECS clustering is supported not only by qualitative structure but also by competitive empirical performance.

---

> > ### Author Rebuttal · Reviewer_jGYk · 2026-04-05
> >
> > Thank you for your responses. My concerns have been adequately addressed

---

> > > ### Author Response · Authors · 2026-04-05
> > >
> > > Thank you very much for your thoughtful review and for taking the time to consider our response. Your constructive comments helped improve the paper, and we are glad that our responses have addressed your concerns.
> > >
> > > We sincerely appreciate your careful consideration of our work and would be happy to clarify any remaining points if useful.

---

### Official Review · Reviewer_JS9f · 2026-03-13

**Soundness:** 4
**Presentation:** 4
**Significance:** 3
**Originality:** 3
**Overall Recommendation:** 5
**Confidence:** 2

**Summary:**

The authors make contributions on two fronts. First, the authors extend the double machine learning framework that was previously for single task setting to MTL. Second, they extend the clustered multitask learning to work accommodate for more general and potentially infinite-dimensional task specific nuisance components. Their contributions are MTL that is adaptive: combines Neyman orthoginality with data driven pairwise fusion. Their results recover the latent clusters with high probability independent of knowledge on the cluster sizes or number. They prove their method is Consistent Asymptotic Normal for their estimator at the intercluster pooled rate, thus showing a gain from MTL.


Their algorithm has two stages:
* The first stages goal is to quantify task similarity (for purposes of the fusion calibration in the second step).
This step can be done with conventional losses that are may not be necessarily orthogonal.
* The second stage returns the estimators where the problem has been penalized with the parameters from the first step. Objective is composed with task specific orthogonal loss functions. As the nuisance parameters must be estimated too, so they split the data, and use say the first half of the data for this and solving for parameters in the second step.

They do experiments on synthetic and real data. They instantiate to their method to partially linear model, Average treatment effect, and difference-in-differences and compare their method to standard single task learning and ARMUL. Their algorithm gives the lowest median RMSE.

**Compliance With Llm Reviewing Policy:**

Affirmed.

**Key Questions For Authors:**

1. A minor point, do the authors have a sense on how to set the hyperparameters introduced in their algorithm?
2. Do the authors believe the algorithm extends to all predictors being different but some predictors being within clusters defined by being close?  Similar to the task relatedness given in Duan & Wang. 2023.
3. I was confused about the notion on R.179 with [h, h]. Is this defined somewhere?

**Limitations:**

The authors point to several extensions in the conclusion that are existing limitations.
The biggest limitation is that the intercluster predictors are the same. This isn't true for  (Duan & Wang,
2023), thus when making comparisons to their work in theory and practice this should be pointed out. I was left wondering that perhaps this work would too require cluster count if intercluster predictors where close but different.

**Strengths And Weaknesses:**

**STRENGTHS**
* The work appears strong and well thought out.
* The work extends prior work in meaningful and significant ways.
* The work is novel.
* I enjoyed the US residential electricity demand experiment section.
* Their analysis, in terms of required assumptions, is exceedingly general and the results thus appear to hold widely.
* The assumptions are standard.

**WEAKNESSES**
* A lot of tuning parameters, in particular in the second stage.
* The setup is not what I would expect when I thought clustered tasks. The authors have that cluster differences are given by nuisance components and data distribution but not predictors. This perspective can cause one to think that there are as many tasks as there are clusters by combining their distributions and knowing which points were affected by which nuisance parameters.  It appears this is what gives you the matching rates compared to the oracle estimator that knows the clusters in advanced. For more information see the limitation section below.

**Minor notes**
* "asymptotically normally" at 265
"retain 50 predictors" or 51?
* R.146 space. before intuitively.
* It's not clear to me what "DOLLAREL" is.

---

> ### Author Rebuttal · Authors · 2026-03-31
>
> **Hyperparameter tuning.** Thank you for this comment. Our method is fairly robust to the choice of the hyperparameters $(\epsilon_n,\tau,c_w,\gamma)$. In particular, Theorem 3.5 only requires $\epsilon_n$ to be sufficiently small, so in practice we typically set it to $0$ or a numerically negligible value. The same theorem also suggests that the procedure is stable over a broad range of $\tau$, which is consistent with our empirical findings. Our experiments also suggest that one may often simply take $\lambda_{jj'} = w_{jj'} = c_w ||\theta_j - \theta_{j'}||^{-\gamma}$ akin to adaptive lasso, i.e., omit $(\tau,\epsilon_n)$ altogether; however, we include them mainly for simplicity of theoretical analysis. For the remaining parameters, we have found empirically that $\gamma \in [2,4]$ works well, and for numerical stability we recommend $\gamma = 2$ or $2.5$. Our results are also stable over $c_w$ in the range $[0.1,10]$. To summarize, the main tuning parameters are $(c_w,\gamma)$, with $c_w$ robust over a broad range and $\gamma = 2$ or $2.5$ working well in our experiments. We will add a detailed discussion in the revised manuscript.
>
> **Def of $[h, h]$.** Thank you for pointing this out. You are correct that we have not defined the notion $[h, h]$ explicitly and will clarify it in the revised version. Here, $D_\eta^2 f(\eta)[h_1,h_2]$ denotes the second Gateaux derivative of $f$ with respect to the nuisance parameter $\eta$, evaluated in the directions $h_1$ and $h_2$, namely $$D_\eta^2 f(\eta)[h_1,h_2]=\left.\frac{\partial^2}{\partial s\,\partial t}f(\eta + s h_1 + th_2)\right|_{s=t=0}.$$
> Accordingly, $D^2\_\eta f(\eta)[h,h]$  means that the second derivative is evaluated twice in the same direction $h$. We will explicitly include this definition in the revised manuscript.
>
> **Dissimilarity of predictors.** Thank you for this insightful comment. We agree that this distinction should be stated more clearly when comparing our work with Duan and Wang (2023). In our current main theorems, we assume that the *target parameters* are exactly equal within each cluster, whereas Duan and Wang allow a more flexible notion of task-relatedness. However, during the rebuttal period, we extended Theorems 3.5 and 3.6 to a near-homogeneous clustered setting, in which the target parameters within a cluster need not be exactly identical. Specifically, for each cluster $S_k$, we allow $$||\theta\_j^\ast - \beta\_k^\ast||\_2 \le \xi\_k,  \qquad j \in S_k,$$
> where $\xi_k$ measures within-cluster heterogeneity. Importantly, this extension still **does not** require the number of clusters to be known, and the numbers of tasks $m$ and clusters $K$ may both grow with sample size. Under Assumptions 3.1-3.3, with the same choice of $(\epsilon_n,\gamma)$ as before, an appropriate choice of $\tau$, and provided $\xi_k \le c_\xi \min_{j \in S_k} n_j^{-1/2}$, we show that: (i) the estimator still achieves exact recovery of the latent clustering, and (ii) for every $j \in S_k$, $$||\hat\theta\_j - \theta\_j^\ast||\_2\le C(1+b\_m)K^{1/r\_1}N_k^{-1/2}t + \tilde C N\_k^{\zeta-1} + 2\xi\_k,$$ with probability at least $1 - t^{-1} - p_{\delta/4}(n_{\min})$, where $$b_m=K^{1-1/r_1} m_k^{1/2} \max_{j \in S_k} n_j^{1/2} s(n_j)^{-2},$$ $\zeta < 1/2$, and $p_{\delta/4}(n_{\min})$ is the same quantity as in the original theorem. Here $m_k = |S_k|$ and $s(n_j)$ denotes the nuisance estimation rate. Thus, the estimation error is the pooled oracle term plus an additional bias term due to within-cluster heterogeneity. In the special case where each cluster contains roughly $m/K$ tasks, each task has sample size $n$, and the nuisance is estimated at rate $n^{-\rho}$, this bound becomes $$||\hat\theta_j - \theta_j^\ast||_2 \lesssim K^{1/r_1} \sqrt{\frac{1 + K^{1-2/r_1} m n^{1-4\rho}}{mn/K}} + \xi_k.$$ Therefore, our result extends Theorem~4.4 of Duan and Wang to the semiparametric setting with heterogeneous, possibly infinite-dimensional nuisance parameters. Notably, both the dimension and structure of the nuisance component may vary from task to task, so the framework is more flexible than standard clustered parametric MTL formulations. Moreover, if $\xi_k = o(N_k^{-1/2})$, then the asymptotic normality result of Theorem 3.6 continues to hold; otherwise, pooled-rate asymptotic normality may fail. We will revise the manuscript to make this distinction and the new extension more explicit.
>
>
> **Minor notes.** Thank you for carefully reading the paper and for providing these detailed comments. We have addressed all of them in the revised version. In particular, we corrected "asymptotically normally", fixed the spacing issue before "Intuitively", clarified the predictor count, and added that "DOLLAREL" is the RECS survey code for total electricity cost in dollars.

---

> > ### Author Rebuttal · Reviewer_JS9f · 2026-04-04
> >
> > I thank the authors for their thoughtful reply.
> >
> > In reading the above my concerns and questions have been addressed. I maintain my score and I'll keep an eye on further discussions on the other threads. I think that the relaxation of the dissimilarity of predictors will resolve concerns that a reader may have about this residing the special structure that makes everything go through. Given the relaxation, the result is clearly more general.

---

> > > ### Author Response · Authors · 2026-04-04
> > >
> > > Thank you very much for your thoughtful review and for taking the time to carefully consider our rebuttal. We sincerely appreciate your positive feedback and are glad that our responses have addressed your concerns.
> > >
> > > We are particularly grateful for your comments on the relaxation of the dissimilarity of predictors, and we agree that this extension broadens the applicability of our results. We appreciate your careful consideration and would be happy to provide any further clarification if helpful.

---

### Official Review · Reviewer_bVQb · 2026-03-15

**Soundness:** 3
**Presentation:** 3
**Significance:** 3
**Originality:** 3
**Overall Recommendation:** 4
**Confidence:** 3

**Summary:**

The work considers clustered multitask learning in a semi-parametric setting where tasks share a latent cluster structure in their target parameters but exhibit heterogeneous, potentially infinite-dimensional nuisance components. Such heterogeneity poses a major challenge for existing multitask learning methods, which typically rely on aligned feature spaces or homogeneous task structures.To address this challenge, the work propose an adaptive fused orthogonal estimator that integrates Neyman-orthogonal losses with data-driven pairwise fusion penalties.  The work theoretically shows that the proposed estimator achieves exact recovery of the latent clustering with high probability and attains pooled parametric convergence rates proportional to cluster size,and empirically shows that the proposed method consistently outperforms strong baselines in various simulation setups and produces interpretable and statistically significant clusters in real data, thus serving as an effective tool that combines both prediction and inference functions.

**Compliance With Llm Reviewing Policy:**

Affirmed.

**Key Questions For Authors:**

1. The theoretical analysis relies on a set of regularity conditions. In practice, especially with complex nuisance components, how can users verify these conditions, and if violated, does the method still provide reliable results? A clear response on practical diagnostics or robustness would strengthen confidence in the method's real-world applicability.
2. The empirical comparison focuses on ARMUL and the personalized estimator. How does the proposed method compare conceptually and empirically to broader MTL families? A clear positioning relative to the broader MTL literature would clarify the contribution, and lack of such positioning may raise questions about incremental novelty.
3. The real data analysis identifies three clusters interpreted through climate zones, but provides no comparison with existing methods. How can we determine whether the identified clusters reflect genuine advantages over baseline methods rather than patterns that could be obtained from domain knowledge alone? A demonstration that the method reveals insights not captured by baselines would strengthen claims of practical utility. Absent this, the findings remain illustrative rather than evidence of effectiveness.

**Limitations:**

yes

**Strengths And Weaknesses:**

Strengths:
1. The submission is technically sound. The work provides both theoretical analysis and experimental validation for the proposed method. Theoretically, under reasonable assumptions, the study verifies the method's capabilities in three aspects: cluster recovery, estimation accuracy, and inference, with the derivation process being largely correct. Experimentally, through simulation studies, the method demonstrates superior performance compared to Task-individual/Personalized and ARMUL across three canonical semiparametric models: PLM, ATE, and DID. Furthermore, by applying the method to the 2020 Residential Energy Consumption Survey (RECS), the study confirms its ability to generate interpretable and statistically significant clusters from real data, substantiating to some extent its dual value as a tool for prediction and inference.
2. The submission is clearly written and well structured.The overall narrative is easy to follow.The work properly position itself in the context of prior/concurrent literature and clearly discuss how it differs.
3. The paper addresses the challenge of clustered multitask learning in a semi-parametric setting, where tasks share a latent cluster structure in their target parameters but are characterized by heterogeneous, potentially infinite-dimensional nuisance components. It highlights that this heterogeneity poses a significant limitation for existing multitask learning methods, which often depend on aligned feature spaces or homogeneous task structures.The method enables efficient estimation of shared targets under heterogeneous nuisance structures, achieving exact cluster recovery, pooled-rate accuracy, and asymptotic normality.The experimental results illustrate its utility as both a predictive and inferential tool.
4. The work introduce a two-stage adaptive multitask learning framework that combines Neyman-orthogonality with data-driven pairwise fusion, enabling principled information sharing across tasks with heterogeneous nuisance structures.The reasoning behind this combination is well-articulated.The work demonstrates that the DML approach has since become a central tool in modern causal inference and semiparametric machine learning and a recent seminal work has developed statistically principled parametric multitask estimators with adaptive clustering guarantees, under known number of clusters.But the DML approach is a single-task procedure, as it neither leverages cross-task similarities directly nor discovers shared structure across multiple environments, and those clustered multitask learning approaches typically assume parametric models and do not accommodate complex or infinite-dimensional task-specific nuisance components. The work bridges the gap in the cross-section of semiparametric multi-task learning and heterogeneity clustering.

Weaknesses:
1. Due to the limited number of baseline methods used for comparison and the lack of comparative experiments on real data, the experimental results may not fully support its conclusions.

---

> ### Author Rebuttal · Authors · 2026-03-31
>
> **Comparison with other baselines.** Thank you for the comment. We added three standard baselines: (i) CN (Jacob et al., 2008), (ii) FC (Zhou & Zhao, 2015), and (iii) MeTaG (Han & Zhang, 2015). These figures and tables can be found at: https://anonymous.4open.science/r/submission-120D. Across all models (PLM, ATE, DID) and separation levels, our adaptive method attains the lowest RMSE while maintaining near-perfect ARI. CN and FC have substantially higher RMSE and generally fail to recover the cluster structure, largely because their squared $\ell_2$ penalties do not induce effective shrinkage across related tasks. MeTaG is closely related to our fixed-$\lambda$ ablation in Appendix I: it can achieve moderate clustering performance, but its inflated RMSE results from its global tuning parameter $\lambda$ failing to achieve an optimal bias-variance trade-off. Overall, our adaptive method achieves a better balance between accurate estimation and reliable cluster recovery.
>
> **Robustness of regularity conditions.** Thank you for this comment. We agree that diagnostics and robustness should be discussed more clearly. Our assumptions are mild and typical in semiparametric estimation and orthogonal learning: Assumption 3.1 is a local curvature/smoothness and finite-moment condition (valid for a wide range of losses), Assumption 3.2 requires nuisance estimation faster than $o(n_j^{-1/4})$ (necessary for orthogonal learning), and Assumption 3.3 only requires consistency of the pilot estimator. In practice, we recommend (i) assessing nuisance quality via cross-fitting and (ii) checking stability of the estimated clustering across reasonable nuisance learners and moderate changes of $(c_w,\gamma,\tau,\epsilon_n)$. These checks help assess whether nuisance error dominates the orthogonalized estimator and whether clustering/inference is stable. If some assumptions fail, the method may still work empirically, but Theorems 3.5 and 3.6 may not hold. For example, if the nuisance is too irregular to achieve the $n_j^{-1/4}$ rate, pooled $\sqrt{N_k}$-rate inference may fail; if the initial estimator is poor, finite-sample cluster recovery may deteriorate. We will add this discussion to the revised version.
>
> **Positioning in the broader MTL literature.** Thank you for this comment. We agree that the positioning relative to the broader MTL literature should be stated more clearly. Many MTL methods are prediction-oriented and rely on shared representations, aligned feature spaces, homogeneous models, or fixed parameter-sharing structures. Our setting differs in two ways: nuisance components may be heterogeneous and even infinite-dimensional across tasks, and our goal is not only improved estimation but also valid inference for a low-dimensional target parameter. Broad prediction-oriented MTL methods are therefore not directly comparable, since they typically do not provide guarantees for semiparametric inference under heterogeneous nuisance structure. Our benchmarks were therefore a no-sharing baseline (personalized estimation) and the closest statistically principled clustered-MTL method, ARMUL. ARMUL is especially relevant because it provides rigorous guarantees, but in a simpler parametric setting with a known cluster count. Our main contribution is to extend this line by allowing (i) unknown cluster number, (ii) fully task-specific, possibly nonparametric nuisance components, and (iii) asymptotic normality for valid inference after adaptive clustering. During the rebuttal period, we also now extend the theory to near-homogeneous clusters, where within-cluster parameters need only be close rather than identical. In the no-nuisance special case, our rate recovers that of Duan and Wang. To the best of our knowledge, this is the first work to address these challenges together within a unified theoretical framework.
>
> **Real data analysis.** Thank you for this comment. We agree that cluster identification alone does not establish a practical advantage over baselines. We now compare methods quantitatively using prediction MSE on held-out data. In addition to RECS, we include a second real-data study based on the 2018 CBECS dataset (details are in the PDF linked above). As in RECS, we fit a partially linear model with $Y$ equal to log annual electricity consumption, $D$ equal to log square footage, and $X$ consisting of 24 additional building characteristics; tasks are defined by principal building activity, yielding 27 tasks. We compare our method with two baselines: (i) a personalized approach and (ii) clustered ARMUL. Since ARMUL is for parametric models and does not allow task-specific heterogeneity, we implement it using a full linear model rather than a partially linear one. The results are reported in Table 2 for CBECS and Table 3 for RECS. Across both datasets, our method achieves low held-out MSE, showing that it combines clustering with task-specific heterogeneity to improve out-of-sample prediction.

---

> > ### Author Rebuttal · Reviewer_bVQb · 2026-04-04
> >
> > I have carefully reviewed the authors' rebuttal and supplementary experimental results. All core concerns have been thoroughly resolved with substantial revisions and supplementary analyses. I maintain my positive evaluation of this work.

---

> > > ### Author Response · Authors · 2026-04-04
> > >
> > > Thank you very much for your thoughtful review and for taking the time to carefully read our rebuttal and supplementary materials. We truly appreciate your positive feedback and are glad that our revisions and additional analyses have addressed your concerns.
> > >
> > > We appreciate your careful consideration and would be happy to provide any further clarification if helpful.

---

### Decision · Program_Chairs · 2026-04-30

**Decision:**

Accept (regular)

**Comment:**

This paper studies clustered multitask learning in a semiparametric approach, where tasks share low dimensional target parameters but have heterogeneous, possibly high  or infinite dimensional nuisance. The authors propose an adaptive two stage method that combines Neyman orthogonal losses with data driven fusion penalties, allowing both clustering and valid inference to be carried out in a unified framework.

Reviewers agree that the problem is well motivated and important. The technical approach is thoughtful and nontrivial: the idea of fusing only the target parameters while keeping nuisance estimation task specific is well aligned with the inferential goals, and the use of orthogonality is appropriate and carefully handled. Theoretical results cover cluster recovery, pooled rate estimation, and asymptotic normality, and are supported by detailed analysis. Several reviewers raised questions about assumptions (e.g., exact clustering, nuisance regularity), empirical scope, and positioning relative to other multitask learning methods. These concerns were addressed seriously in the rebuttal.

Overall, the work represents a meaningful advance at the intersection of multitask learning, semiparametric inference, and causal ML, and it is likely to be built upon by future work.